# River flooding mechanisms and their changes in Europe revealed by explainable machine learning

Shijie Jiang[1], Emanuele Bevacqua[1], Jakob Zscheischler[1]

[1]Department of Computational Hydrosystems, Helmholtz Centre for Environmental Research, Leipzig, 04318, Germany

*Correspondence to*: Shijie Jiang (shijie.jiang@ufz.de)

**Abstract.** Climate change may systematically impact hydro-meteorological processes and their interactions, resulting in changes in flooding mechanisms. Identifying such changes is important for flood forecasting and projection. Currently, there is a lack of observational evidence regarding trends in flooding mechanisms in Europe, which requires reliable methods to disentangle emerging patterns from the complex interactions between flood drivers. Recently, numerous studies have demonstrated the skill of machine learning (ML) for predictions in hydrology, e.g., for predicting river discharge based on its relationship with meteorological drivers. The relationship, if explained properly, may provide us with new insights into hydrological processes. Here, by using a novel explainable ML framework, combined with cluster analysis, we identify three primary patterns that drive 53,968 annual maximum discharge events in around a thousand European catchments. The patterns can be associated with three catchment-wide river flooding mechanisms: recent precipitation, antecedent precipitation (i.e., excessive soil moisture), and snowmelt. The results indicate that over half of the studied catchments are controlled by a combination of the above mechanisms, especially recent precipitation in combination with excessive soil moisture, which is the dominant mechanism in one-third of the catchments. Over the past 70 years, significant changes in the dominant flooding mechanisms have been detected within a number of European catchments. Generally, the number of snowmelt-induced floods has decreased significantly whereas floods driven by recent precipitation have increased. The detected changes in flooding mechanisms are consistent with the expected climate change responses, and we highlight the risks associated with the resulting impact on flooding seasonality and magnitude. Overall, the study offers a new perspective on understanding changes in weather and climate extreme events by using explainable ML and demonstrates the prospect of future scientific discoveries supported by artificial intelligence.

## 1 Introduction

River flooding is a pervasive natural hazard that regularly causes substantial economic, societal, and environmental damages worldwide (Tellman et al., 2021; Merz et al., 2021). With a warming atmosphere, flooding risk is projected to increase due to an intensification of the water cycle over large areas (Hirabayashi et al., 2013; Alfieri et al., 2017). For Europe, large-scale studies have revealed changes in flooding frequency, seasonality, and magnitude over the past decades, with considerable variations across catchments (Blöschl et al., 2017; Hall and Blöschl, 2018; Bertola et al., 2020; Blöschl et al., 2019; Alfieri et

al., 2015). The spatial inconsistency in these trends reflects differences in flood generating processes across the continent, which underscores the need for a better understanding of flood drivers (Keller et al., 2018).

In recent years, numerous studies have investigated river flooding mechanisms and some of them have provided European-scale assessments (e.g., Berghuijs et al., 2019; Kemter et al., 2020; Bertola et al., 2021; Berghuijs et al., 2016; Stein et al., 2020). Catchment-level floods can typically be attributed to the interaction of hydro-meteorological processes, such as extreme precipitation, soil moisture excess, and snowmelt (Merz and Blöschl, 2003; Tarasova et al., 2019). The dominant controlling processes in catchments were usually identified either qualitatively by comparing the observed flood trends with the

contemporaneous changes in flooding drivers (e.g., Blöschl et al., 2019; Blöschl et al., 2017) or quantitatively by calculating the seasonal similarities between flood events and potential drivers (e.g., Berghuijs et al., 2019; Berghuijs et al., 2016). Such analyses revealed the dominant flood generating processes at a catchment level, improving the understanding of climate change effects on flooding magnitude and timing. However, the methods often implicitly assume temporally consistent flood processes within a catchment (Merz et al., 2012), making it difficult to detect possible changes in flooding mechanisms themselves in a

warming climate.

Flooding mechanisms that dominate one catchment are not always immutable but might shift over time, particularly in light of climate change (Hall et al., 2014). For example, increasing temperatures can affect snow dynamics in cold regions and result in more rainfall extremes, which could make snowmelt-dominated catchments more susceptible to extreme rainfall and thereby

alter the regional flood seasonality and magnitudes (Davenport et al., 2020; Rottler et al., 2021; Vormoor et al., 2016). Therefore, a systematic investigation of the changes in flooding mechanisms is necessary. Yet few studies have been able to quantify how the mechanisms evolved over time on a continental scale in Europe. The identification of specific trends in flooding mechanisms requires a comprehensive understanding of hydrological processes underlying individual events (Stein et al., 2020). Currently available studies that attempted to classify river flooding processes on an event basis typically rely on

multicriteria approaches, which require predefining thresholds for a variety of hydrometeorological indicators, such as the storm duration and snowmelt amount (e.g., Nied et al., 2014; Stein et al., 2021). Using a multicriteria approach, Kemter et al. (2020) identified the flooding mechanisms in Europe by classifying approximately 174,000 flood peaks and revealed their trends over the past 50 years. Likewise, Stein et al. (2020) analyzed flood events over 4,155 catchments worldwide and classified them into five flood-generating processes. Despite the computational efficiency of using multicriteria approaches,

the obtained insights are often dependent on the careful choice of indicators and thresholds. For example, in some cases, a small change in a threshold value modifies the classification, potentially compromising the robustness of the results (Sikorska et al., 2015). Alternatively, some studies grouped flood events by inductive analyses, which adopted clustering methods to obtain flood types from hydrometeorological indicators (e.g., Turkington et al., 2016; Keller et al., 2018). However, the chosen indicators (e.g., snow-covered area, day of occurrence, and 95th percentile of spatial precipitation distribution) did not

unambiguously indicate flooding mechanisms since they were not indicative of the causal contribution of flood drivers to peak discharges (Tarasova et al., 2019).

An effective way to identify flooding mechanisms for individual flood events is to quantify the contribution of possible drivers to its occurrence, which involves uncovering the implicit connections that may exist between flood events and meteorological

observations. This can be achieved by machine learning (ML), which has been receiving increasing attention in Earth and climate sciences for its remarkable ability to identify and generalize predictive relations with a high-level abstract representation (Reichstein et al., 2019; Yu and Ma, 2021). In hydrology particularly, one excellent example is the prevalence of long short-term memory (LSTM) neural networks (Kratzert et al., 2018; Shen, 2018), which have been demonstrated to learn patterns conceptually consistent with qualitative understandings of how hydrological systems work as opposed to simply

trivial coincidences (Kratzert et al., 2019a). Extraction of captured patterns from "black-box" ML models with feature attribution techniques (i.e., ML interpretations) may lead to theoretical advances and can assist in making new scientific discoveries, as recently demonstrated for climate, ocean, and weather applications (e.g., Toms et al., 2020; Barnes et al., 2020; Labe and Barnes, 2021), including the identification of flooding mechanisms (Jiang et al., 2022).

In this study, we revisit flooding mechanisms in Europe over the period 1950–2020 by using an improved framework based on the explainable ML methods developed by Jiang et al. (2022) and compare the results with existing studies. We base the analysis on around 1,000 catchments and the only dynamic information necessary for the analysis is precipitation, temperature, and streamflow. These three variables can be readily measured, thereby reducing the reliance on possibly uncertain estimations of fluxes and state variables (such as soil moisture). The combination of supervised learning-based feature attribution and

unsupervised learning-based cluster analysis reduces subjectivity and uncertainty for the selection of appropriate indicators and thresholds in the categorization of flood drivers. Moreover, taking an event-level perspective, we quantify the changes that occurred in these mechanisms in the past seven decades, and discuss the possible reasons and implications of the detected changes. Overall, the study contributes to a better understanding of river flood risk and how it is affected by climate change and illustrates how explainable ML can advance knowledge about the Earth system.

## 2 Data and methodologies

### 2.1 Data

The study considers 1,077 catchments in the domain of Europe (Fig. 1a) based on the data availability of daily river discharge observations from the Global Runoff Data Centre (GRDC) dataset (https://www.bafg.de/GRDC). We restricted our analysis to catchments having a minimum of 20 years of discharge records within 1950–2020 to ensure sufficient samples to train the

ML models. The catchment areas range between 8 km$^2$ and 10,000 km$^2$ — very large catchments, where the effect of spatial heterogeneity of flood drivers tends to be substantial, were not considered. For those catchments, the sample size of daily

discharge records ranges from 7,300 to 25,753, with a median of 20,455 time steps. Overall, the selected catchments encompass a variety of geographical and climatic conditions, as illustrated by the catchment distributions in terms of average elevation, average slope, catchment size, aridity index, snowfall fraction, and flood mean date (Fig. 1). The elevation, slope, and size

were derived from the Global Streamflow Indices and Metadata Archive (GSIM) (Do et al., 2018), the aridity index and snowfall fraction were calculated from the catchment-averaged precipitation and temperature described later. In the study, floods are defined as the annual maxima (peaks) of river discharge time series in line with common practices (e.g., Blöschl et al., 2019; Blöschl et al., 2017). The above properties will also be used to discuss their relevance to the catchment-level dominant flood mechanisms.


We considered precipitation, temperature, and day length as input variables of the ML models. Using the 0.1° daily gridded precipitation and mean surface temperature data from the E-OBS dataset (version 23.1e) (Haylock et al., 2008), we calculated the catchment-averaged time series of these variables based on area-weighted averages of the data pixels within the catchment boundary. The weight of each pixel was determined by the fraction of its area covered by the relevant catchment. The

catchment boundaries were obtained from readily available GRDC (Lehner, 2012) and GSIM (Do et al., 2018) databases, with GRDC being prioritized when the boundary of a catchment was available in both databases. Note that for smaller catchments under 100 km$^2$ (approximately 0.1° × 0.1°), uncertainties may exist due to the relatively coarser spatial resolution of the meteorological data. Nonetheless, those catchments with large uncertainties will not be considered for the subsequent attribution analysis if ML models cannot capture the relationship between inputs and outputs effectively. Day length was

included in the study since it was shown to improve model accuracy in a series of preliminary tests, including the cases where only precipitation and temperature were used and day length was additionally incorporated. Catchments where day length largely improves accuracy are mainly located in northern Europe. Day length was calculated based on the day of the year and the latitude of the catchment center by the Brock model following Forsythe et al. (1995).

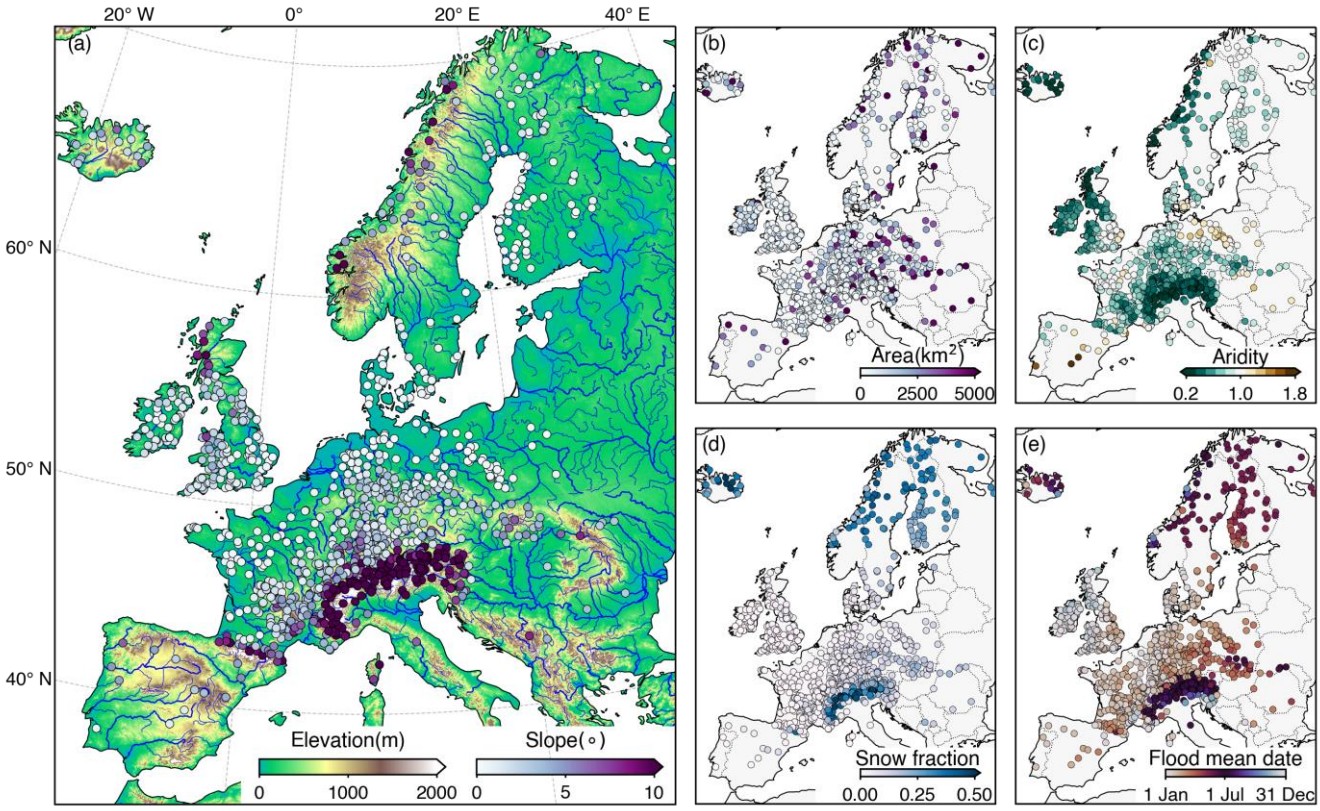

**Figure 1: An overview of the 1,077 catchments and their properties, including average (a) elevation and slope of the catchments, (b) the catchment size, (c) the aridity index, expressed by the ratio between mean annual potential evapotranspiration (PET) over mean annual precipitation, (d) the fraction of precipitation falling as snowfall (i.e., precipitation falling with temperature below 0 °C), and (e) the seasonality of annual maximum discharges. PET was estimated via Hamon's formulation (Hamon, 1961).**

## 2.2 Attribution framework and ML model

Figure 2 illustrates the framework of using explainable ML methods for flooding attribution in the present study, which was originally developed by Jiang et al. (2022) and involves three main steps. First, we built ML models for individual catchments to establish the nonlinear predictive maps from meteorological factors (i.e., precipitation, temperature, and day length) to daily discharges (Fig. 2a). Secondly, an ML interpretation technique was applied to interpret the trained models to quantify the contributions of the three input variables at each time step (i.e., time-wise feature importance) to the generation of respective flood events (Fig. 2b). The time-wise feature importance was further aggregated into contributions of specific features. Finally, cluster analysis was used to group the specific feature contributions from multiple flood events that had similar patterns into several categories, from which we then identified different flood mechanisms. Detailed explanations of the methods are given below.

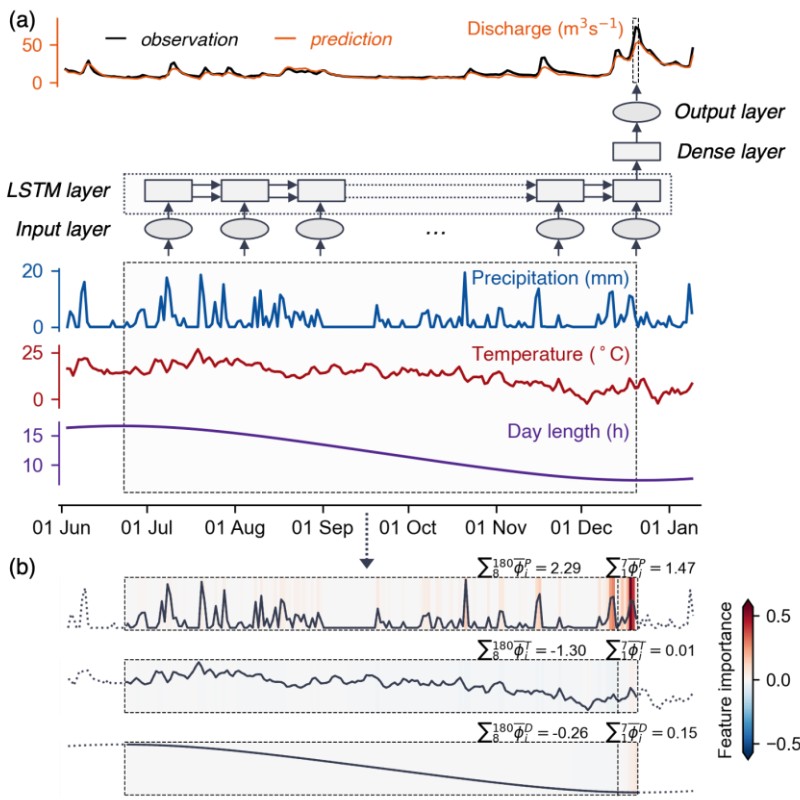

**Figure 2: The workflow of using explainable ML methods for attributing flood peaks (annual maxima of river discharge) to their drivers. (a) Diagrammatic representation of the used LSTM models. The window in the time series of discharge highlights the target output (which is a point) and the window in the inputs indicates the input features used to predict the illustrated peak discharge sample. (b) The feature importance of the inputs for predicting the peak discharge shown in (a), which was obtained by using the ML interpretation technique (namely integrated gradient). The vertical dashed lines in the windows separate the feature importance into a recent 7-day period and an earlier period to calculate the aggregated feature contributions (see main text).**

In the study, we used the classical LSTM network (Hochreiter and Schmidhuber, 1997) as the ML model. The LSTM is one of the most popular ML architectures for modeling dynamic hydrological variables (e.g., Kratzert et al., 2018; Lees et al., 2021), which can effectively capture nonlinear and temporal dependencies between variables owing to its recurrent structure and unique gating mechanism (Gers et al., 1999). The effectiveness of the LSTM is partially due to the comparability of its formulation to the hydrological behavior of a catchment. Specifically, the backbone of the LSTM network is composed of recurrent cells that can store previous information from input sequences, which is conceptually similar to the way meteorological information (e.g., precipitation) is stored in the form of soil moisture or snowpack (Lees et al., 2022). The physically realistic mapping from inputs to outputs facilitates gaining hydrologically meaningful insights from subsequent model interpretations. Figure 2a illustrates the data flow of one sample in the LSTM model, with the dashed windows highlighting the predictors and the target variable. The input layer of the model brings in precipitation ($P$), temperature ($T$),

and day length ($D$) over the past 180 days (i.e., $[X_1^P, X_2^P,..., X_{180}^P; X_1^T, X_2^T,..., X_{180}^T; X_1^D, X_2^D,..., X_{180}^D]$) and the output layer produces the discharge of the same day (i.e., $y_1$). Note that we included predictors on the same day as the output in the model, since precipitation on that day could also affect the discharge, especially in small catchments with quick catchment response times. However, the conclusions do not change even if using LSTM models to predict discharge on the next day (i.e., the prediction models consider the lagged meteorological forcings up till the day before each daily discharge). The hidden layers consist of a single LSTM layer and a dense layer with 32 units. The number of time steps and hidden units were determined by considering both the model performance and efficiency, which had been evaluated in preliminary experiments. Preliminary experiments also suggest using fewer time steps (e.g., 90 days) would not impair the conclusions of the study about flooding mechanisms, because contributions from inputs at very early time steps to output are limited in LSTM models (i.e., memory decay) (Su and Kuo, 2019). Here, we skip the technical details of the LSTM architecture and refer to Sherstinsky (2020) for a comprehensive explanation of the fundamentals of LSTM networks.

To improve the robustness of model evaluation and analysis, we fitted 10 independent LSTM models for each of the 1,077 catchments. Specifically, the data for each catchment was divided into 10 folds without shuffling the temporal sequence, and each fold was tested once with a model trained with the remaining 9 folds. The predictive performance of each model was evaluated independently based on testing data, i.e., 1/10 of the data for each catchment, which ranged from 2 to 7 years due to the 20–70 years of sample size available in studied catchments. During the training process, a portion of the training data (70%) was repeatedly used to update the model parameters every epoch until no further decrease in the loss function was observed on the remaining 30% (also known as validation data). The initial learning rate and maximum training epoch number were configured to 0.01 and 200, respectively, with the adaptive moments estimation (Adam) algorithm (Kingma and Ba, 2015) being used for training the models.

## 2.3 Model interpretations and cluster analysis

The integrated gradient (IG) technique developed by Sundararajan et al. (2017) was employed to interpret the trained models, which allows for obtaining the time-wise feature importance of the three input variables for each sample of the output (i.e., daily discharges). The IG method is a gradient-based interpretation technique that exploits the gradient of the model's output to its input features to trace back the specific contributions of the inputs. It aims to assign an importance score to each feature (e.g., to the precipitation at each time step prior to the flooding). A large positive score indicates that the feature substantially increases the network output (e.g., that the precipitation at a certain time step contributes to increasing the flooding), a large negative score indicates a decrease in the network output, and a score close to zero indicates little influence on the output. The IG score for the input feature $x$ (e.g., precipitation at the $i$-th time step) is formulated as:

$$\phi_i(x) = (x_i - x_i') \int_{\alpha=0}^{1} \frac{\partial f(x' + \alpha(x - x'))}{\partial x_i} \, d\alpha \tag{1}$$

where $\frac{\partial f(x' + \alpha(x - x'))}{\partial x_i}$ denotes the local gradient of the network $f$ at a point interpolated from a baseline input ($x'$, when $\alpha = 0$), which is meant to represent the "absence" of feature input, to the target input ($x$, when $\alpha = 1$). An important property of the IG is completeness, which states that the IG scores add up to the difference between the output of $f$ at the target input $x$ and the baseline input $x'$, i.e., $\sum_i \phi_i(x) = f(x) - f(x')$. Therefore, the model output can be decomposed into the sum of features' individual contributions, and it enables us to examine the contribution of a group of features by summing up their individual IG scores.

In the study, we focus specifically on the IG scores for annual maximum peak discharge events to gain insights into flooding mechanisms. Given that we trained 10 independent models, 10 sequences of time-wise feature importance were generated for each peak discharge, with each sequence having the same dimensions as the input variables (i.e., $[\phi_1^P, \phi_2^P, ..., \phi_{180}^P; \phi_1^T, \phi_2^T, ..., \phi_{180}^T; \phi_1^D, \phi_2^D, ..., \phi_{180}^D]$). Then, the 10 sequences were averaged into one sequence (i.e., $[\bar{\phi}_1^P, \bar{\phi}_2^P, ..., \bar{\phi}_{180}^P; \bar{\phi}_1^T, \bar{\phi}_2^T, ..., \bar{\phi}_{180}^T; \bar{\phi}_1^D, \bar{\phi}_2^D, ..., \bar{\phi}_{180}^D]$, which is simplified as $\{\bar{\phi}_i\}$ hereafter) to reduce the impact of the stochasticity associated with training the different LSTMs. Figure 2b exemplifies the averaged IG scores corresponding to the sample shown in Fig. 2a, i.e., it shows the contribution of the three input variables to the selected annual maxima of river discharge. The warm or cool colors in the heatmap denoting the input variable at the particular time step has increased or decreased the network output, while white indicates little effect. Note that the averaged IG scores for an individual peak were computed by averaging the scores obtained from all the independent 10 models, regardless of whether the peak was part of the training data or the testing data in the models. Overall, the IG scores extracted from the 10 models for each target peak discharge generally follow a similar pattern, though with inevitable differences due to randomness and uncertainties in training processes (see Figs. S1–S3 in the Supplementary Material for examples). Note that using the IG scores based on the target peaks in testing datasets alone does not yield substantial impacts on our conclusion in subsequent analyses (see Figs. S4–S5 in the Supplementary Material).

In the following step, the sequences of averaged IG scores $\{\bar{\phi}_i\}$ can be clustered directly using time series clustering techniques based on their similar shapes, such as using the K-means method with the dynamic time warping algorithm (DTW) as the distance metric (Tavenard et al., 2020). However, the main drawback of clustering time series is the heavy computational burden. The DTW distance between any two samples has a quadratic time complexity with respect to the sequence length, which would make clustering long feature importance sequences a time-consuming process, and it would be especially challenging when dealing with tens of thousands of sequences (Salvador and Chan, 2007). Moreover, for this large-sample study that aims to understand flood mechanisms at a continental scale, it might not be necessary to distinguish the daily contributions of meteorological drivers in detail. Therefore, before carrying out the cluster analysis, we aggregated each sequence of averaged IG scores $\{\bar{\phi}_i\}$ by using a 7-day separating window, which generate a low-dimensional contribution

vector with only six elements $[\sum_1^7 \bar{\phi}_i^P, \; \sum_8^{180} \bar{\phi}_i^P, \sum_1^7 \bar{\phi}_i^T, \; \sum_8^{180} \bar{\phi}_i^T, \sum_1^7 \bar{\phi}_i^D, \; \sum_8^{180} \bar{\phi}_i^D]$. Here $\sum_1^7 \bar{\phi}_i$ and $\sum_8^{180} \bar{\phi}_i$ represent contributions of a variable in recent 7 days and an earlier antecedent period, respectively. The separating window size should cover the period of precipitation and snowmelt events leading to each peak discharge, which depends highly on the local characteristics. After examining the relationship between catchment area and mean event response time, Stein et al. (2020) suggested a synoptic window of 7 days should be sufficient to guarantee the response time for large catchments. As a result, this study used a 7-day period, similar to the practice in most studies that examined flooding causes (e.g., Blöschl et al., 2017; Berghuijs et al., 2019). However, using a shorter period (e.g., 5 days) does not affect the conclusions about dominant flooding mechanisms and their trends (see discussion in Section 3.7). Figure 2b demonstrates the values of the aggregated feature contributions based on respective daily IG scores represented by the heatmap.

To obtain an overall picture from the individual aggregated feature contributions, we used the K-means method to cluster the results for all annual maximum peak discharges pooled from all considered catchments. Considering that the feature importance values are correlated to the magnitude of the predicted peak discharge due to the completeness property, we normalized each accumulated vector by its Manhattan norm (i.e., dividing each element by the sum of its absolute values while keeping its sign) to make the contributions comparable across different floods. To determine the optimal cluster number for the K-means algorithm, we evaluated the cluster characteristics for candidate cluster numbers ranging from 2 to 8 using the silhouette coefficient (Rousseeuw, 1987), which reflects the separation distance between the resulting clusters. The silhouette coefficient for an individual sample is calculated as $(b - a)/\max(b - a)$, where $a$ represents the mean distance between the sample to all other points within the same cluster, and $b$ represents the mean distance between the sample and all other points in the next nearest cluster. The average silhouette coefficient over all samples is an indicator of the goodness of a clustering result, which ranges from -1 to 1, with a higher score generally indicating a better cluster number choice.

## 2.4 Trend analysis of flooding mechanisms

Based on the clustering results, we can identify the mechanism responsible for each annual maximum peak discharge and calculate the proportions of different flooding mechanisms at either the continental or catchment scale. The trend magnitude in these proportions was then analyzed by Theil-Sen's Estimator, with the modified Mann-Kendall test (Hamed and Rao, 1998) being used to determine the significance of the trend. Specifically, at the continental scale, we estimated the overall trends of various flooding mechanisms based on their respective proportions within all the annual maximum peak discharges per year. At the catchment scale, to capture the variations of flooding mechanisms over different periods, we calculated the proportion series using a 20-year moving window in each catchment. The 20-year time frame was used to ensure an adequate sample size for reliably estimating the intra-period proportions and also to guarantee enough periods to observe decadal variability (Pagano and Garen, 2005). Only proportions that were calculated with at least 10 years of peak discharge data in each window were used to estimate the trend slope.

Moreover, in order to analyze the possible causes of trends, we selected a number of regions where most catchments present consistent trends in certain mechanisms. We investigated those catchments exhibiting significant changes in flooding mechanisms and compared the temporal regional changes in flooding mechanisms with changes in potential flooding drivers. The time series of proportions in regions were calculated by applying the previously described 20-year moving window to

peak discharge classifications for the considered catchments. The flooding drivers considered include annual maximum 7-day total precipitation, mean spring temperatures (January to April), and 30-day precipitation preceding the 7-day window of recent precipitation, which is a common proxy for soil moisture prior to flooding (e.g., Bertola et al., 2021). All the drivers were averaged across the catchments and then smoothed by using a 20-year moving average window as well.

## 3 Results and discussion

### 3.1 Model predictive performance and interpretations

Before moving to the analysis of annual maximum peak discharges, we used the Nash-Sutcliffe efficiency (NSE) (Nash and Sutcliffe, 1970) to assess model accuracy in predicting discharges. The NSE value ranges from negative infinite to 1.0, and NSE > 0.5 is generally deemed satisfactory for discharge simulations (Moriasi et al., 2015). Based on the NSE value computed in the testing period for each model in the 10-fold cross-validation, we acquired the average and standard deviation of NSE

values for each of the 1,077 catchments, as shown in Fig. 3. The overall warm colors in the map (Fig. 3a) indicate that the model performed satisfactorily for most catchments, with the median of NSE averages reaching 0.74 (Fig. 3b). The standard deviations of NSE values (Fig. 3c) further indicate robust model performance in most cases. Accordingly, the models have effectively captured the generalizable predictive relationship between meteorological factors and discharges. As an accurate predictive relation is essential for deriving meaningful information from ML models (Murdoch et al., 2019), the subsequent

analyses focus specifically on the 977 catchments (out of 1,077; 91%) with average NSE values above 0.5. In the following, we move to the analysis of annual maximum peak discharges.

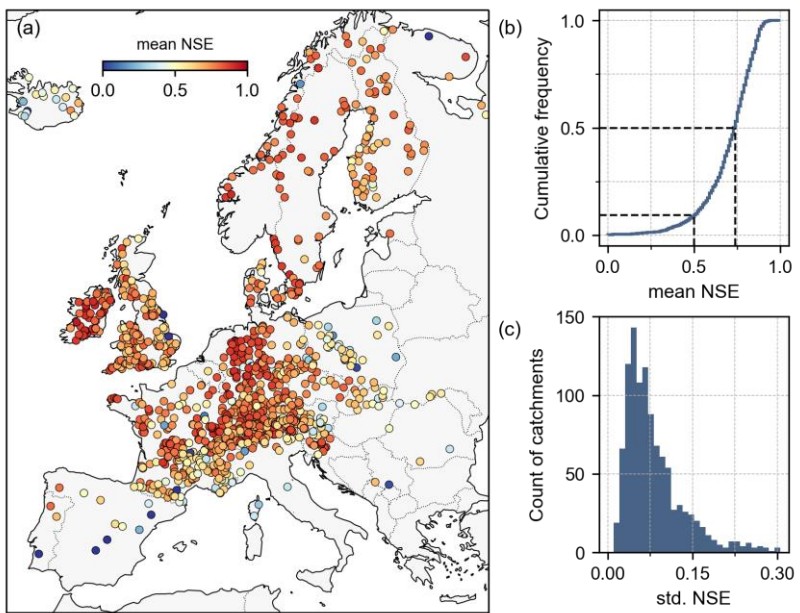

**Figure 3: (a) Nash-Sutcliffe efficiency (NSE) values in the testing period averaged over the 10-fold cross-validation. (b) The cumulative frequency of the averaged NSE values. (c) The distribution of the standard deviation values for the NSE values across the 10-fold cross-validation. The NSE values were calculated using all samples in respective testing datasets.**

A total of 53,968 annual maximum discharges were identified from the 977 catchments (20–70 peaks per catchment). By using the IG method, we can obtain 53,968 feature importance sequences averaged across the models from the 10-fold cross-validation. In the case shown in Fig. 2b, precipitation is the dominant driver behind the annual maximum peak discharge occurrence, showing consistently non-negative feature importance with the precipitation peaks that occur closer to the target flood peak having a greater influence (see pronounced positive contributions in red). Nevertheless, the total contribution from antecedent precipitation is more important in predicting the peak compared with the contribution from recent precipitation, as indicated by the aggregated scores $\sum_1^7 \bar{\phi}_i^P$ and $\sum_8^{180} \bar{\phi}_i^P$. The temperature, on the other hand, has an overall negative impact, which may be related to evapotranspiration that could decrease the discharge magnitude, while the influence of the day length is relatively negligible. Additionally, Fig. 4 further illustrates two other typical cases of feature importance patterns, where the contribution from recent precipitation (i.e., $\sum_1^7 \bar{\phi}_i^P$) and temperature (i.e., $\sum_1^7 \bar{\phi}_i^T$), respectively, is dominant in predicting target peak discharges. The distinct patterns of predictor contribution to annual maximum peak discharge predictions suggest that these flood events were triggered by different mechanisms.

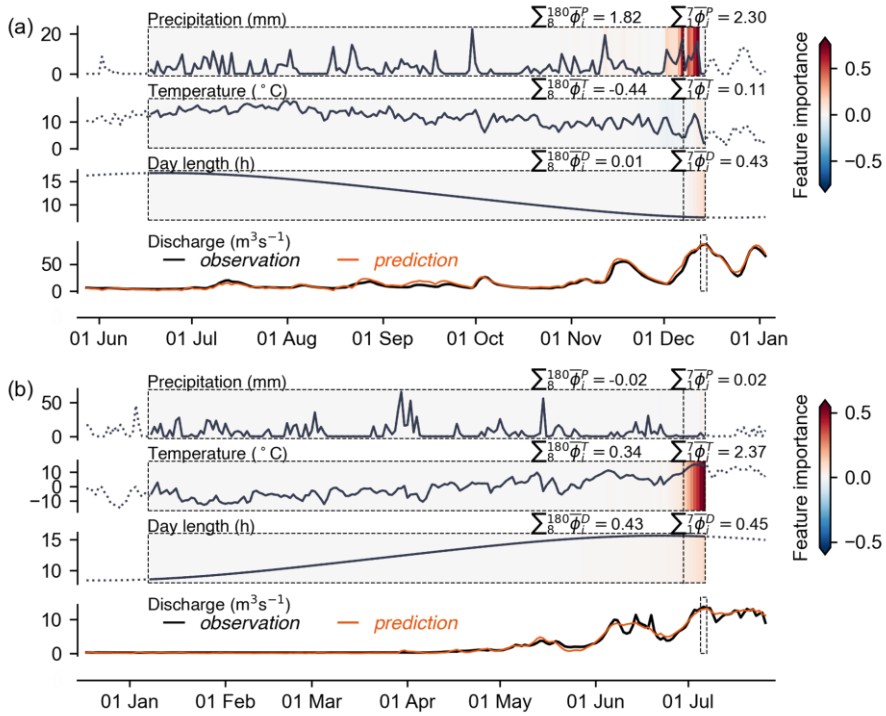

**Figure 4: Additional examples to the case shown in Fig. 2, which illustrate the importance pattern of temperature, precipitation, and day length in predicting two discharge peaks from other catchments. (a) Recent precipitation contributes most to the discharge peak. (b) Recent temperature contributes most strongly to the discharge peak.**

### 3.2 Flooding types revealed by cluster analysis

To separate the 53,968 annual maximum peak discharges into discrete groups characterized by distinct patterns of predictor contributions, we performed K-means clustering on the normalized contribution vectors. The results of the silhouette analysis suggest that clustering into three main groups would lead to the best clustering quality, because it achieves the high average silhouette coefficient and silhouette coefficients for individual samples are reasonably distributed within each cluster (see Fig. A1 in Appendix A for more details). It should be noted that the clustering results here only reveal major patterns widespread in data, with certain local and specific mechanisms unlikely to be detected.

Figures 5a–c show the distinct patterns of the three identified clusters, with cluster 1 featuring high importance of recent temperature (Fig. 5a, a positive contribution in line with high temperature favoring snowmelt), cluster 2 featuring the dominant contributions from recent precipitation (Fig. 5b), and cluster 3 featuring the importance of antecedent precipitation events (Fig. 5c). Compared to cluster 1, clusters 2 and 3 show a generally negative effect of antecedent temperature, in line with drying favored by evapotranspiration. Moreover, annual maximum peak discharges in cluster 1 are characterized by higher

contributions from day length (Fig. 5a) when compared to the other two clusters. The role of day length implies that the magnitude of these peak discharges can be partially explained by the seasonality presented by day length, which peaks around the June solstice. In contrast, the main differences between clusters 2 and 3 are due to the fractions of $\sum_1^7 \bar{\phi}_i^P$ and $\sum_8^{180} \bar{\phi}_i^P$. Overall, each cluster accounts for 15.5%, 49.9%, and 34.6% of all the identified annual maximum peak discharges, respectively.

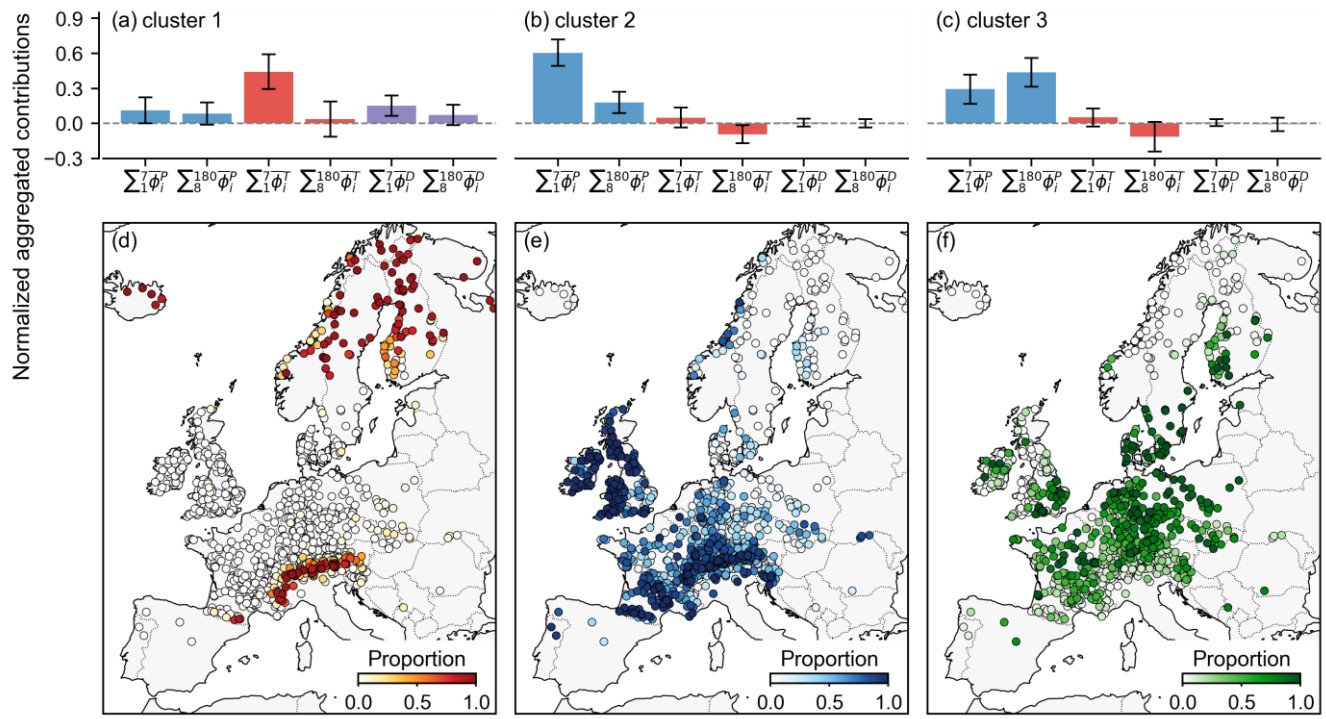

**Figure 5: The cluster centroids and variance for the three clusters and their respective proportions of all peak discharge events in each catchment. The bars and error bars in (a), (b), and (c) represent the cluster centroids and standard deviations of the six aggregated feature contributions. The proportions in (d), (e), and (f) correspond to clusters 1–3, respectively.**

Figures 5d–f illustrate the distributions in terms of the proportion of annual maximum peak discharges associated with each cluster within a catchment. Annual maximum peak discharges associated with high contributions from temperature (cluster 1) mainly occur in northern Europe and in mountainous regions such as the Alps (Fig. 5d), i.e., in regions with high snowfall

fractions (Fig. 1d) where rising air temperature can lead to snowmelt. The spatial distribution together with the feature pattern shown in Fig. 5a indicates that these floods were probably driven by snowmelt events. In contrast, catchments with cluster 2, where recent precipitation played a decisive role in causing most floods (Fig. 5b), are primarily located in regions that have a west-facing or north-west-facing coast or mountain range, such as Ireland, Scotland, Wales, the Norwegian coast, north-west of the Iberian Peninsula, as well as the area extending from the Alps, the Massif Central and the Pyrenees (Figs. 5e and 1a).

These regions are characterized by a generally humid climate (Schiemann et al., 2018), as also indicated by Fig. 1c, and are strongly affected by the Northern Atlantic polar front and the associated storm tracks (Bengtsson et al., 2006) and/or by the presence of mountain barriers perpendicular to the prevailing flow direction, which force moist air to lift and condense (Isotta et al., 2014). Previous studies indicate that flooding in the regions could be largely explained by individual heavy precipitation events (Gobiet et al., 2014; Whan et al., 2020; Blanchet and Creutin, 2017), some of which are associated with atmospheric rivers (Lavers and Villarini, 2013).

Catchments associated with cluster 3 are mostly located over the North European Plain, South Scandinavia, and parts of the British Isles (Fig. 5f). Here, information from antecedent precipitation has an overall higher weight than that from recent precipitation or other predictors (Fig. 5c), suggesting that recent precipitation alone would not suffice to explain annual maximum peak discharges. Therefore, flooding in these areas presents additionally heavy reliance on antecedent precipitation that is stored in the form of soil moisture. For example, Nied et al. (2014) revealed that in the Elbe River basin some weather patterns only cause flooding in case of preceding soil saturation. Also, Ledingham et al. (2019) found that in southeast England fewer than 15% of daily flood events correspond to extreme precipitation events, lower than in the rest of Britain, which was attributed to the relevant contribution of soil moisture storage to flooding.

It should be noted that the three kinds of flooding mechanisms (i.e., snowmelt-driven, recent precipitation-driven, and antecedent precipitation-driven) identified from the cluster analysis using the optimal cluster number only indicate which features carry greater weights for peak discharge predictions, and they are not necessarily mutually exclusive. Particularly, the peak discharge events near the decision boundaries between the three clusters, such as those with similar Euclidean distances to at least two different "closest" centroids, are likely affected by two or more flooding processes simultaneously. For example, the events categorized as snowmelt-driven floods are probably impacted additionally by saturated soils or extreme precipitation, such as rain-on-snow events (Cohen et al., 2015). These events generally represent compound flood events that arise from several drivers occurring concurrently (Bevacqua et al., 2021; Zscheischler et al., 2018). Recently, compound events have received increasing attention (Zscheischler et al., 2020), however, this study will only focus on the main flooding types obtained from the clustering results, regardless of whether compound effects were involved.

### 3.3 Dominant flooding mechanisms in Europe

The result of event-based flooding classification allows us to identify the dominant flooding mechanisms (among clusters 1–3, Fig. 5) for each catchment (Fig. 6a). A mechanism is considered dominant in a catchment if the proportion of the annual maximum peak discharges exceeded the maximum proportion of the other annual maximum peak discharges by more than 70%. Otherwise, the catchment was regarded as being dominated by a mixture of flooding mechanisms. The mixture of mechanisms could be further classified into specific combinations based on which clusters were present in the catchment. Accordingly, for the catchments investigated in the study, 52.1% were dominated by a mixture of mechanisms, while snowmelt,

recent precipitation, and antecedent precipitation solely accounted for 10.1%, 26.9%, and 10.9% of catchments, respectively. Among the mixtures of mechanisms, the combination of recent precipitation and antecedent precipitation accounted for 33.8%

of all the catchments, followed by the combination of all three mechanisms (15.8%), the combination of recent precipitation and snowmelt (2.1%), and the combination of antecedent precipitation and snowmelt (0.4%).

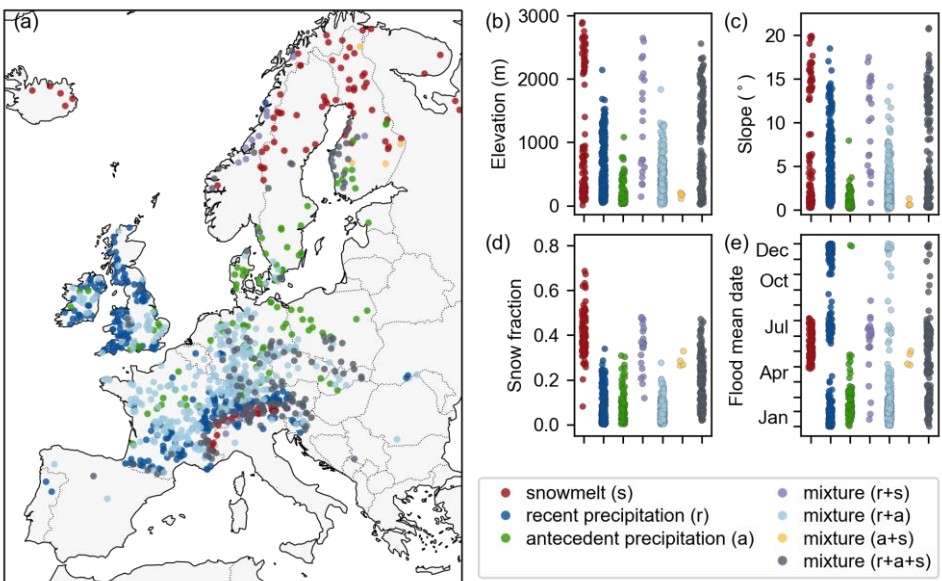

**Figure 6: The dominant flooding mechanisms and their relevance to catchment attributes and seasonality. Each dot in (b), (c), (d),**
**and (e) represents one catchment. Mixture means the associated catchments are dominated by two or more flooding mechanisms.**
**For example, mixture (r+s) indicates either recent precipitation (r) or snowmelt (s) is the primary cause of the annual maximum**
**discharges for the associated catchments, and the difference between the two proportions is less than 70%.**

It is worth noting again that the presence of a mixture of flooding mechanisms in a catchment only indicates that annual
maximum discharges in the catchment are not uniformly caused by the same mechanism, rather than signifying whether individual annual maximum peak discharge events are driven by multiple processes (i.e., compound events). Despite this, floods in catchments with a mixture of flooding mechanisms, in general, are more likely to be affected by two or more flooding processes, since the classification of floods in these catchments can be ambiguous (e.g., the events near the decision boundaries between clusters). For example, floods caused by both heavy precipitation and excessive soil moisture tend to present high
reliance on both recent precipitation and antecedent precipitation, which results in the catchment presenting a mixture of flooding mechanisms, depending on which feature importance is superior. Using 0.10 as a distance threshold to define events near the cluster decision boundaries (i.e., the difference between the distance from one point to its closest centroids and to its second-closest centroids is less than 0.10), 78.9% of such events were found in catchments dominated by a mixture of mechanisms, whereas only 21.1% were found in catchments dominated by single mechanisms.

In Figs. 6b–e, we further examine the relevance of dominant mechanisms to catchment physiographic and hydroclimatic characteristics demonstrated in Fig. 1. Unsurprisingly, snowmelt dominates flooding in regions with high snowfall fractions and obvious characteristics in latitude and altitude, where floods usually occur from May to July. The catchments dominated by antecedent precipitation are within plain terrains, where flooding occurs mainly during the winter and spring. Catchments with a gentle slope generally tend to have thicker soil, slower transmission, and therefore more potential to store antecedent precipitation (Hallema et al., 2016). In contrast, recent precipitation-dominated catchments have a broader spectrum of slopes and elevations and experience also summer floods. The distribution of catchment attributes from catchments dominated by a mixture of mechanisms is consistent with what we found based on catchments dominated by a single mechanism. For example, catchments dominated by snowmelt mixed with recent precipitation (purple in Fig. 6) or antecedent precipitation (yellow in Fig. 6) have relatively high snowfall fractions, with the former mainly occurring on areas with steep slopes (mainly in the Alps and Scandinavian mountains) and the latter mainly occurring on gentle slopes (such as parts of Finland). The catchments controlled by both recent and antecedent precipitation (light blue in Fig. 6) are located mostly in western Europe, suggesting that floods there were likely to be affected by the interaction between extreme precipitation and antecedent soil moisture, and their respective relative importance has varied between events. In addition, some catchments in the Alps, Germany, and Poland are impacted by all three mechanisms (slate grey in Fig. 6). In summary, these findings indicate that dominant flooding mechanisms differ substantially across catchments and are related to their geographic and climatic characteristics. In addition to elevation, slope, and snow fraction, the study by Stein et al. (2021) on catchments in the United States demonstrated that other catchment characteristics (e.g., aridity, precipitation seasonality, and mean precipitation) also significantly influence flood generating processes. An in-depth investigation of how geographic and climatic characteristics affect flood mechanisms in European catchments can be expected in future studies.

### 3.4 Comparative analysis with other studies

A better understanding of the generating processes of river flooding is crucial for interpreting past flood changes and improving future flood-risk predictions. In recent years, large-scale quantitative investigations of flooding mechanisms specifically for Europe have been undertaken in several studies, with different methodologies and scales applied. For example, by using circular statistics analysis, Berghuijs et al. (2019) examined the relative importance of three flooding mechanisms based on the seasonality of floods and three potential drivers such as the largest daily precipitation, the largest daily soil moisture excess, and the largest daily snowmelt. Bertola et al. (2021) attributed changes in the magnitude of flood quantiles to changes in possible drivers by using regression analysis and determined their contributions to flood changes accordingly. In contrast to these analyses conducted at catchment or coarser levels, Kemter et al. (2020) and Stein et al. (2020) performed event-based classifications to determine flooding mechanisms in respective regions or catchments, both using predefined criteria but with different indicators and thresholds. Table 1 summarizes the main findings in these studies regarding the major flooding mechanisms per geographic subregion of Europe and compares them with those identified in this study.

**Table 1. Comparisons of identified flooding mechanisms in Europe by different methods.**

| | Methods used | Research scales | Catchment sizes (km$^2$) | Northern Europe | Western Europe | Central Europe | Southern Europe | Alpine |
|---|---|---|---|---|---|---|---|---|
| This study | Machine learning | Event-based | 8 – 10,000 | Snowmelt | Antecedent precipitation+ recent precipitation | Antecedent precipitation+ recent precipitation, snowmelt | *Lack of samples* | Recent precipitation, snowmelt |
| Berghuijs et al. (2019) | Seasonality analysis | Catchment-based | ~10 – ~100,000 | Snowmelt | Soil moisture | Soil moisture, snowmelt | Soil moisture | Extreme precipitation, snowmelt |
| Bertola et al. (2021) | Changes attribution | 200 km × 200 km | 5 – 100,000 | Snowmelt | Extreme precipitation | Extreme precipitation, snowmelt | Soil moisture | Extreme precipitation, snowmelt |
| Kemter et al. (2020) | Multi-criteria | Event-based | 1 – 800,000 | Snowmelt | Soil moisture | Rain-on-snow, soil moisture | Soil moisture | Stratiform rainfall |
| Stein et al. (2020) | Multi-criteria | Event-based | 1 – ~2,000,000[*] | Snowmelt | Excess rainfall | Snow/rain, Excess rainfall | Excess rainfall | Short rainfall |

*Note: The summaries above were compiled from relevant figures or qualitative descriptions in the respective studies, and the subregions of Europe were not strictly defined. The definitions of various flooding mechanisms were not identical between the studies. [*] The catchment size range was not stated in the paper, and we calculated it from the original results provided by the authors.*

As indicated in Table 1, despite the different definitions, methods, and standards in recognizing flooding mechanisms, the five studies present some consistency, especially in Northern Europe and the Alps, which are dominated by snowmelt or by snowmelt combined with extreme precipitation. Among the four previous studies, this study shows the largest consistency with Berghuijs et al. (2019), especially when it comes to the contribution of meteorological drivers to flood generation in individual catchments. However, Berghuijs et al. (2019) and Kemter et al. (2020) regarded floods in regions from northern

France to northern Germany as a consequence of soil moisture excess almost exclusively. In contrast, Bertola et al. (2021) and this study included extreme precipitation also as a crucial factor, and we have demonstrated that floods in those regions are driven by a combination of both heavy precipitation and saturated soil moisture.

In addition to methodological differences, the inconsistent catchment samples are also responsible for the divergent attribution

results in different studies. As shown in Table 1, the catchments examined in this study are generally smaller, which tend to be more susceptible to rainfall with high intensity. Moreover, discrepancies in the estimation of soil moisture might be an additional reason. In the absence of direct observations, soil moisture in the four previous studies was explicitly estimated by using simple water balance models (Berghuijs et al., 2019; Stein et al., 2020), reanalysis data (Kemter et al., 2020), and a proxy based on antecedent precipitation (Bertola et al., 2021). The uncertainty associated with soil moisture estimates may, however,

make a difference in determining whether floods are triggered by extreme precipitation or soil moisture excess. Tarasova et al.

(2020) conducted a rigorous uncertainty analysis of input data for a runoff event classification framework, emphasizing the importance of developing novel indicators to reduce these uncertainties. Here, profiting from the memory property of LSTM models, the present study identified flooding mechanisms based on long-term predictive relationships between precipitation, temperature, day length and discharge. The method has reduced the need for accurate catchment wetness estimates, yet such uncertainty is not eliminated completely, particularly since we chose a 7-day window to separate between antecedent and recent precipitation. Compared to analyses at catchment or coarser levels, event-based investigations of flooding mechanisms have the advantage of allowing for the detection of stronger signals about their potential changes over time, since averaged information tends to obscure information about individual event processes and thus makes the trends imperceptible. For example, Berghuijs et al. (2019) found no discernible change in the relative importance of flood drivers for most regions in Europe, while some regional studies (e.g., Vormoor et al., 2016; Beniston and Stoffel, 2016) and event-based studies (e.g., Kemter et al., 2020) have indicated such changes.

## 3.5 Temporal evolutions of flooding mechanisms

To test whether the dominant mechanism has changed over the period 1950–2020, we first compared the catchment-level dominant mechanisms separately for 1950–1985 and 1985–2020 by applying the procedure implemented in Section 3.3. Only the 818 catchments with at least 15 years of records in each period were considered. Figure 7a summarizes the proportions of the single dominant mechanisms (represented by colorful blocks) and their combinations (represented by grey blocks) during each period along with shifts between them. The Sankey plot indicates that a majority of catchments (79.6%) retain their dominant mechanisms, and there has not been a shift from one dominant mechanism to another (see the absence of data flow between two different blocks from left to right). However, some catchments with single mechanisms have become dominated by a mixture of mechanisms (i.e., flowing from colorful blocks to grey ones, which accounts for 7.2% of the total), while some behave in the opposite way (7.3%). In a few catchments with a mixture of mechanisms (5.9%), the dominant mechanisms have also changed, though they remain mixed.

Despite only a few fractions of catchments presenting a change in their dominant flooding mechanisms, Fig. 7b reveals tendencies for specific mechanisms at event levels when considering all annual maximum peak discharges in the 818 catchments over the past seven decades. In particular, the annual maximum peak discharges driven by snowmelt have been declining by 0.8% per decade. In contrast, recent precipitation has become more dominant in causing floods, increasing by 1.1% per decade, despite weaker significance probably due to the inconsistent changes from 2005 onward. Both frequency changes are probably associated with the warming atmosphere, which causes decreased snowpack (Fontrodona-Bach et al., 2018). Also, because of the rising temperatures, the atmosphere has a higher moisture holding capacity, leading to an increase in precipitation extremes on average (Trenberth, 2011; Fischer and Knutti, 2016). These factors make it more likely that the annual floods are driven by recent precipitation and less frequently by snowmelt. Additionally, we observe an overall slight decrease in soil moisture excess-driven floods as a result of counterbalancing the other two trends, though the trend is not

statistically significant when considering the entire period. The above conclusions hold when considering a smaller subset of catchments (460 in the case) with at least 25 years of records in each period.

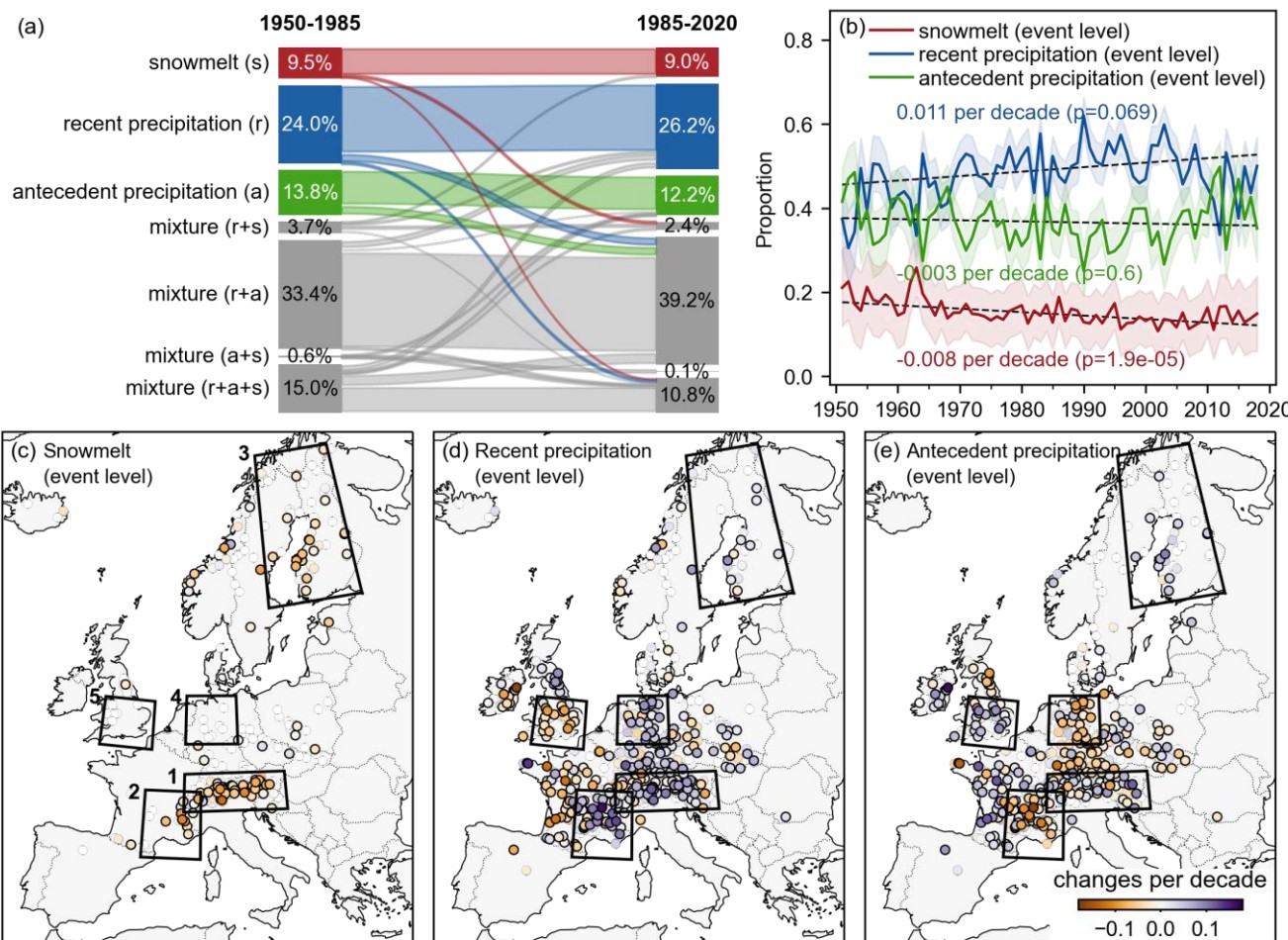

**Figure 7: (a) Sankey plot indicating the proportions of single dominant flood generating mechanisms and their combinations during two time periods, with the flow lines indicating shifts between them. The proportions were calculated based on the 818 catchments that have at least 15 years of records available in each period. (b) The evolution of the proportions of annual maximum peak discharges with the three flooding mechanisms. The shades denote the 95% confidence interval of the proportions, which was calculated as $\hat{p} \pm 1.96 \times \sqrt{\frac{\hat{p}\,(1-\hat{p})}{n}}$ ($\hat{p}$ is the estimated proportion and $n$ is the sample size). The dashed black lines indicate the slope of their trends estimated by Theil-Sen's Estimator, with their significance being assessed by the modified Mann-Kendall test. (c), (d), and (e) The spatial trends in different event-based flooding mechanisms, where the trends indicated by the colorful dots were calculated using a 20-year moving window. Markers with black edges denote catchments with significant trends ($\alpha$=0.05). The black boxes highlighted five hotspot regions that are discussed in the main text.**

Note that Fig. 7b only presents the overall trends in flooding mechanisms at the continental scale, while disparate trends may exist in different regions that could cancel each other out. Therefore, we further examined the trends in different event-based

mechanisms in the 818 catchments (Figs. 7c–e), with the color representing the Theil-Sen slopes computed on the time series of respective proportions in individual catchments. The results indicate that most catchments in the Alps, which are typically dominated by snowmelt, have experienced significant decreases in snowmelt-driven floods, while similar cases have occurred in Scandinavia as well (Fig. 7c). In contrast, extreme precipitation has become a more frequent cause of annual maximum discharges in the Massif Central, North European Plain, and the Alps, while decreased trends are observed in some regions of

Western Europe and especially southeast England (Fig. 7d). As for soil moisture-induced floods, their proportion generally shows opposing trends relative to those of extreme precipitation (Fig. 7e).

The decreasing trend in snowmelt-driven floods was also detected by Kemter et al. (2020), with 1.65% per decade, mainly occurring in eastern Europe, which was outside of our study area. In addition, they detected an increase in stratiform rainfall-

driven floods (0.49% per decade) mainly along the Mediterranean coast and an increase in soil moisture excess-driven floods (1.55% per decade) in the British Isles and central and northern Europe. The difference between Kemter et al. (2020) and this study probably arises from the varying study areas (the former additionally includes a large number of eastern and southern European catchments), as well as the definition of flood types. For example, their study defined soil moisture excess-driven floods as non-snowmelt floods when the mean soil water content was above 70% before a time window, and the remainder

were stratiform rainfall-driven floods. In contrast, this study used cluster analysis for the actual contributions of precipitation events before floods, and soil moisture-induced floods were related to annual maximum peak discharges where the contribution from antecedent precipitation is more important than recent precipitation.

### 3.6 Possible causes and implications of the trends

To gain insights into the causes of the identified trends, we analyze five selected regions highlighted in Figs. 7c–e (see region

numbers in panel c), which feature consistent trends in certain mechanisms. For region 1 (the Alps) and region 3 (northeast Scandinavia), catchments with significant decreasing trends in snowmelt-driven events were considered. For region 2 (southeast France) and region 4 (northern Germany), we considered catchments with significant increasing trends in extreme precipitation-driven events, as well as those presenting significant decreases for region 5 (southeast England). Figure 8 shows the temporal regional evolution of the event-level mechanisms within the considered catchments, along with the change in

magnitude of the annual maximum 7-day precipitation and mean spring temperatures over the past 70 years. For the two regions with significant soil moisture effect on flooding (i.e., regions 4 and 5), we additionally added the averaged trends of antecedent soil moisture conditions prior to flooding for analysis.

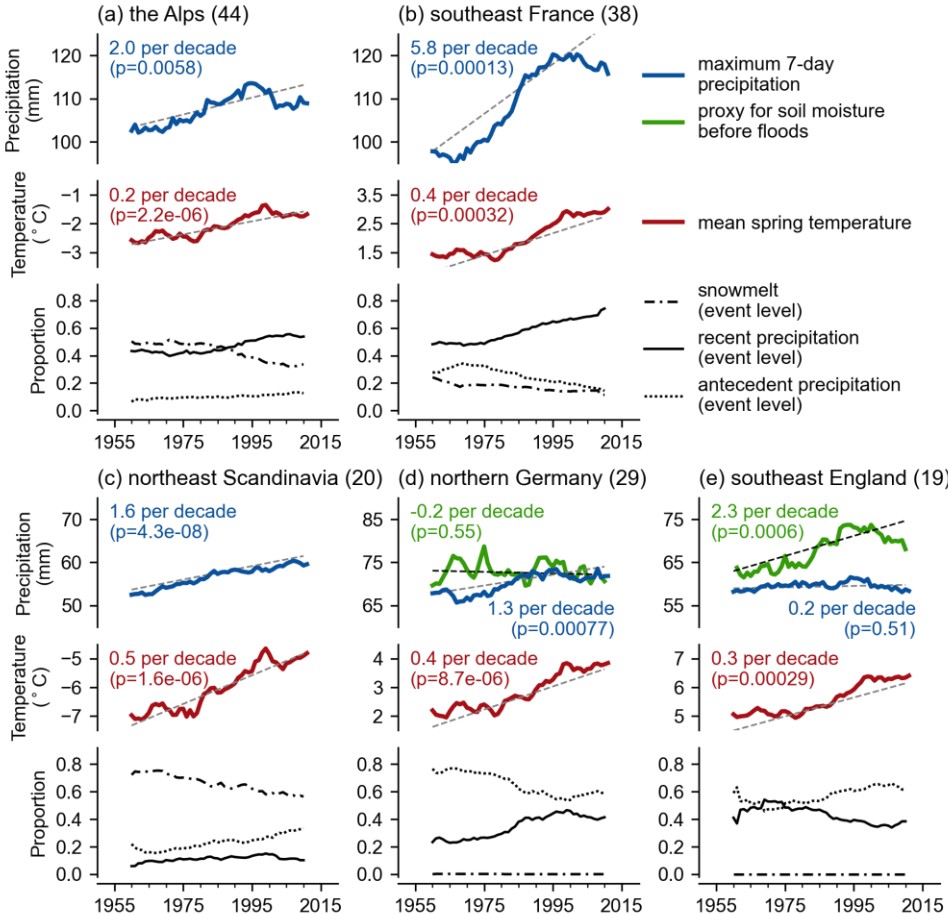

**Figure 8: The temporal changes of the event-level mechanisms in relevant catchments within the five selected regions (see Fig. 7c), as well as the changes in average extreme precipitation (represented by annual maximum 7-day total precipitation), mean spring temperatures (represented by average temperature between January and April), and antecedent soil moisture conditions prior to flooding (represented by the 30-day total precipitation preceding the 7-day window of recent precipitation). The numbers in panel titles indicate the number of catchments considered. The proportions were calculated by a 20-year moving window, while precipitation and temperature were smoothed by using a 20-year moving average window, with their values at central positions in time windows. The dashed grey lines indicate the slope of relevant trends with their significance.**

Mean spring temperatures have increased significantly in all five regions (Fig. 8), confirming the previous explanations for the reduced influence of snowmelt on river discharge annual maxima in snowy areas (regions 1-3) (Vormoor et al., 2016; Beniston and Stoffel, 2016). Furthermore, in regions 1–4, the increased magnitude of maximum 7-day precipitation can explain the rise in proportions of annual maximum peak discharges driven by extreme precipitation events. In contrast, the maximum 7-day precipitation in southeast England (region 5) remained almost unchanged (Fig. 8e). Nonetheless, soil moisture conditions before discharges might have increased in southeast England, as indicated by the increasing antecedent precipitation accumulations, which causes annual maximum discharge there to be more likely driven by soil moisture excesses than by

recent precipitation. Blöschl et al. (2017) stated that the region has a large subsurface water storage capacity, which is capable of storing a large amount of water that continuously increases until flooding occurs. In comparison, in northern Germany (region 4), the antecedent precipitation before annual maximum peak discharges changed more slightly (Fig. 8d), while the increase in precipitation extremes likely caused an increase in floods driven by recent heavy precipitation. Note that here we merely examined the monotonic trends within data over the 70 years, while the trends may vary piecewise (e.g., the changes in maximum weekly precipitation in the Alps and southeast France), the impact of which on flooding mechanisms deserves further research. These figures are robust against spatial variability within regions (see Fig. S6 in the Supplementary Material).

A change in flooding mechanisms may affect the seasonality and magnitude of flooding, which might ultimately impair the current flood risk management measures. For example, in catchments previously dominated by snowmelt, increasing floods from extreme precipitation and soil moisture excess may lead to shifted flood mean dates and less concentrated seasonal patterns (as exemplified in Fig. B1 in Appendix B). By simulating daily discharge for a reference period (1961–1990) and a future period (2071–2099), Vormoor et al. (2015) predicted that floods in some Nordic catchments could even shift from spring to autumn as rain replaced snowmelt as the dominant flood-inducing process. These results suggest that, in a warmer climate, flood risk predictions in snowmelt-affected catchments should consider the interconnection between changes in flooding drivers and seasonality.

As for the impact on flooding magnitude, while it is challenging to link observed changes in individual flooding drivers alone to changes in flooding magnitudes, a link may appear especially in light of climate change (Blöschl et al., 2019). For example, the catchments where floods are dominated by recent precipitation tend to be more susceptible to changes in extreme 7-day precipitation (Fig. B2 in Appendix B). Despite a lack of sufficient observational evidence that the magnitude of floods increases with more extreme precipitation (Sharma et al., 2018), the trend of which is often determined jointly by both changes in rainfall and changes in antecedent soil moisture, some studies demonstrated the changed precipitation severity could vary the relationship between precipitation and streamflow (Bennett et al., 2018). When recent rainfall increases, changes in antecedent moisture conditions would become less important in modulating the response to rainfall (Wasko and Nathan, 2019). Brunner et al. (2021) indicated that it is possible to identify a catchment-specific extremeness threshold, above which precipitation increases clearly produce greater flood magnitudes, and below which flood magnitude is strongly modulated by soil moisture. Therefore, the persistent risk that recent extreme precipitation would have an increasingly decisive role in flood generation for a large proportion of catchments, as implied by Fig. 7, cannot be disregarded. Recognizing the impact of such mechanism shifts in flooding mechanisms is crucial for understanding the link between changes in precipitation and flood risk in a warming climate.

## 3.7 Limitations and outlooks

In this study, we trained LSTM models in a local fashion (i.e., training the model individually for each catchment), rather than a regional fashion (training a single model across multiple catchments), since the main objective of the study is to identify distinguishable patterns of meteorological variables' contributions at local scales. From a prediction standpoint, particularly for unprecedented events and ungauged basins (Nearing et al., 2021; Frame et al., 2022), regional modeling may be a better choice because it is capable of learning more general relationships from a larger variety of hydrological data (Kratzert et al., 2019b). However, for the regional modeling, both meteorological time series and static catchment attributes are used as inputs to distinguish response behaviors across time and space. Adding such static attributes would introduce substantial multicollinearities among the considered variables (see Fig. S7 in the Supplementary Material for illustration). Multicollinearity might not be a problem for ML models when they are used for prediction, as long as the collinearity between variables remains stationary (Dormann et al., 2013). Nevertheless, for our study that aims to interpret the effects of predictors on responses, high multicollinearity in predictors indicates considerable information may be shared among the collinear sets. This would result in difficulties in separating the physical effects of these variables – this is also the case in traditional regression models (Hartono et al., 2020). Therefore, interpreting flooding mechanisms with regional LSTM models may become more challenging than with local LSTM models that use only meteorological time series, since some catchment attributes would confound the interpretation. In this study, we therefore employed simple local models, which avoids confounding and multicollinearity resulting from static catchment attributes. However, in light of the benefit of regional modeling that can provide insights into how flooding mechanisms vary spatially by geographic and climatic characteristics of catchments, how to deal with these challenges in the interpretation merits more exploration in future studies. An immediate question to address is whether adopting different modeling strategies will result in different interpretations regarding the gradient contributions of meteorological forcings, which ultimately leads to alternative understandings of flooding mechanisms. The emerging differences may provide us with an opportunity to gain new insights into flooding mechanisms from these models.

The multicollinearity also exists in meteorological drivers at daily scales, which requires careful handling of the interpretation results if adding more predictors. For example, radiation is usually an important driver of snowmelt that favors flooding (Merz and Blöschl, 2003), but the interpretation method might not assign it high importance when it is combined with day length as an additional predictor due to the high correlation between the two variables (see Fig. S8 in the Supplementary Material for an example). This is because the used interpretation technique does not measure how important a feature is in the real world, but how important it is to the model. Therefore, it is not necessarily better to add more input features to a model in terms of process understanding, which can be even misleading if the interpretation results are not justified by sufficient physical knowledge (Kroll and Song, 2013). In this study, instead of using more predictors that result in less interpretability, we restricted ourselves to few input features whose effect can be relatively easily interpreted and understood. Therefore, we only

selected daily precipitation, temperature, and day length as meteorological inputs, the combination of which results in uncovering three well-known flooding mechanisms. The results are physically interpretable and comparable with findings from other studies that used classical methods. Incorporating more meteorological drivers into the model might, in theory, allow for the identification of additional flooding mechanisms that may be overlooked. However, multicollinearity and confounding can pose a challenge to interpretability, especially when the recognized patterns cannot be linked to fundamental physical processes. Therefore, we leave how to resolve the trade-off as an open question for future studies.

In the clustering procedure, we chose to use a 7-day window to aggregate the daily IG scores into a low-dimensional contribution vector for the sake of efficiency in clustering lengthy time series, which could induce inevitable uncertainties and subjectivity. Despite this, additional tests indicate that our findings are similar when using a 5-day window, which is also a common interval to consider flooding drivers (e.g., Rottler et al., 2021). Specifically, based on the 5-day window, the events identified with snowmelt, recent precipitation, or antecedent precipitation as the primary causes account for 15.3%, 48.9%, and 35.8% of all the 53,968 annual maximum peak discharges (Fig. S9 in the Supplementary Material), which is only slightly different from using a 7-day window. As for the three mechanisms in individual catchments, decreasing the window length has the least impact on identifying snowmelt-driven floods, with the absolute changes in their proportions within 1% for 81.5% of catchments and within 5% for 97.2% of catchments. In comparison, the proportion changes for two other flooding types are more sensitive, with changes within 5% for 76.6% (78.0%) of catchments in terms of recent (antecedent) precipitation-driven flooding. However, this does not affect the conclusion regarding the respective trends in flooding mechanisms (Fig. S10 in the Supplementary Material), indicating the robustness of the methodology. Despite this sensitivity analysis, we would like to emphasize that the selection of the separating window remains somewhat subjective, and further exploration is needed to avoid a possible bias due to arbitrary judgments in identifying flooding mechanisms.

## 4 Conclusions

Flooding in rivers is usually caused by complex interactions between heavy precipitation, high soil moisture, and melting snow. Climate change has resulted in an overall decreased snowpack and more intense short-term precipitation extremes, which might systematically alter the interaction between flood drivers at the catchment level. To investigate whether flooding mechanisms have changed in European catchments, this study introduced a novel explainable ML method to identify flooding mechanisms. Compared with conventional classification approaches, where the results are usually dependent on appropriate flood process definitions and sensitive to the selected indicators and threshold parameters, the combination of explainable ML and cluster analysis is able to avoid such predefinitions and reduces subjectivities in identification processes. With the ML-captured feature importance of precipitation, temperature, and day length for predicting annual maximum discharges, we aggregated driver contributions in the recent 7 days and an earlier period (back to 180 days) and then applied cluster analysis to group them based on similar patterns. As a result, the method identifies three major patterns that induce floods across 977

European catchments, corresponding to three typical flooding mechanisms, including recent precipitation (responsible for 49.9% of the annual maximum discharge events), antecedent precipitation (i.e., excessive soil moisture, accounting for 34.6%), and snowmelt (15.5%). The results indicate that for 26.9% of catchments, recent precipitation is the typical main contributor to floods, while floods are typically controlled by antecedent precipitation (linked to excessive soil moisture) in 10.9% of catchments. In around one-third (33.8%) of catchments, floods are dominated by a combination of recent heavy precipitation and antecedent precipitation events, meaning that some floods there were caused by recent rains, and others were primarily driven by antecedent precipitation, although many of them were likely due to the compound effect between the two drivers. The remaining catchments are dominated by snowmelt (10.1%), or by combinations of snowmelt with the other two drivers. The spatial distribution of the dominant flooding mechanisms reflects the variation of the catchment's geographic and climatic characteristics and is generally consistent with results reported in earlier studies, some of which did not perform event-based classifications but rather identified the overall mechanisms within individual catchments.

We further detected changes in dominant flooding mechanisms over the last 70 years in over 20.4% of European catchments, especially some catchments that were previously dominated by single mechanisms became dominated by a mixture of mechanisms and some catchments show opposite shifts. Despite no regime shift from one single flooding mechanism to another single one, tendencies in their mechanisms at event levels were found. Specifically, when taking all annual maximum discharge events into account, those triggered by snowmelt have significantly decreased, with their proportion dropping by 0.8% per decade. Recent 7-day precipitation, on the other hand, has become increasingly important for flooding, with flooding triggered by such recent heavy precipitation increasing by 1.1% per decade. The changes in flooding mechanisms present a largely consistent pattern with climate change responses, and we discuss the potential risks associated with the resulting effects on flooding seasonality and magnitude.

Overall, this study highlights the usability of explainable ML in helping uncover complex and possibly non-linear changes in weather and climate extreme events in the warming Earth system. With more large-sample hydrometeorological datasets becoming readily accessible, one next step is to extend the research to a larger scale for a better understanding of variations in flooding mechanisms globally. Still, many challenges remain for future work, providing potential research opportunities. For example, the clustering procedure can be improved by developing algorithms to aggregate daily feature importance adaptively, thereby avoiding the predefined separation window while maintaining high efficiency. Moreover, regional LSTM models that incorporate static catchment attributes can be employed to capture the spatial variations in flooding mechanisms and quantify the influence of catchments' geographical and climatic conditions on flooding processes. In addition to the integrated gradient method used in this study, other interpretation techniques might be explored further to uncover potentially valuable information when more input variables are included.

## Appendix A

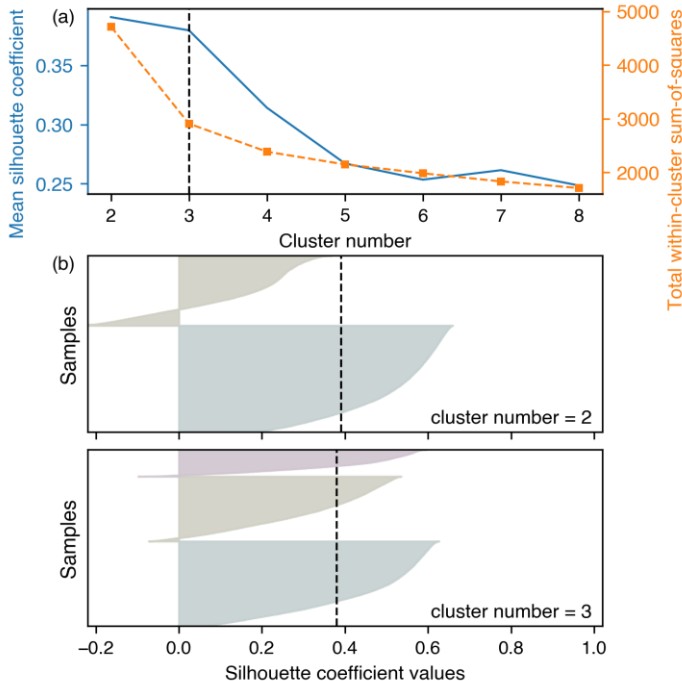

**Figure A1: Determination of optimal cluster number. (a) The average silhouette coefficients and total within-cluster sum-of-squares**
**assessed for respective candidate cluster numbers. (b) The silhouette plots for various clusters when the cluster number being 2 or**
**3, where the x-axis represents the silhouette coefficient for individual samples, and they were ordered by the coefficients and grouped**
**by clusters in the y-axis. (a) suggests that clustering the samples into either two or three groups can achieve the similarly highest**
**average silhouette coefficients, while the silhouette plots for individual samples under the two candidate numbers in (b) further**
**suggest that clustering into three groups would be the best choice because a cluster with all below-average silhouette coefficients is**
**present when clustering into two groups. Therefore, we cluster annual maximum peak discharges into three main groups in the**
**main text.**

## Appendix B

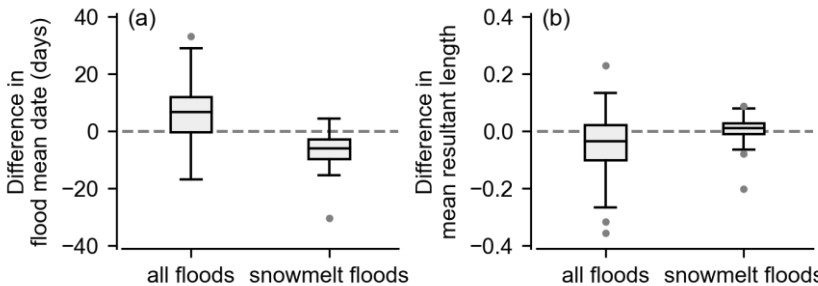

**Figure B1: (a) Change in flooding mean dates (difference from 1985–2020 to 1950–1985) in 44 catchments with a significant**
**reduction of snowmelt-driven floods in the Alps (region 1 in Fig. 7c) for snowmelt-driven floods and all floods irrespective of their**
**cause. For these catchments, the overall proportion of annual maximum discharges caused by snowmelt has decreased from 49.0%**

in 1950–1985 to 36.8% in 1985–2020. (b) The differences in mean resultant length of flood dates for the same cases as in (a). The mean resultant length is a measure in circular statistics between 0 and 1 that reflects the spread of a circular variable, with 0 representing the spread of flood dates evenly distributed over the year and 1 representing the spread concentrated at one day. It can be deducted from (a) that following the temperature increase, snowmelt-driven floods generally occur earlier in the year during 1985–2020 compared to 1950–1985, with a median shift of -5.9 days. On the other hand, annual peak discharges occur later in more than half of the catchments due to the increasing presence of other types of floods. Furthermore, (b) shows that the seasonality of annual maximum discharges has become more diffuse (decreasing mean resultant length) in most catchments for the same reason, though snowmelt-driven floods remain relatively stable.

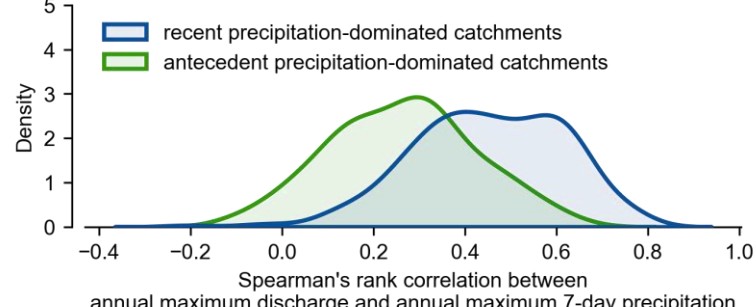

Figure B2: The distribution of Spearman's correlations between annual maximum discharge and annual maximum 7-day precipitation for two groups of catchments (blue, recent precipitation-dominated catchments; green, antecedent precipitation-dominated catchments, based on Fig. 6a). It shows that the catchments where floods are dominated by recent precipitation tend to have higher correlations than antecedent precipitation-dominated catchments, which implies that the former might be more susceptible to changes in extreme 7-day precipitation.

**Data and code availability**

The river discharge data can be obtained from the GRDC dataset (https://www.bafg.de/GRDC). The E-OBS gridded precipitation and temperature dataset is available at https://www.ecad.eu/download/ensembles/download.php(Haylock et al., 2008). Catchment attributes and boundaries are available at https://doi.pangaea.de/10.1594/PANGAEA.887477 (Do et al., 2018) and https://www.bafg.de/GRDC/EN/02_srvcs/22_gslrs/222_WSB/watershedBoundaries.html (Lehner, 2012). The 30 arc-second elevation data shown in Fig. 1a is accessible at http://doi.org/10.5066/F7DF6PQS. The code for the explainable machine learning framework is available at https://doi.org/10.5281/zenodo.4686106.

**Author contribution**

SJ and JZ conceived the study. SJ performed all analyses and wrote the initial draft. All authors substantially contributed to the final draft.

**Competing interests**

The authors declare that they have no conflict of interest.

**Acknowledgements**

The authors acknowledge the European COST Action DAMOCLES (CA17109) and the Helmholtz Initiative and Networking Fund (Young Investigator Group COMPOUNDX; grant agreement no. VH-NG-1537). This project has received funding from the European Union's Horizon 2020 research and innovation programme under grant agreement No 101003469 (XAIDA). We would like to acknowledge the use of the computing resources provided by Dr. Yi Zheng and thank Peter Miersch for proofreading the manuscript.

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
