# Peer review of "River flooding mechanisms and their changes in Europe revealed by explainable machine learning"

_Hydrology and Earth System Sciences, 2022_

## Referee Comment (RC2)

**Review on "River flooding mechanisms and their changes in Europe revealed by explainable machine learning" by Shijie Jian et al.**

Review by Frederik Kratzert

**Summary**

This paper presents a large-sample study to detect different flooding mechanisms across Europe. I have to admit that initially, I was skeptical about this study. But while reading the manuscript for this review, I became quite excited about the presented work.

To my knowledge, it is the first time that such an analysis (detecting flooding mechanisms and analyzing the change over time) is made using a) deep learning models (here LSTMs) and b) methods from the field of explainable AI (here integrated gradients). My views on LSTMs is no secret and I have often said that you can do more than "just fitting streamflow records" with these models, so naturally, I am quite excited to see someone coming up with such an idea.

Additionally, I think this paper is exceptionally well written and at least for me personally, everything seemed pretty clear and reasonable. For example, the authors make a couple of assumptions (like grouping the integrated gradient signal into two groups of a) the last 7 days and b) all other days before that), but their reasoning for all these assumptions is clearly articulated and to me, they make sense.

In general, I think this is a very interesting study that fits into the scope of HESS and I only have a few general comments. Note, I already spoke to the first author during the EGU GA but for the sake of transparency, I will add all points here again. Please also note that, due to the overlap of this research with our own research in the past, in many of the studies that I reference I'm either the first or a co-author. I do not mention these studies here, because I want them to be cited in the manuscript, but I think they help to explain my reasoning.

1. **Training setup**
   You train LSTM models individually for each basin, instead of one model on the combined data of all basins. Again, I'm very biased on this topic but I think there are multiple studies that show that the recommended way for training LSTMs is the latter (on all basins at once, using meteorological timeseries features and static attributes). The regional modeling setup was introduced in 2019 (see Kratzert et al. 2019) and further discussed in Nearing et al. (2021) (see Fig 2). One study that follows the regional training scheme is even cited in the manuscript (Lees et al. 2021).
   The question is, is this important in the context of this study? This is a good question that I asked myself quite a lot over the last few days. On one side, I think it is important to

follow best practices when working with any model. The benefit of the LSTM is that it can learn a very general understanding of the underlying processes if it is trained on a variety of basins. Nobody would probably train a conceptual model in a regional calibration scheme, if she/he is only interested in a particular basin. On the other hand, the authors are not interested in getting the best-possible streamflow performance, but to learn about flooding-mechanisms from the model.

From my experience, I would assume that changing the training setup would not change the results of this study (i.e. the clusters of different flooding mechanisms found here). What might change is the number of basins that are considered in their study (because of the NSE threshold). However, even (or especially?) if the results of this study do not change, I would suggest training an LSTM on the combined data of all basins and to re-run the analysis, to reflect the best-practices of the chosen model. During the EGU, I offered Shijie Jiang help with setting up such a run as I have the code + resources available. I would be happy to help and don't want/expect any co-authorship/acknowledgements for that.

2. **Input variables**

In this study, you only use precipitation, temperature and day-length (as a proxy for solar radiation) as model inputs. I agree that these are the main-drivers for flooding mechanisms but I think the exciting thing about data-driven models is that if there is anything else, these models are pretty good at finding it. That is, if provided with more input features, the models would find other flooding mechanisms, if they are deducible from the data. When I discussed this point with Shijie Jiang at the EGU, he mentioned that he has/had a hard time to interpret the contribution signal for different features, and I agree, it is much simpler and more intuitive to reason about the meaning of high feature importance of e.g. temperature in the recent days. However, how exciting would it be to find that the model finds a flooding mechanism that is not straightforwardly explainable with the patterns we already know? If you have easy access to more input features, I think it could be interesting to run the models with more input features. If not, I think it is ok not to do it, but then it could be worth adding this to the discussion.

3. **Data split**

In L 138f, you mention that some hyper parameters were determined considering the model performance and efficiency. To me it is unclear which data was used for this hyperparameter tuning. Usually, the validation split (note, here I am referring to a 3-fold split with a train, validation and test set) is used for these kinds of hyperparameter tunings. However, from the explanation in L146, I can only estimate that this was done with a 2-fold split, thus on the test data?

4. **Data split pt. 2**
In L146, you say that the timeseries data was randomly split in a "7-to-3 proportion". I am not 100% sure but I think randomly splitting timeseries data is not optimal, especially

when considering the overlap of different samples because of their input window size. I think much more common is a k-fold cross validation in time (none random).

5. **Data split pt. 3**
   I am curious, if you train/eval the models in a 7-to-3 split random fashion, how could you a) guarantee an equal number of model predictions per timestep and b) guarantee that every time step was e.g. evaluated at least once and c) guarantee that the flood peak was in the validation and not the training period? Because from L 169, it seems like you used all 10 models for all peaks, but some peaks are certainly in the training period, right?

References

Kratzert, F., Klotz, D., Shalev, G., Klambauer, G., Hochreiter, S., and Nearing, G.: Towards learning universal, regional, and local hydrological behaviors via machine learning applied to large-sample datasets, Hydrol. Earth Syst. Sci., 23, 5089–5110, https://doi.org/10.5194/hess-23-5089-2019, 2019.

Nearing, G. S., Kratzert, F., Sampson, A. K., Pelissier, C. S., Klotz, D., Frame, J. M., et al. (2021). What role does hydrological science play in the age of machine learning?. Water Resources Research, 57, e2020WR028091. https://doi.org/10.1029/2020WR028091

---

## Author Comment (AC1)

This manuscript proposes a new method for classification of flood generation mechanisms using machine learning that provides the information on the importance of different indicators on the generation of the particular flood event. The method has a potential to overcome the subjective choice of classification thresholds of the previously developed methods. It was tested across European catchments with particular focus on how flood mechanisms were changing in the past decades.

The proposed method has certainly some clear advantages compared to the previous methods and provides a new perspective on classification of flood events. It a very well-written manuscript and I have enjoyed reading it very much. I am certain that this is a substantial contribution to the current knowledge on flood generation processes and their changes. However, although the proposed method has the potential to avoid subjective thresholds, I do not think that the authors have succeeded in overcoming this issue completely. A more prominent attention has to be paid to this issue in the manuscript. I also have some minor suggestions that might help to improve the manuscript and clarify its novelty. Please see my detailed comments below.

Kind regards,

Larisa Tarasova

**Response:**

We thank the reviewer for the thoughtful comments and suggestions which we believe will greatly improve the manuscript. In the revision, the limitation regarding subjectivity has been emphasized appropriately. The replies to the comments follow.

General comments

Abstract: I think the abstract puts too much focus on the changes in flood mechanisms in Europe that is not the most novel findings and instead fails to elaborate on the machine learning approach used here and its advantages compared to previously existing methods. The method implemented in this study is the real novelty, while there are already several studies on changes in flood generation processes in Europe. Therefore, I suggest the authors to consider to put more stress on the methodological aspects in abstract to show how this study stands out.

**Response:**

Thank you for pointing this out. The explainable machine learning methodology was originally proposed in our previous paper (https://doi.org/10.1029/2021WR030185), and the present manuscript aimed to apply the developed framework to tackle practical problems (i.e., the changes in flood mechanisms in Europe). Therefore, we focused more on the scientific aspect instead of the methodology itself. However, your suggestions are appreciated, and we will make appropriate modifications to highlight more on the methodology by adding "*Recent years have witnessed the increasing prevalence of machine learning in hydrological modeling and its predictive power has been demonstrated in numerous studies. Machine learning makes hydrological predictions by recognizing generalizable relationships between variables, which, if explained properly, may provide us with further scientific insights into hydrological processes*". Moreover, we will modify the last sentence in the abstract as "*Overall, the study provides a new perspective on understanding changes in weather and climate extreme events by using explainable*

*machine learning and demonstrates the prospect of artificial intelligence-assisted scientific discovery in the future.*"

Selection of thresholds: The proposed methodology has a very strong advantage that it can avoid arbitrary decisions on how the indicators and their threshold are selected for the event classification. However, the authors did not avoid that issue as they have selected the periods for which the effect of recent and antecedent precipitation was accumulated to avoid additional computational effort. This pragmatic choice is understandable and is in line with the subjective choices previous classification studies were making, but it has to be properly stated in the manuscript and a sensitivity analysis on the effect of this choice on the results of the study will be very welcome. Please also see my detailed comments to the corresponding part of the manuscript.

**Response:**

Thank you for pointing this out. We agree that choosing a 7-day window to separate between antecedent and recent precipitation will introduce subjectivities and uncertainties, as we admitted in the original manuscript "The method has reduced the need for accurate catchment wetness estimates, yet such uncertainty is not eliminated completely, particularly since we chose a 7-day window to separate between antecedent and recent precipitation".

Your suggestion about analyzing the sensitivity of the selection of the separating window is appreciated. In the revision, we supplemented an analysis that uses a 5-day window to separate recent contributions and antecedent contributions, and we found that the main conclusion is not affected by this change. We added the analysis into the new section "3.6 Limitations and outlooks", as:

"*In the clustering procedure, we used a 7-day window to aggregate the daily IG scores into a low-dimensional contribution vector for the sake of efficiency in clustering lengthy time series, which could induce evitable uncertainties and subjectivity. However, additional tests indicate that our findings will not be compromised if we separate contributions of a variable in recent days and an earlier antecedent period by using a 5-day window, which is also a common interval to consider flooding drivers (e.g., Rottler et al., 2021). Based on the 5-day window, the events identified with snowmelt, recent precipitation, or antecedent precipitation as the primary causes account for 15.0%, 47.9%, and 37.1% of all the 55,828 annual maximum peak discharges, which is only slightly different from using a 7-day window. As for the three mechanisms in individual catchments, decreasing the window length has the least impact on identifying snowmelt-driven floods, with the absolute changes in their proportions within 1% for 84.5% of catchments and within 5% for 98.7% of catchments. In comparison, the proportion changes for two other flooding types are more sensitive, with changes within 5% for 83.2% (82.7%) of catchments in terms of recent (antecedent) precipitation-driven flooding. However, this does not affect the conclusion regarding the respective trends in flooding mechanisms (see Fig. S4 in the supplementary material), indicating the robustness of the methodology. Despite this sensitivity analysis, we would like to emphasize that the selection of the separating window remains somewhat subjective, and further exploration is needed to avoid a possible bias due to arbitrary judgments in identifying flooding mechanisms.*"

[Figure]

*Figure S4: The same case as in Fig. 7 in the main text, but a 5-day window was used to separate contributions of a variable in recent days and an earlier antecedent period.*

Detailed comments

Line 40-42 I suggest to also mention here the study of Kemter et al (2020) on changes of flood mechanisms in Europe and global analysis of Stein et al (2020)

**Response:**

Thank you for the suggestion. We will add "*Using a multicriteria approach, Kemter et al. (2020) identified the flooding mechanisms in Europe by classifying approximately 174,000 flood peaks and revealed their trends over the past 50 years. Likewise, Stein et al. (2020) analyzed flood events over 4,155 catchments worldwide and classified each event into one of 5 hydro-climatological flood generating processes.*"

Line 48-49: Here I miss mentioning the study of Kemter et al (2020) that did exactly that.

**Response:**

Please refer to our response to the previous comment.

Line 88: Please indicate if the size of catchments was limited to avoid the effect of human influence or was there any other reason for this selection?

**Response:**

Thank you for the suggestion. We will add the reason: "*The catchment areas range between 8 km$^2$ and 10,000 km$^2$, with overly large catchments being excluded where the effect of spatial heterogeneity of flood drivers tends to be substantial.*"

Line 100: Please indicate also the lower boundary of catchment sizes to clarify if the size study catchments comparable with the spatial resolution of the hydrometeorological datasets.

**Response:**

Thank you for pointing this out. In addition to indicating the lower boundary of catchment sizes (see the response to the previous comment), we will add a clarification "*Note that smaller catchments under 100 km$^2$ (approximately 0.1° × 0.1°) may encounter unexpected uncertainties due to the relatively coarser spatial resolution of the meteorological data. Nonetheless, those catchments with large uncertainties will not be considered for the subsequent attribution analysis if ML models cannot capture the relationship between inputs and outputs accurately.*"

Line 103: Please elaborate how the catchment boundaries from two datasets were merged. Are they identical?

**Response:**

We will add the clarification "*The catchment boundaries were obtained from readily available GRDC (Lehner, 2012) and GSIM (Do et al., 2018) databases, with GRDC being prioritized when the boundary of a catchment was available in both databases.*"

Line 104-106: I miss here a more motivated choice for the day duration as an indicator for classification. It seems to me that essentially it is a combination of the location and day of the year information. Please provide more information on the nature of the preliminary test performed, particularly if other potential indicators were examined.

**Response:**

In the revision, after "Day length was included in the study since it was shown to improve model accuracy in a series of preliminary tests", we will add "*..., including the cases where only precipitation and temperature were used or day length was additionally incorporated. Catchments with obvious accuracy improvements are primarily located in northern Europe and the Alps.*" The role of day length is as we explained in the original manuscript: "The role of day length implies that the magnitude of these peak discharges can be partially explained by the seasonality presented by day length, which peaks around the June solstice."

Figure 2: The interpretation arrow is not so clear, why does it return back to the input layer? At this point in the manuscript the meaning of the integrated gradients for the features is not yet explained and looks confusing in this Figure. Please add clarification in the caption. Consider indicating the target maximum annual flood in panel a as a point and not as a window. The panel c is rather confusing as there is only one event is being displayed in the panels a and b and the cluster plot is not set in any particular space (i.e., the axes are not indicated). Consider omitting this panel, I think that idea of clustering is understandable without this example only brings more confusion.

**Response:**

Thank you for the suggestions. We removed panel c in the figure. Replacing the window that shows the target peak sample with a point may be confusing because it contains both the observed (black) and predicted (orange) values. To clarify the used window, we will add "*The window in the time series of discharge highlights the target output (which is a point)...*" into the caption instead. We further simplified the arrows used and rewrote the caption. The figure and caption now read:

[Figure]

*Figure 2: The workflow of using explainable ML methods for attributing flood peaks (annual maxima of river discharge) to their drivers. (a) Diagrammatic representation of the used LSTM models. The window in the time series of discharge highlights the target output (which is a point) and the window in the inputs indicates the input features used to predict the illustrated peak discharge sample. (b) The feature importance of the inputs for predicting the peak discharge shown in (a), which was obtained by using the ML interpretation technique (namely integrated gradient). The vertical dashed lines separate the feature*

*importance into a recent 7-day period and an earlier period to calculate the aggregated feature contributions (see main text).*

Line 139-144: I agree that presenting LSTM model in detail is not necessary here, but I think I more detailed explanation on how the structure of LSTM suited for capturing short-term and long-term interaction will be very helpful here, as it can provide the readers with the insights on why particularly this method is more applicable than classifications that are based on subjectively selected thresholds.

**Response:**

Thank you for the good suggestion. We will add a more detailed explanation as: "*The effectiveness of the LSTM is partially due to the comparability of its formulation to the hydrological behavior of a catchment. Specifically, the backbone of the LSTM network is formed by recurrent cells that can store previous information from input sequences, which is conceptually similar to the way meteorological information (e.g., precipitation) is stored in the form of soil moisture or snow depth (Lees et al., 2022). The physically realistic mapping from inputs to outputs facilitates gaining hydrologically meaningful insights from subsequent model interpretations.*"

Line 145: It is not clear. Does it mean for each catchment? Please clarify.

**Response:**

The original sentence will be revised as "*To improve the robustness of model evaluation and analysis, we fitted 10 independent LSTM models for each of the 1,077 catchments.*"

Line 153: What is the sample in this case? Maximum annual floods? Please clarify.

**Response:**

The original sentence will be revised as "*...which allows for obtaining the time-wise feature importance of the three input variables for each sample of the output (i.e., daily discharges).*"

Line 185-187: I do understand authors' arguments on why they had to make this decision and restrict quantification of the effect to 7 and 180 days only. Although, I find it somewhat disappointing. The authors have stated earlier in the manuscript the main advantage of the proposed ML-based method is that one can avoid selecting subjective indicators and their thresholds. In my opinion selecting here 7 and 180 days is nothing else but exactly that kind of subjective threshold that partially impairs the main advantage of the method. If clustering indeed is very time consuming (which is actually surprising to me as in my experience k-mean clustering is not the most time consuming procedure and computational power is hardly a limitation with cluster resources available) at least a sensitivity analysis has to be performed to analyze how the selection of these thresholds affects the results.

**Response:**

Thank you for pointing this out. Please refer to our response to the general comment.

For the efficiency of clustering time series, K-mean clustering with Euclidean distance metric is indeed not a time-consuming procedure, but we have to point out that time series clustering tends to be much more complex as it usually needs Dynamic Time Warping (DTW) as the distance metric. The DTW has a quadratic complexity with respect to the length of sequences (in our study it is 180) compared with the linear complexity of Euclidean distance. Furthermore, in our preliminary tests, we have tried clustering the whole time series as did in our previous study that first attempted such methodology in the US catchments (https://doi.org/10.1029/2021WR030185). The preliminary results that we obtained are similar to those reported in the present study (three clusters). However, since we anticipate the methodology to be used in a larger-scale analysis in future studies, in this study, we improve the efficiency of the original framework proposed in our previous study by introducing a separating window.

Line 188-189: I cannot agree with this statement. The duration will be strongly affected by catchment size and mechanism. The build-up period of snowmelt floods in larger catchments can take up to several months. I also do not think that the provided reference is up to date. Please revise.

**Response:**

Thank you for pointing this out. We were meant to describe the hydrological response time to precipitation and snowmelt events, instead of the build-up periods. We will rewrite our original statements to justify the selection of 7 days, as "*The separating interval has to cover the period of precipitation and snowmelt leading to each peak discharge, which depends highly on the local characteristics. Following a check of the relationship between catchment area and mean event response time, Stein et al. (2020) suggested a synoptic window of 7 days should be sufficient to guarantee the response time for large catchments. As a result, this study used a 7-day period, similar to the practice in most studies that examined flooding causes (e.g., Blöschl et al., 2017; Berghuijs et al., 2019). However, using a shorter period (e.g., 5 days) will not affect subsequent conclusions about dominant flooding mechanisms and their trends (see discussion in Section 3.6).*"

Line 189-190: It is consistent with previous studies, but they also did not examine if these thresholds are appropriate. Please revise.

**Response:**

Please refer to our response to the previous comment.

Line 192: It is not clear what is the role of multiple-peak discharges here and how they were considered. Please clarify.

**Response:**

Sorry for the confusion. Rather than multiple-peak discharges, we referred to multiple peak-discharges here. To avoid ambiguity, we will remove "multiple" in the revision.

 Please clarify if clustering procedure was performed for all catchments simultaneously or if they were considered individually. If it was performed simultaneously for all catchments, does it mean that if a catchment has very local and specific mechanisms they likely not to be detected by the procedure?

**Response:**

Yes, the clustering procedure was performed for all peak discharges pooled from all catchments simultaneously. To clarify it, the original sentence will be revised as: "*To obtain an overall picture from the individual aggregated feature contributions, we used the K-means method to cluster the results for all annual maximum peak discharges pooled from the 1,009 catchments.*" In the section presenting the cluster results, we further added "*Note that the clustering results reflect only major patterns widespread in data, with certain local and specific mechanisms unlikely to be detected.*"

 Are these maximum annual peak discharges? If yes, please indicate it clearly here and elsewhere.

**Response:**

Thank you for pointing this out. We will specify the peak discharges as "*annual maximum peak discharges*" throughout the manuscript.

Figure 3: Does this figure display NSE only for annual maxima or for the complete streamflow time series? Please clarify.

**Response:**

Thank you for pointing this out. In the figure caption, we will add "*The NSE values were calculated using all samples in respective testing datasets.*"

 I think it is a rather a stretch to call streamflow generation that occurs due to excess of soil storage capacity and heavy precipitation as we cannot guarantee that heavy precipitation generates overland flow. In case it is first contribute to increase of soil moisture storage the physical process of streamflow generation will be the same for both drivers. Please revise.

**Response:**

Thank you for pointing this out. We will remove this statement in the revision.

 I think "mixed mechanisms" is not an accurate term here as it refers to the occurrence of different mechanisms in the same catchment, but not necessarily simultaneously. Consider using "mixture of mechamisms" instead.

**Response:**

Thank you for the suggestion. We will replace "mixed mechanisms" with "*mixture of mechanisms*" throughout the revised manuscript.

Figure 6: Please add an explanation for the mixtures in the caption. Please also clarify how the classes for two processes are formed, do the corresponding two processes have to generate more than 70% of annual maxima?

**Response:**

We defined a catchment dominated by a mixture of two processes as the case where the difference between the proportions of the two processes is less than 70% (say one is 35% and another is 65%), in order to distinguish it from single mechanisms. We will add an explanation in the figure caption, as: "*Mixture means the associated catchments are dominated by two or more flooding mechanisms. For example, mixture (r+s) indicates either recent precipitation (r) or snowmelt (s) is the primary cause of the annual maximum discharges for the associated catchments, and the difference between the two proportions is less than 70%.*"

Line 338-340: I think it will be helpful to relate here to the findings of Stein et al (2021) (doi: 10.1029/2020WR028300) on the controls of catchment characteristics on the dominance of different flood mechanisms

**Response:**

Thank you for the suggestion. We will add "*In addition to elevation, slope, and snow fraction, the study by Stein et al. (2021) on catchments in the United States demonstrated that other catchment characteristics (e.g., aridity, precipitation seasonality, and mean precipitation) also significantly influence flood generating processes. An in-depth investigation of how geographic and climatic characteristics affect flood mechanisms in European catchments is expected in future studies.*"

Line 350: Consider using term "pre-defined" criteria instead of "manual" as it is not so clear.

**Response:**

Replaced as suggested.

Table 1: Consider also adding catchment sizes to the comparison as I expect that there is a difference between these studies also in that regard.

**Response:**

Will be added.

Line 379: I think the study of Kemter et al 2020 also should be mentioned here.

**Response:**

We will add the reference as suggested.

Line 381-383: This note would be more helpful earlier before the comparison of the results. Consider moving this part up.

**Response:**

We will move it up.

Line 385-389: I think it might be worth mentioning here the work of Tarasova et al 2020 (doi: 10.1029/2019WR026951) that investigates how using different data sources for the same indicator affects event classification

**Response:**

Thank you for providing the useful reference. We will add "*A work worth mentioning is Tarasova et al. (2020), which conducted a rigorous uncertainty analysis of input data for a runoff event classification framework, emphasizing the importance of developing novel indicators to reduce these uncertainties.*"

Line 404-407, 427-431: These parts would be more suitable in the dedicated Method section

**Response:**

Thank you for the suggestion. We moved these parts to the new subsection "*2.4 Trend analysis of flooding mechanisms*" in Method.

Figure 7: Please indicate how many catchments are the basis for Sankey plot in the caption. Please also clarify the origins of the p value in the caption. The information provided on methodological aspect of trends in this caption is not sufficient. Please add a corresponding section in the Methods. Panel b: I am wondering if the results of trend analysis are not so clear due to regional differences in the direction of trends. Looking at the results of Kemter et al (2020) it seems that there are disparate trends for different regions that can be obscured when mixed together. Perhaps something worth mentioning in the corresponding text.

**Response:**

Thank you for the suggestions.

In the caption, we will add "*The proportions were calculated based on 846 catchments, where at least 15 years of records were available in each period.*" and "*..., with their significance being assessed by the modified Mann-Kendall test.*" The details will be provided in the new section "2.4 Trend analysis of flooding mechanisms".

For panel b, before introducing the results of trends in individual catchments, we will add "*Note that Fig. 7b only presents the overall trends in flooding mechanisms at the continental scale, while disparate trends may exist in different regions that could cancel each other out.*"

Line 455-459: It would be helpful if this information is provided in the dedicated Method section.

**Response:**

Thank you for the suggestion. We will move these parts to the new subsection "*2.4 Trend analysis of flooding mechanisms*" in Method.

Figure 8: It is not clear why the lines of the plot do not correspond to the whole extent of time axis. Please clarify or correct. In region 1 and region 2 it seems that there is certain periodicity in the data, it would be helpful if the authors would add a short discussion on suitability of monotonic trends analysis in such cases. Please also consider adding geographical indications for regions instead of numbers. This will make this figure easier to interpret. Please also add the number of catchments in each of the considered regions.

**Response:**

For the time axis, because the 20-year moving window is used, the range reduces to from 1950-2020 to 1960-2010. In the caption, we will add, "*The proportions were calculated by a 20-year moving window, while precipitation and temperature were smoothed by using a 20-year moving average window, with their values at central positions in time windows*."

For the possible periodicity in data, we will add "*Note that here we merely examined the monotonic trends within data over the 70 years, while the trends may vary piecewise (e.g., the changes in maximum weekly precipitation in the Alps and southeast France), the impact of which on flooding mechanisms deserves further research.*"

For the geographical indications and number of catchments, we will add them to the figures as suggested.

Line 492, 499. Caption of Figure 9: It is not clear which length is meant here. Please clarify.

**Response:**

In the caption of Figure 9 (now Figure B1 in Appendix B), we explained the length as "*The mean resultant length is a measure in circular statistics between 0 and 1 that reflects the spread of a circular variable, with 0 representing the spread of flood dates evenly distributed over the year and 1 representing the spread concentrated at one day.*"

Line 486-504: This part is not very well connected to the previous narration and provides yet another new results for which methods were not clearly elaborated in the Method section. Consider omitting it or revise.

**Response:**

Thank you for pointing this out. We will move the description of Figure 9 into the Appendix and simplify the part in lines 486-504, and it now reads: "*A change in flooding mechanisms may affect the seasonality and magnitude of flooding, which might ultimately impair the current flood risk management measures. For example, in catchments previously dominated by snowmelt, increasing floods from extreme precipitation and soil moisture excess may lead to shifted flood mean dates and less concentrated seasonal patterns (as exemplified in Fig. B1 in Appendix B). By simulating daily discharge for a*

*reference period (1961–1990) and a future period (2071–2099), Vormoor et al. (2015) predicted that floods in some Nordic catchments could even shift from spring to autumn as rain replaced snowmelt as the dominant flood-inducing process. These results suggest that, in a warmer climate, flood risk predictions in snowmelt-affected catchments should consider the interconnection between changes in flooding drivers and seasonality.*"

Line 539-543: I would recall here how "recent" and "antecedent" precipitation were defined in this study, because despite what this part claims the definition of these two indicators were set arbitrary by selecting corresponding number of days during which the effect was evaluated.

**Response:**

Thank you for the suggestion. In the revision, we will update the relevant sentence to read as follows: "*With the ML-captured feature importance of precipitation, temperature, and day length for predicting annual maximum discharges, we aggregated driver contributions in the recent 7 days and an earlier period (back to 180 days) and then applied cluster analysis to group them based on similar patterns.*"

Line 549: The term "perspective of catchment average" is not very clear here without the context. I think it would be clearer to just indicate that these methods did not perform an event-based classification and instead identified one single dominant driver per catchment.

**Response:**

Thank you for the suggestion, we will modify "...some of which were obtained taking a perspective on catchment averages" as "...*some of which did not perform event-based classifications but rather identified the overall mechanisms within individual catchments.*"

Conclusion section: A statement about the dependence of the results on the performance of the ML model for the proposed classification method would be very welcome in this section. Moreover, same as for abstract more focus on the newly developed ML-based classification method instead of changed in the mechanisms will be welcome here to highlight the novelty of this study.

**Response:**

Thank you for the suggestion. To clearly highlight the novelty, in the first paragraph, we will replace the relevant sentences with "*To investigate whether flooding mechanisms changed in European catchments, this study introduced a novel explainable ML method to identify flooding mechanisms. Compared with conventional classification approaches, where the results are highly dependent on appropriate flood process definitions and sensitive to the selected indicators and threshold parameters, the combination of explainable ML and cluster analysis is able to avoid such predefinitions and reduces subjectivities in identification processes. With the ML-captured feature importance of precipitation, temperature, and day length for predicting annual maximum discharges, we aggregated driver contributions in the recent 7 days and an earlier period (back to 180 days) and then applied cluster analysis to group them based on similar patterns.*"

Moreover, at the end of the conclusion section, we will replace the original outlook with an outlook for improving the methodology, and it now reads "*Overall, this study highlights the usability of explainable ML in helping uncover complex and possibly non-linear changes in weather and climate extreme events in the warming Earth system. With more large-sample hydrometeorological datasets becoming readily accessible, one next step is to extend the research to a larger scale for a better understanding of variations in flooding mechanisms globally. Still, many challenges remain for future work, forming exciting research opportunities. For example, the clustering procedure can be improved by adopting algorithms to aggregate daily feature importance adaptively, which would allow the predefined separating window to be avoided while maintaining high efficiency. Moreover, regional LSTM models with static catchment attributes incorporated can be employed to capture the spatial variations in flooding mechanisms and quantify the influence of catchments' geographical and climatic conditions on flooding processes. In addition to the integrated gradient method used in this study, other interpretation techniques might be explored further to uncover potentially valuable information when more input variables are included.*"

Line 563-565: I think the authors have to be more cautious here with this statement, because there might be strong regional differences (i.e., there are disparate patterns in precipitation changes in Europe). Moreover, the term extreme precipitation is much more often related to very short precipitation (i.e., less than 1 day), while 7-day long precipitation can substantially affect the storage of the catchment and lead to soil moisture excess floods and hence the resultant magnitude of the flood will depend much more on the initial storage conditions compared to the floods that are generated by short and extreme precipitation. Finally, the authors have examined here maximum annual 7-day precipitation which does not guarantee that this is the same 7-day precipitation sum that have caused a maximum annual flood in the corresponding year.

**Response:**

Thank you for pointing this out. In the revision, we will remove the relevant statement and replace it with an outlook for improving the methodology. Please refer to our response to the previous comment.

Editorial comments

Line 264: regions with winter snowpack accumulation

**Response:**

Modified as suggested.

Line 276: catchments associated

**Response:**

Modified as suggested.

---

## Author Response (AR1)

*Throughout this response, the editor's and reviewers' comments are presented in blue, and our responses are presented in black. The line numbers in red correspond to those in the clean version of the revised manuscript.*

**To Editor:**

The two reviewers both provided detailed reviews for your manuscript and agree that your work is very interesting and the manuscript well written. I am looking forward to reading a revised version of your manuscript that addresses the points raised by the two reviewers.

**Response:**

Thank you very much for handling our manuscript. We are grateful for the constructive comments from the two reviewers, which are invaluable to us in improving the quality of the manuscript. We have carefully addressed all the comments received.

**To Reviewer 1:**

This manuscript proposes a new method for classification of flood generation mechanisms using machine learning that provides the information on the importance of different indicators on the generation of the particular flood event. The method has a potential to overcome the subjective choice of classification thresholds of the previously developed methods. It was tested across European catchments with particular focus on how flood mechanisms were changing in the past decades.

The proposed method has certainly some clear advantages compared to the previous methods and provides a new perspective on classification of flood events. It a very well-written manuscript and I have enjoyed reading it very much. I am certain that this is a substantial contribution to the current knowledge on flood generation processes and their changes. However, although the proposed method has the potential to avoid subjective thresholds, I do not think that the authors have succeeded in overcoming this issue completely. A more prominent attention has to be paid to this issue in the manuscript. I also have some minor suggestions that might help to improve the manuscript and clarify its novelty. Please see my detailed comments below.

**Response:**

We thank the reviewer for the thoughtful comments and suggestions which we believe will greatly improve the manuscript. In the revision, the limitation regarding subjectivity has been emphasized appropriately. The replies to the comments follow.

**General comments**

Abstract: I think the abstract puts too much focus on the changes in flood mechanisms in Europe that is not the most novel findings and instead fails to elaborate on the machine learning approach used here and its advantages compared to previously existing methods. The method implemented in this study is the real novelty, while there are already several studies on changes in flood generation processes in Europe.

Therefore, I suggest the authors to consider to put more stress on the methodological aspects in abstract to show how this study stands out.

**Response:**

Thank you for pointing this out. The explainable machine learning methodology was originally proposed in our previous paper (https://doi.org/10.1029/2021WR030185), and the present manuscript aimed to apply the developed framework to tackle practical problems (i.e., the changes in flood mechanisms in Europe). Therefore, we focused more on the scientific aspect instead of the methodology itself. However, your suggestions are appreciated, and we have made appropriate modifications to highlight more on the methodology by adding "*Recently, numerous studies have demonstrated the skill of machine learning (ML) for predictions in hydrology, e.g., for predicting river discharge based on its relationship with meteorological drivers. The relationship, if explained properly, may provide us with new insights into hydrological processes*" (**lines 11-14**). Moreover, we modified the last sentence in the abstract as "*Overall, the study offers a new perspective on understanding changes in weather and climate extreme events by using explainable ML and demonstrates the prospect of future scientific discoveries supported by artificial intelligence*" (**lines 23-25**).

Selection of thresholds: The proposed methodology has a very strong advantage that it can avoid arbitrary decisions on how the indicators and their threshold are selected for the event classification. However, the authors did not avoid that issue as they have selected the periods for which the effect of recent and antecedent precipitation was accumulated to avoid additional computational effort. This pragmatic choice is understandable and is in line with the subjective choices previous classification studies were making, but it has to be properly stated in the manuscript and a sensitivity analysis on the effect of this choice on the results of the study will be very welcome. Please also see my detailed comments to the corresponding part of the manuscript.

**Response:**

Thank you for pointing this out. We agree that choosing a 7-day window to separate between antecedent and recent precipitation will introduce subjectivities and uncertainties, as we explicitly indicated in the original manuscript "The method has reduced the need for accurate catchment wetness estimates, yet such uncertainty is not eliminated completely, particularly since we chose a 7-day window to separate between antecedent and recent precipitation".

Your suggestion about analyzing the sensitivity of the selection of the separating window is appreciated. In the revision, we supplemented an analysis that uses a 5-day window to separate recent contributions and antecedent contributions, and we found that the main conclusion is not affected by this change. We added the analysis into the new section "**3.7 Limitations and outlooks**", as:

"*In the clustering procedure, we chose to use a 7-day window to aggregate the daily IG scores into a low-dimensional contribution vector for the sake of efficiency in clustering lengthy time series, which could induce inevitable uncertainties and subjectivity. Despite this, additional tests indicate that our findings are similar when using a 5-day window, which is also a common interval to consider flooding drivers (e.g., Rottler et al., 2021). Specifically, based on the 5-day window, the events identified with snowmelt, recent precipitation, or antecedent precipitation as the primary causes account for 15.0%, 47.9%, and 37.1% of all the 55,828 annual maximum peak discharges, which is only slightly different from using a 7-day window.*

*As for the three mechanisms in individual catchments, decreasing the window length has the least impact on identifying snowmelt-driven floods, with the absolute changes in their proportions within 1% for 84.5% of catchments and within 5% for 98.7% of catchments. In comparison, the proportion changes for two other flooding types are more sensitive, with changes within 5% for 83.2% (82.7%) of catchments in terms of recent (antecedent) precipitation-driven flooding. However, this does not affect the conclusion regarding the respective trends in flooding mechanisms (see Fig. S4 in the Supplementary Material), indicating the robustness of the methodology. Despite this sensitivity analysis, we would like to emphasize that the selection of the separating window remains somewhat subjective, and further exploration is needed to avoid a possible bias due to arbitrary judgments in identifying flooding mechanisms*" (**lines 582-595**).

[Figure]

*Figure S4: The same case as in Fig. 7 in the main text, but a 5-day window was used to separate contributions of a variable in recent days and an earlier antecedent period.*

Detailed comments

Line 40-42 I suggest to also mention here the study of Kemter et al (2020) on changes of flood mechanisms in Europe and global analysis of Stein et al (2020)

**Response:**

Thank you for the suggestion. In **lines 56-59,** We added "*Using a multicriteria approach, Kemter et al. (2020) identified the flooding mechanisms in Europe by classifying approximately 174,000 flood peaks and*

*revealed their trends over the past 50 years. Likewise, Stein et al. (2020) analyzed flood events over 4,155 catchments worldwide and classified them into five flood-generating processes*".

Line 48-49: Here I miss mentioning the study of Kemter et al (2020) that did exactly that.

**Response:**

Please refer to our response to the previous comment.

Line 88: Please indicate if the size of catchments was limited to avoid the effect of human influence or was there any other reason for this selection?

**Response:**

Thank you for the suggestion. We added the reason in **lines 95-97**: "*The catchment areas range between 8 km$^2$ and 10,000 km$^2$ — very large catchments, where the effect of spatial heterogeneity of flood drivers tends to be substantial, were not considered*".

Line 100: Please indicate also the lower boundary of catchment sizes to clarify if the size study catchments comparable with the spatial resolution of the hydrometeorological datasets.

**Response:**

Thank you for pointing this out. In addition to indicating the lower boundary of catchment sizes (see the response to the previous comment), we added a clarification "*Note that for smaller catchments under 100 km$^2$ (approximately 0.1° × 0.1°), uncertainties may exist due to the relatively coarser spatial resolution of the meteorological data. Nonetheless, those catchments with large uncertainties will not be considered for the subsequent attribution analysis if ML models cannot capture the relationship between inputs and outputs effectively*" (**lines 110-113**).

Line 103: Please elaborate how the catchment boundaries from two datasets were merged. Are they identical?

**Response:**

We added the clarification "*The catchment boundaries were obtained from readily available GRDC (Lehner, 2012) and GSIM (Do et al., 2018) databases, with GRDC being prioritized when the boundary of a catchment was available in both databases*" (**lines 108-110**).

Line 104-106: I miss here a more motivated choice for the day duration as an indicator for classification. It seems to me that essentially it is a combination of the location and day of the year information. Please provide more information on the nature of the preliminary test performed, particularly if other potential indicators were examined.

**Response:**

In the revision, after "Day length was included in the study since it was shown to improve model accuracy in a series of preliminary tests", we added "*..., including the cases where only precipitation and temperature were used and day length was additionally incorporated. Catchments where day length largely improves accuracy are mainly located in northern Europe*" in **lines 114-116**. The role of day length is as we explained in the original manuscript: "The role of day length implies that the magnitude of these peak discharges can be partially explained by the seasonality presented by day length, which peaks around the June solstice."

Figure 2: The interpretation arrow is not so clear, why does it return back to the input layer? At this point in the manuscript the meaning of the integrated gradients for the features is not yet explained and looks confusing in this Figure. Please add clarification in the caption. Consider indicating the target maximum annual flood in panel a as a point and not as a window. The panel c is rather confusing as there is only one event is being displayed in the panels a and b and the cluster plot is not set in any particular space (i.e., the axes are not indicated). Consider omitting this panel, I think that idea of clustering is understandable without this example only brings more confusion.

**Response:**

Thank you for the suggestions. We removed panel c in the figure. Replacing the window that shows the target peak sample with a point may be confusing because it contains both the observed (black) and predicted (orange) values. To clarify the used window, we added "*The window in the time series of discharge highlights the target output (which is a point)...*" into the caption instead (**lines 137-138**). We further simplified the arrows used and rewrote the caption. The figure and caption now read:

[Figure]

*Figure 2: The workflow of using explainable ML methods for attributing flood peaks (annual maxima of river discharge) to their drivers. (a) Diagrammatic representation of the used LSTM models. The window in the time series of discharge highlights the target output (which is a point) and the window in the inputs indicates the input features used to predict the illustrated peak discharge sample. (b) The feature importance of the inputs for predicting the peak discharge shown in (a), which was obtained by using the ML interpretation technique (namely integrated gradient). The vertical dashed lines in the windows separate the feature importance into a recent 7-day period and an earlier period to calculate the aggregated feature contributions (see main text).*

Line 139-144: I agree that presenting LSTM model in detail is not necessary here, but I think I more detailed explanation on how the structure of LSTM suited for capturing short-term and long-term interaction will be very helpful here, as it can provide the readers with the insights on why particularly this method is more applicable than classifications that are based on subjectively selected thresholds.

**Response:**

Thank you for the good suggestion. We added a more detailed explanation as: "*The effectiveness of the LSTM is partially due to the comparability of its formulation to the hydrological behavior of a catchment. Specifically, the backbone of the LSTM network is composed of recurrent cells that can store previous information from input sequences, which is conceptually similar to the way meteorological information (e.g., precipitation) is stored in the form of soil moisture or snowpack (Lees et al., 2022). The physically*

*realistic mapping from inputs to outputs facilitates gaining hydrologically meaningful insights from subsequent model interpretations*" (**lines 146-151**).

Line 145: It is not clear. Does it mean for each catchment? Please clarify.

**Response:**

The original sentence has been revised as "*To improve the robustness of model evaluation and analysis, we fitted 10 independent LSTM models for each of the 1,077 catchments*" (**lines 162-163**).

Line 153: What is the sample in this case? Maximum annual floods? Please clarify.

**Response:**

The original sentence has been revised as "*...which allows for obtaining the time-wise feature importance of the three input variables for each sample of the output (i.e., daily discharges)*" (**lines 175-176**).

Line 185-187: I do understand authors' arguments on why they had to make this decision and restrict quantification of the effect to 7 and 180 days only. Although, I find it somewhat disappointing. The authors have stated earlier in the manuscript the main advantage of the proposed ML-based method is that one can avoid selecting subjective indicators and their thresholds. In my opinion selecting here 7 and 180 days is nothing else but exactly that kind of subjective threshold that partially impairs the main advantage of the method. If clustering indeed is very time consuming (which is actually surprising to me as in my experience k-mean clustering is not the most time consuming procedure and computational power is hardly a limitation with cluster resources available) at least a sensitivity analysis has to be performed to analyze how the selection of these thresholds affects the results.

**Response:**

Thank you for pointing this out. First, please see above our response to the general comment.

For the efficiency of clustering time series, K-mean clustering with Euclidean distance metric is indeed not a time-consuming procedure, but we have to point out that time series clustering tends to be much more complex as it usually needs Dynamic Time Warping (DTW) as the distance metric. The DTW has a quadratic complexity with respect to the length of sequences (in our study it is 180) compared with the linear complexity of Euclidean distance. Furthermore, in our preliminary tests, we have tried clustering the whole time series as did in our previous study that first implemented such methodology for catchments in the US (https://doi.org/10.1029/2021WR030185). The preliminary results that we obtained are similar to those reported in the present study (three clusters). However, since we anticipate the methodology to be used in a larger-scale analysis in future studies, in the present study we improve the efficiency of the original framework proposed in our previous study by introducing a separating window.

In the Conclusion section, we further added a remark as "*the clustering procedure can be improved by developing algorithms to aggregate daily feature importance adaptively, thereby avoiding the predefined separation window while maintaining high efficiency*" (**lines 633-635**).

Line 188-189: I cannot agree with this statement. The duration will be strongly affected by catchment size and mechanism. The build-up period of snowmelt floods in larger catchments can take up to several months. I also do not think that the provided reference is up to date. Please revise.

**Response:**

Thank you for pointing this out. We meant to describe the hydrological response time to precipitation and snowmelt events, instead of the build-up periods. We revised our original statements to justify the selection of 7 days, as "*The separating window size should cover the period of precipitation and snowmelt events leading to each peak discharge, which depends highly on the local characteristics. After examining the relationship between catchment area and mean event response time, Stein et al. (2020) suggested a synoptic window of 7 days should be sufficient to guarantee the response time for large catchments. As a result, this study used a 7-day period, similar to the practice in most studies that examined flooding causes (e.g., Blöschl et al., 2017; Berghuijs et al., 2019). However, using a shorter period (e.g., 5 days) does not affect the conclusions about dominant flooding mechanisms and their trends (see discussion in Section 3.7)*" (**lines 214-220**).

Line 189-190: It is consistent with previous studies, but they also did not examine if these thresholds are appropriate. Please revise.

**Response:**

Please refer to our response to the previous comment.

Line 192: It is not clear what is the role of multiple-peak discharges here and how they were considered. Please clarify.

**Response:**

We apologize for the confusion. Rather than multiple-peak discharges, we referred to multiple peak-discharges here (note the position of the hyphen). To avoid ambiguity, we removed "multiple" in the revision.

Line 197-203: Please clarify if clustering procedure was performed for all catchments simultaneously or if they were considered individually. If it was performed simultaneously for all catchments, does it mean that if a catchment has very local and specific mechanisms they likely not to be detected by the procedure?

**Response:**

Yes, the clustering procedure was performed for all peak discharges pooled from all catchments simultaneously. To clarify it, the original sentence has been revised as: "*To obtain an overall picture from the individual aggregated feature contributions, we used the K-means method to cluster the results for all annual maximum peak discharges pooled from all considered catchments*" (**lines 223-224**). In the section presenting the cluster results, we further added "*It should be noted that the clustering results here only reveal major patterns widespread in data, with certain local and specific mechanisms unlikely to be detected*" (**lines 294-295**).

Line 206, 277, 402 and elsewhere: Are these maximum annual peak discharges? If yes, please indicate it clearly here and elsewhere.

**Response:**

Thank you for pointing this out. We specified the peak discharges as "*annual maximum peak discharges*" throughout the manuscript.

Figure 3: Does this figure display NSE only for annual maxima or for the complete streamflow time series? Please clarify.

**Response:**

Thank you for pointing this out. In the figure caption, we added "*The NSE values were calculated using all samples in respective testing datasets*" (**line 270**).

Line 294: I think it is a rather a stretch to call streamflow generation that occurs due to excess of soil storage capacity and heavy precipitation as we cannot guarantee that heavy precipitation generates overland flow. In case it is first contribute to increase of soil moisture storage the physical process of streamflow generation will be the same for both drivers. Please revise.

**Response:**

Thank you for pointing this out. We have removed this statement in the revision.

Line 303 and elsewhere: I think "mixed mechanisms" is not an accurate term here as it refers to the occurrence of different mechanisms in the same catchment, but not necessarily simultaneously. Consider using "mixture of mechamisms" instead.

**Response:**

Thank you for the suggestion. We will replace "mixed mechanisms" with "*mixture of mechanisms*" throughout the revised manuscript including figures.

Figure 6: Please add an explanation for the mixtures in the caption. Please also clarify how the classes for two processes are formed, do the corresponding two processes have to generate more than 70% of annual maxima?

**Response:**

We defined a catchment dominated by a mixture of two processes as the case where the difference between the proportions of the two processes is less than 70% (say one is 35% and another is 65%), in order to distinguish it from single mechanisms. We added an explanation in the figure caption, as: "*Mixture means the associated catchments are dominated by two or more flooding mechanisms. For example, mixture (r+s) indicates either recent precipitation (r) or snowmelt (s) is the primary cause of the annual maximum*

*discharges for the associated catchments, and the difference between the two proportions is less than 70%*" (**lines 360-362**).

Line 338-340: I think it will be helpful to relate here to the findings of Stein et al (2021) (doi: 10.1029/2020WR028300) on the controls of catchment characteristics on the dominance of different flood mechanisms

**Response:**

Thank you for the suggestion. We added "*In addition to elevation, slope, and snow fraction, the study by Stein et al. (2021) on catchments in the United States demonstrated that other catchment characteristics (e.g., aridity, precipitation seasonality, and mean precipitation) also significantly influence flood generating processes. An in-depth investigation of how geographic and climatic characteristics affect flood mechanisms in European catchments can be expected in future studies*" (**lines 391-395**).

Line 350: Consider using term "pre-defined" criteria instead of "manual" as it is not so clear.

**Response:**

Replaced as suggested (**line 405**).

Table 1: Consider also adding catchment sizes to the comparison as I expect that there is a difference between these studies also in that regard.

**Response:**

Thank you for the good suggestion. We added the catchment sizes for comparison in **Table 1** and in **lines 429-431**, we further added "*In addition to methodological differences, the inconsistent catchment samples are also responsible for the divergent attribution results in different studies. As shown in Table 1, the catchments examined in this study are generally smaller, which tend to be more susceptible to rainfall with high intensity*".

Line 379: I think the study of Kemter et al 2020 also should be mentioned here.

**Response:**

We added the reference as suggested (**line 446**).

Line 381-383: This note would be more helpful earlier before the comparison of the results. Consider moving this part up.

**Response:**

Thank you for the suggestion. We have moved the notes next to other explanations for the divergences in results (**lines 431-435**), which we thought might be more appropriate.

Line 385-389: I think it might be worth mentioning here the work of Tarasova et al 2020 (doi: 10.1029/2019WR026951) that investigates how using different data sources for the same indicator affects event classification

**Response:**

Thank you for providing the useful reference. We added "*Tarasova et al. (2020) conducted a rigorous uncertainty analysis of input data for a runoff event classification framework, emphasizing the importance of developing novel indicators to reduce these uncertainties*" (**lines 435-437**).

Line 404-407, 427-431: These parts would be more suitable in the dedicated Method section

**Response:**

Thank you for the suggestion. We moved these parts to the new subsection "***2.4 Trend analysis of flooding mechanisms***" in Method.

Figure 7: Please indicate how many catchments are the basis for Sankey plot in the caption. Please also clarify the origins of the p value in the caption. The information provided on methodological aspect of trends in this caption is not sufficient. Please add a corresponding section in the Methods. Panel b: I am wondering if the results of trend analysis are not so clear due to regional differences in the direction of trends. Looking at the results of Kemter et al (2020) it seems that there are disparate trends for different regions that can be obscured when mixed together. Perhaps something worth mentioning in the corresponding text.

**Response:**

Thank you for the suggestions.

In the caption, we added "*The proportions were calculated based on 846 catchments, where at least 15 years of records were available in each period*" (**lines 474-475**) and "..., *with their significance being assessed by the modified Mann-Kendall test*" (**lines 477-478**). The details have been provided in the new section "***2.4 Trend analysis of flooding mechanisms***", as "*Specifically, at the continental scale, we estimated the overall trends of various flooding mechanisms based on their respective proportions within all the annual maximum peak discharges per year. At the catchment scale, to capture the variations of flooding mechanisms over different periods, we calculated the proportion series using a 20-year moving window in each catchment. The 20-year time frame was used to ensure an adequate sample size for reliably estimating the intra-period proportions and also to guarantee enough periods to observe decadal variability (Pagano and Garen, 2005). Only proportions that were calculated with at least 10 years of peak discharge data in each window were used to estimate the trend slope*" (**lines 238-244**).

For panel b, before introducing the results of trends in individual catchments, we added "*Note that Fig. 7b only presents the overall trends in flooding mechanisms at the continental scale, while disparate trends may exist in different regions that could cancel each other out*" (**lines 481-482**).

Line 455-459: It would be helpful if this information is provided in the dedicated Method section.

**Response:**

Thank you for the suggestion. We moved these parts to the new subsection "**2.4 Trend analysis of flooding mechanisms**" in Method.

Figure 8: It is not clear why the lines of the plot do not correspond to the whole extent of time axis. Please clarify or correct. In region 1 and region 2 it seems that there is certain periodicity in the data, it would be helpful if the authors would add a short discussion on suitability of monotonic trends analysis in such cases. Please also consider adding geographical indications for regions instead of numbers. This will make this figure easier to interpret. Please also add the number of catchments in each of the considered regions.

**Response:**

For the time axis, because the 20-year moving window is used, the range reduces to from 1950-2020 to 1960-2010. For clarification, in the caption, we added "*The proportions were calculated by a 20-year moving window, while precipitation and temperature were smoothed by using a 20-year moving average window, with their values at central positions in time windows*" (**lines 518-520**).

For the possible periodicity in data, we added "*Note that here we merely examined the monotonic trends within data over the 70 years, while the trends may vary piecewise (e.g., the changes in maximum weekly precipitation in the Alps and southeast France), the impact of which on flooding mechanisms deserves further research*" (**lines 532-535**).

For the geographical indications and number of catchments, we added them to the figures as suggested.

Line 492, 499. Caption of Figure 9: It is not clear which length is meant here. Please clarify.

**Response:**

In the caption of Figure 9 (now **Figure B1** in **Appendix B**), we explained the length as "*The mean resultant length is a measure in circular statistics between 0 and 1 that reflects the spread of a circular variable, with 0 representing the spread of flood dates evenly distributed over the year and 1 representing the spread concentrated at one day*" (**lines 654-656**).

Line 486-504: This part is not very well connected to the previous narration and provides yet another new results for which methods were not clearly elaborated in the Method section. Consider omitting it or revise.

**Response:**

Thank you for pointing this out. We moved the description of Figure 9 into **Appendix B** and simplify the part original in lines 486-504, and it now reads: "*A change in flooding mechanisms may affect the seasonality and magnitude of flooding, which might ultimately impair the current flood risk management measures. For example, in catchments previously dominated by snowmelt, increasing floods from extreme precipitation and soil moisture excess may lead to shifted flood mean dates and less concentrated seasonal patterns (as exemplified in Fig. B1 in Appendix B). By simulating daily discharge for a reference period (1961–1990) and a future period (2071–2099), Vormoor et al. (2015) predicted that floods in some Nordic catchments could even shift from spring to autumn as rain replaced snowmelt as the dominant flood-*

*inducing process. These results suggest that, in a warmer climate, flood risk predictions in snowmelt-affected catchments should consider the interconnection between changes in flooding drivers and seasonality*" (**lines 537-544**).

Line 539-543: I would recall here how "recent" and "antecedent" precipitation were defined in this study, because despite what this part claims the definition of these two indicators were set arbitrary by selecting corresponding number of days during which the effect was evaluated.

**Response:**

Thank you for the suggestion. In the revision, we updated the relevant sentence to read as follows: "*With the ML-captured feature importance of precipitation, temperature, and day length for predicting annual maximum discharges, we aggregated driver contributions in the recent 7 days and an earlier period (back to 180 days) and then applied cluster analysis to group them based on similar patterns*" (**lines 603-606**).

Line 549: The term "perspective of catchment average" is not very clear here without the context. I think it would be clearer to just indicate that these methods did not perform an event-based classification and instead identified one single dominant driver per catchment.

**Response:**

Thank you for the suggestion. We modified "...some of which were obtained taking a perspective on catchment averages" as "...*some of which did not perform event-based classifications but rather identified the overall mechanisms within individual catchments*" (**lines 616-617**).

Conclusion section: A statement about the dependence of the results on the performance of the ML model for the proposed classification method would be very welcome in this section. Moreover, same as for abstract more focus on the newly developed ML-based classification method instead of changed in the mechanisms will be welcome here to highlight the novelty of this study.

**Response:**

Thank you for the suggestion. To clearly highlight the novelty, in the first paragraph, we replaced the relevant sentences with "*To investigate whether flooding mechanisms have changed in European catchments, this study introduced a novel explainable ML method to identify flooding mechanisms. Compared with conventional classification approaches, where the results are usually dependent on appropriate flood process definitions and sensitive to the selected indicators and threshold parameters, the combination of explainable ML and cluster analysis is able to avoid such predefinitions and reduces subjectivities in identification processes. With the ML-captured feature importance of precipitation, temperature, and day length for predicting annual maximum discharges, we aggregated driver contributions in the recent 7 days and an earlier period (back to 180 days) and then applied cluster analysis to group them based on similar patterns*" (**lines 599-606**).

Moreover, at the end of the conclusion section, we replaced the original outlook with an outlook for improving the methodology, and it now reads "*Overall, this study highlights the usability of explainable ML in helping uncover complex and possibly non-linear changes in weather and climate extreme events in*

*the warming Earth system. With more large-sample hydrometeorological datasets becoming readily accessible, one next step is to extend the research to a larger scale for a better understanding of variations in flooding mechanisms globally. Still, many challenges remain for future work, providing potential research opportunities. For example, the clustering procedure can be improved by developing algorithms to aggregate daily feature importance adaptively, thereby avoiding the predefined separation window while maintaining high efficiency. Moreover, regional LSTM models that incorporate static catchment attributes can be employed to capture the spatial variations in flooding mechanisms and quantify the influence of catchments' geographical and climatic conditions on flooding processes. In addition to the integrated gradient method used in this study, other interpretation techniques might be explored further to uncover potentially valuable information when more input variables are included*" (**lines 629-638**).

Line 563-565: I think the authors have to be more cautious here with this statement, because there might be strong regional differences (i.e., there are disparate patterns in precipitation changes in Europe). Moreover, the term extreme precipitation is much more often related to very short precipitation (i.e., less than 1 day), while 7-day long precipitation can substantially affect the storage of the catchment and lead to soil moisture excess floods and hence the resultant magnitude of the flood will depend much more on the initial storage conditions compared to the floods that are generated by short and extreme precipitation. Finally, the authors have examined here maximum annual 7-day precipitation which does not guarantee that this is the same 7-day precipitation sum that have caused a maximum annual flood in the corresponding year.

**Response:**

We appreciate you pointing this out and we agree with you. In the revision, we have removed the relevant statement and replaced it with an outlook for improving the methodology. Please refer to our response to the previous comment.

Editorial comments

Line 264: regions with winter snowpack accumulation

**Response:**

Modified as suggested (**lines 314-315**).

Line 276: catchments associated

**Response:**

Modified as suggested (**line 327**).

**To Reviewer 2:**

This paper presents a large-sample study to detect different flooding mechanisms across Europe. I have to admit that initially, I was skeptical about this study. But while reading the manuscript for this review, I became quite excited about the presented work.

To my knowledge, it is the first time that such an analysis (detecting flooding mechanisms and analyzing the change over time) is made using a) deep learning models (here LSTMs) and b) methods from the field of explainable AI (here integrated gradients). My views on LSTMs is no secret and I have often said that you can do more than "just fitting streamflow records" with these models, so naturally, I am quite excited to see someone coming up with such an idea.

Additionally, I think this paper is exceptionally well written and at least for me personally, everything seemed pretty clear and reasonable. For example, the authors make a couple of assumptions (like grouping the integrated gradient signal into two groups of a) the last 7 days and b) all other days before that), but their reasoning for all these assumptions is clearly articulated and to me, they make sense.

In general, I think this is a very interesting study that fits into the scope of HESS and I only have a few general comments. Note, I already spoke to the first author during the EGU GA but for the sake of transparency, I will add all points here again. Please also note that, due to the overlap of this research with our own research in the past, in many of the studies that I reference I'm either the first or a co-author. I do not mention these studies here, because I want them to be cited in the manuscript, but I think they help to explain my reasoning.

**Response:**

We thank the reviewer for the positive comments and insightful suggestions of our work, which will not only help us to improve the present manuscript but also provide us with good ideas for future works. The replies to the comments follow.

**1. Training setup**

You train LSTM models individually for each basin, instead of one model on the combined data of all basins. Again, I'm very biased on this topic but I think there are multiple studies that show that the recommended way for training LSTMs is the latter (on all basins at once, using meteorological timeseries features and static attributes). The regional modeling setup was introduced in 2019 (see Kratzert et al. 2019) and further discussed in Nearing et al. (2021) (see Fig 2). One study that follows the regional training scheme is even cited in the manuscript (Lees et al. 2021).

The question is, is this important in the context of this study? This is a good question that I asked myself quite a lot over the last few days. On one side, I think it is important to follow best practices when working with any model. The benefit of the LSTM is that it can learn a very general understanding of the underlying processes if it is trained on a variety of basins. Nobody would probably train a conceptual model in a regional calibration scheme, if she/he is only interested in a particular basin. On the other hand, the authors are not interested in getting the best-possible streamflow performance, but to learn about flooding-mechanisms from the model.

From my experience, I would assume that changing the training setup would not change the results of this study (i.e. the clusters of different flooding mechanisms found here). What might change is the number of

basins that are considered in their study (because of the NSE threshold). However, even (or especially?) if the results of this study do not change, I would suggest training an LSTM on the combined data of all basins and to re-run the analysis, to reflect the best-practices of the chosen model. During the EGU, I offered Shijie Jiang help with setting up such a run as I have the code + resources available. I would be happy to help and don't want/expect any co-authorship/acknowledgements for that.

**Response:**

Thanks for the insightful and enlightening suggestions that are well worth considering.

First of all, we agree with you that training a regional model (i.e., training one single model for all catchments, including both meteorological time series and static catchment attributes as inputs) is a good practice, with the benefit that the ML model can learn relationships from a large sample of hydrological variability and enables to use the learned relationships for better predictions in individual catchments. However, as you mentioned as well in your comment, the choice of which strategy to apply (local model vs. regional model) should be in line with the research purpose.

The key idea of the study is to identify flood generation mechanisms based on distinguishable patterns of meteorological variables' contributions. With the local models, since meteorological variables are the only inputs, we are able to focus on how they explain the temporal variation of discharges. In comparison, for a regional model that is supposed to capture both temporal and spatial variations in discharge peaks, it is challenging to distinguish how meteorological variables contribute to temporal variation in flooding within catchments from how they contribute to the spatial variation in flooding across catchments. Moreover, the feature importance of meteorological variables may also be obscured by the importance of catchment characteristics, for the sake of explaining (possibly larger) spatial variation. This can result in changed contribution patterns of meteorological variables to individual peak discharges.

To demonstrate the above point, we have conducted a small-scale test on around half of the studied catchments (530 catchments) using the regional LSTM model, in which static attributes were directly concatenated with meteorological variables at every time step. The static variables contain 12 catchment attributes available in the GSIM dataset, such as the area, slope, elevation, snow fraction, climate type, etc. Here we tested a random subset of catchments instead of all the 1,077 catchments mainly because of the limitation of our memory resources (the median sample size of individual catchments is 20,455, and each sample has a dimension of [180, 15]). As a result, we found that the feature importance pattern of meteorological variables had changed significantly because the feature importance had been re-assigned to some catchment attributes to reflect the difference in flooding schemes across catchments. For example, the model outputs show a large reliance on elevation in the Alps area. However, elevation can affect a variety of aspects of flow behavior because mountainous catchments are typically smaller, receive greater precipitation, and have a higher snow fraction, resulting in a possible confounding factor to different flooding mechanisms. Introducing confounding factors will make it challenging to distinguish flooding mechanisms based on drivers, which is however out of the scope of the present study.

As a clarification that we used individual models instead of the regional model, we added a short discussion in the new section "**3.7 Limitations and outlooks**": "*In this study, we trained LSTM models individually for each catchment, while some studies have suggested that training a regional model for all catchments at once may be a better practice (e.g., Nearing et al., 2021). In the latter case, both meteorological time series and static catchment attributes are used as inputs to distinguish response behaviors across time and space, with the benefit that the ML model can learn more general relationships from a larger sample of*

*hydrological variability (Kratzert et al., 2019b). However, introducing catchment attributes may prevent the identification of flood generation mechanisms based on distinguishable patterns of meteorological variables' contributions, which is the main objective of this study. Interpreting flooding mechanisms with regional LSTM models may become more challenging than with local LSTM models that use only meteorological time series, since some catchment attributes would confound the interpretation. Therefore, here we employed local models. Nevertheless, we note that using regional models can provide insight into how flooding mechanisms vary spatially, particularly for how the spatial distribution is affected by the geographic and climatic characteristics of catchments, and it merits more exploration in future studies*" (**lines 561-571**).

In the Conclusion section, we added an outlook as "*Moreover, regional LSTM models that incorporate static catchment attributes can be employed to capture the spatial variations in flooding mechanisms and quantify the influence of catchments' geographical and climatic conditions on flooding processes*" (**lines 634-636**).

2. Input variables

In this study, you only use precipitation, temperature and day-length (as a proxy for solar radiation) as model inputs. I agree that these are the main-drivers for flooding mechanisms but I think the exciting thing about data-driven models is that if there is anything else, these models are pretty good at finding it. That is, if provided with more input features, the models would find other flooding mechanisms, if they are deducible from the data. When I discussed this point with Shijie Jiang at the EGU, he mentioned that he has/had a hard time to interpret the contribution signal for different features, and I agree, it is much simpler and more intuitive to reason about the meaning of high feature importance of e.g. temperature in the recent days. However, how exciting would it be to find that the model finds a flooding mechanism that is not straightforwardly explainable with the patterns we already know? If you have easy access to more input features, I think it could be interesting to run the models with more input features. If not, I think it is ok not to do it, but then it could be worth adding this to the discussion.

**Response:**

Thank you for the constructive suggestions. We agree that including more inputs is beneficial to uncovering patterns related to flood mechanisms that are likely to be overlooked. During our preliminary tests, we had run models with daily averaged sea level pressure, relative humidity, and radiation as additional inputs, which did indeed lead to more clusters in terms of feature importance patterns. However, considering the purpose of the study, we simplified the inputs to include only precipitation, temperature, and day length as input variables in the final manuscript, since their results corresponded to three well-known flood mechanisms at the catchment level. This allows us to directly compare our findings with those of other studies conducted with classical methods and perform the subsequent trend analyses. The performance did not drop much by not including more variables.

To highlight the possibility that the methods can provide insight into previously seldom known patterns, in the new section "**3.7 Limitations and outlooks**", we add "*Moreover, to strike the balance between model interpretability and accuracy, we only selected daily precipitation, temperature, and day length as meteorological inputs. The combination of inputs results in uncovering three well-known flooding mechanisms, which allows us to make a direct comparison with findings from other studies that used*

*classical methods. However, with more input variables incorporated into the model, the methodology may be able to recognize more distinct patterns in terms of input contributions. In principle, this could allow identifying flooding mechanisms that are often overlooked and may not be easily explainable with well-known processes. However, it would likely be much more challenging to make sense of the flooding mechanisms from the likely much more complicated patterns in input feature importance, which we leave for future studies*" (**lines 573-580**).

**3. Data split**

In L 138f, you mention that some hyper parameters were determined considering the model performance and efficiency. To me it is unclear which data was used for this hyperparameter tuning. Usually, the validation split (note, here I am referring to a 3-fold split with a train, validation and test set) is used for these kinds of hyperparameter tunings. However, from the explanation in L146, I can only estimate that this was done with a 2-fold split, thus on the test data?

**Response:**

Sorry for the unclear statements. In the original manuscript, we omitted to report that we actually performed a 3-fold split (training:validation:testing = 49%:21%:30%) on the dataset when training models. We modified the original sentences as: "*Each independent model was trained and tested based on samples that were randomly split in a 7-to-3 proportion. During the training process, a portion of the training data (70%) was repeatedly used to update the model parameters every epoch until no further decrease in the loss function was observed on the remaining 30% (also known as validation data). The trained models were independently evaluated on the testing datasets*" (**lines 163-166**).

**4. Data split pt. 2**

In L146, you say that the timeseries data was randomly split in a "7-to-3 proportion". I am not 100% sure but I think randomly splitting timeseries data is not optimal, especially when considering the overlap of different samples because of their input window size. I think much more common is a k-fold cross validation in time (none random).

**Response:**

Thank you for pointing this out. We performed a random split instead of the more common temporal k-fold cross-validation due to two reasons. Firstly, runoff data available in the GRDC dataset is not temporally complete in many catchments in Europe, with missing data sometimes occurring for several months or years irregularly. This complicates carrying out a unified temporal k-fold cross-validation across these catchments. Secondly, using a random split rather than temporal k-fold cross-validation is based on the purpose of our study, which facilitates our subsequent trend analysis for flooding mechanisms, as we emphasized in the revised manuscript, "*Note that here we adopted a random sampling strategy instead of the time-series splitting strategy with fixed time intervals in order to enable capturing the overall hydrometeorological variability observed across various periods*" (**lines 166-168**). In **lines 168-170**, we further added a remark as "*It should be emphasized that while the random sampling strategy is appropriate with respect to the purpose of this study, it might not be the best practice if the models were developed for prediction tasks, particularly if they were to be applied to new datasets*".

I am curious, if you train/eval the models in a 7-to-3 split random fashion, how could you a) guarantee an equal number of model predictions per timestep and b) guarantee that every time step was e.g. evaluated at least once and c) guarantee that the flood peak was in the validation and not the training period? Because from L 169, it seems like you used all 10 models for all peaks, but some peaks are certainly in the training period, right?

**Response:**

Thank you for highlighting this unclear point in the original manuscript. The average feature importance score for one peak was computed by averaging the scores from all 10 models, regardless of whether the peak appeared in the training or testing dataset. We should emphasize that the model was used for statistical purposes instead of a prediction task, thus the split of the training dataset and the testing dataset is only to ensure the model has learned a generalizable relationship between variables. The generalizable relationship should hold not only for the testing dataset but also for the training dataset. Because of this, the interpretation results from the training dataset were not excluded, similar to other geoscientific studies that aimed to gain insights through machine learning interpretations (e.g., Barnes et al., 2020; Toms et al., 2020).

To better clarify this point, we will add Figure S1-S3 in the supplementary material in the revision, which gives the respective feature importance scores from the 10 models used to derive the examples of Figure 2b and Figure 4 in the main text. Some of the scores are derived when the flood peaks are in the training dataset (the solid lines), while others are in the testing dataset (the dashed lines). It can be seen that these feature importance scores present consistent patterns. We will add the following description: "*Note that the averaged IG scores for an individual peak were computed by averaging the scores obtained from all the independent 10 models, regardless of whether the peak was part of the training data or the testing data in the models. Overall, the IG scores extracted from the 10 models for each target peak discharge generally follow a similar pattern, though with inevitable differences due to randomness and uncertainties in training processes (see Figs. S1–S3 in the Supplementary Material for examples)*" (**lines 198–202**).

[Figure]

*Figure S1: The integrated gradient (IG) scores of precipitation, temperature, and day length extracted from 10 independent models for predicting the peak discharge that is illustrated in Fig. 2a in the main text. The 10 models were trained and tested with different randomly split datasets, where the target discharges in*

*runs #1, #3, #6, #7, and #10 were in the testing dataset (indicated by dashed lines), and the target discharges in the remaining runs were in the training dataset (solid lines). All the y-axes use the same scale for better comparisons. The average of the 10 sequences across different models generates the heatmaps shown in Fig. 2b in the main text.*

[Figure]

*Figure S2: The same case as in Fig. S1 to generate heatmaps illustrated in Fig. 4a in the main text.*

[Figure]

*Figure S3: The same case as in Fig. S1 to generate heatmaps illustrated in Fig. 4b in the main text.*

**References:**

Toms, B. A., Barnes, E. A., & Ebert-Uphoff, I. (2020). Physically interpretable neural networks for the geosciences: Applications to Earth system variability. Journal of Advances in Modeling Earth Systems, 12, e2019MS002002. https://doi.org/10.1029/2019MS002002

Barnes, E. A., Toms, B., Hurrell, J. W., Ebert-Uphoff, I., Anderson, C., & Anderson, D. (2020). Indicator patterns of forced change learned by an artificial neural network. Journal of Advances in Modeling Earth Systems, 12, e2020MS002195. https://doi.org/10.1029/2020MS002195

---

## Referee Report (RR1)

In the first rounds of reviews, I commented on three topics:
1. Model setup (single basin LSTMs vs regional LSTMs).
2. Selection of input variables.
3. Data splits.

In most parts, the authors argued that my concerns do not require any change to their manuscript apart from adding a new section on "Limitations and Outlooks" that basically lists these concerns.

In one point (selection of input variables), I think it is a pity that the authors limit themselves to their selection and do not report potential novel insights (see detailed comment below). On another point (model setup), I do not agree with the argumentation of the authors, why they think that an inferior model calibration procedure is justified, but in the end it is their decision. The last point (data splits), is probably the most critical. Reading that "*the model was used for statistical purposes instead of a prediction task*", which according to the authors justifies the use of a data split that is not independent of the training data is in my opinion a red flag for any kind of study that involves a model. Even if the results would not change (what I sincerely hope for the authors) it is a scientifically wrong thing to do. And to justify it by saying a per-gauge time split is more complicated is in my opinion a really bad argument.

I added some specific comments to the three points below but in general, I have nothing much to add to what I said in the first round. Ultimately, it is up to the editor to decide how to proceed.

**1. Model Setup**

The topic of discussion is, whether it is necessary/recommended or not for this study, to use a regional training setup with LSTMs rather than the per-basin calibrated LSTMs the authors chose to use. Here you say

"*However, as you mentioned as well in your comment, the choice of which strategy to apply (local model vs. regional model) should be in line with the research purpose.*"

No, this is not what I have mentioned. I deeply believe that no matter what you want to do with a model, you should always set up a model in the state-of-the-art for whatever model you use. Again, you would probably not calibrate a conceptual hydrology model in a regional fashion if you are interested in a single basin.

There is no reason to believe that a model that is consistently and significantly better than another model (or the same model but trained with a different setup) should not have a better understanding of the underlying physical processes (i.e. flood generation).

LSTMs trained to local basins suffer from two problems:
1. They witness saturation effects (which is easily visible when plotting discharge time series and e.g. comparing the max peak the model is able to output during the training period vs validation period).

2. They are unable to model any process that did not happen in that particular basin during the training period.

The solution to both of these problems is training LSTMs on a multi-basin dataset, which afaik should be the one and only way how LSTMs should be applied for rainfall-runoff modeling. Looking at e.g. Frame et al. (2022), one can see how the regionally trained LSTM is able to predict unprecedented events in individual basins (e.g. being trained only on data with q < HQ5 and then tasked to predict > HQ100), which a locally trained model will never be able to do.

"*With the local models, since meteorological variables are the only inputs, we are able to focus on how they explain the temporal variation of discharges. In comparison, for a regional model that is supposed to capture both temporal and spatial variations in discharge peaks, it is challenging to distinguish how meteorological variables contribute to temporal variation in flooding within catchments from how they contribute to the spatial variation in flooding across catchments.*"

I think I disagree with everything you have said here. First, just because something is "more challenging", that doesn't mean in my opinion that the easier (but potentially wrong/biased) approach is justified. And in more detail, I also don't see the problem of looking at flood generating processes through integrated gradients, if static features are also model input arguments. Sure, the static arguments influence the model dynamics, but so does the fact that you have different model weights for every catchment. Just as an example, imagine you have a regionally calibrated model and two different basins A and B. Now we force the model with the same timeseries of meteorological features but using the different sets of static features, corresponding to basin A and basin B. Now imagine that in one basin, the model would produce a flood peak that, by looking at the integrated gradients of the meteorological features, is driven by the precipitation of the last few days, while in the other basin it produces a flood peak that is driven by temperature of the recent days and precipitation of some weeks ago. The difference here is certainly, because the static attributes are different. But does this have an influence on your analysis? I think not, and most likely this model has a much better understanding of the underlying processes. In your case, you would identify these two peaks as generated by different processes, which is most likely correct. So sure, static features have an influence on the model dynamics, but your analysis will still tell you which meteorological inputs are the most important for the peak flow. To be entirely honest, I fail to see what is the real challenge here.

"*...some studies have suggested that training a regional model for all catchments at once may be a better practice (e.g., Nearing et al., 2021).*"

I come back to this point, because it sounds like this is just a "suggestion". Let me tell you from experience from the operational side (not only at Google but various companies/agencies) but also from literature: I am not aware of anyone that applies LSTMs that is not using regionally trained LSTMs. In my view, there is really no discussion happening around whether you should use regionally or single basin calibrated models. Again, it is not *only* about better performance but also *why* the model can achieve this higher performance (see list above). I don't want to

step on someone's toes but from my point of view, the reason why people are sticking to single basin LSTMs is because it was hammered into their head for decades that single basin models are better at capturing local processes than regional models, which is true for conceptual models but it is not anymore for LSTM-based models.

**2. Input feature selection**

"*We agree that including more inputs is beneficial to uncovering patterns related to flood mechanisms that are likely to be overlooked. During our preliminary tests, we had run models with daily averaged sea level pressure, relative humidity, and radiation as additional inputs, which did indeed lead to more clusters in terms of feature importance patterns*"

Reading that you already did these experiments, which led to different results, but decided to not include them because they probably don't agree with the "known" patterns is in my opinion simply sad. I agree with the first reviewer on something he mentioned in a slightly different context, which is that you artificially limit yourself in this study. You have the potential to report new findings and even if you can't explain them, you could still present them and potentially start a new discussion in this area. Instead you limit yourself and the model to only check if your method identifies the same patterns as previous studies with different approaches.

"*The performance did not drop much by not including more variables*"

I think this is not important here, but as you said yourself, the clusters of flood generating processes changed. So which one is "the truth"?

I have nothing to add to this point and ultimately, it is the decision of the authors. I think it is just a missed opportunity and generally, I am not happy with the fact that all major reviewer comments are put into a new "Limitations and outlooks" section and that this suffices.

**2. Data split**

I think this is probably the most worrisome point to me. I think the applied data splitting (random in time) introduces a severe data leakage. You are not testing your method on unseen data, not even in these cases where you predict on a "test sample". The reason is that if you randomly select timesteps to be train/test data then three adjacent timesteps could e.g. be "train - test - train". Since each timestep is predicted from an input sequence of 180 days, e.g. the first train sample and the test sample only differ in a single time step of data. And without any doubt, the discharge data is highly auto-correlated. In Figure S1-S3, it is e.g. possible that for the models for which this time step appears to be in the test data (dashed lines) the previous and next timestep of that event is a training step. And in this case you can not at all argue that this is

independent test data. And that these plots show that the signal is the same for all 10-folds could be just because of this effect, because there is not really any point in your input time series that wasn't seen during training of any of these models.

"*Firstly, runoff data available in the GRDC dataset is not temporally complete in many catchments in Europe, with missing data sometimes occurring for several months or years irregularly. This complicates carrying out a unified temporal k-fold cross-validation across these catchments.*"

This is related to what I wrote above: Just because something is "more complicated", this doesn't mean you shouldn't do it. In fact, it isn't that hard to loop over basins and do time series splits per basin. How difficult it is to include different data splits for each basin in your training pipeline depends on your code. It is a built-in function in the open source library NeuralHydrology (https://github.com/neuralhydrology/neuralhydrology/ Disclaimer: I am one of the developers) and would work out of the box if you would use this for training your models.

"*We should emphasize that the model was used for statistical purposes instead of a prediction task, thus the split of the training dataset and the testing dataset is only to ensure the model has learned a generalizable relationship between variables.*"

I don't understand your point here. For any kind of statistical analysis with any model (data driven or not) you want to make the analysis on an independent dataset. The point is, that your test dataset is not independent of the training dataset as there is data leakage.

"*The generalizable relationship should hold not only for the testing dataset but also for the training dataset.*"

This is not necessarily true. First, in the extreme case you could overfit on the training data, meaning your model remembers every sample and thus is not generalizing at all. Second, have you ever looked at e.g. the NSE of an LSTM during the training period and compared this to the test period (with a non-random splitting). You will see that the LSTM achieves a much higher NSE during the training period and I would be more than cautious to draw any conclusions from this on the models generalization capabilities.

References:

Frame, J. M., Kratzert, F., Klotz, D., Gauch, M., Shelev, G., Gilon, O., Qualls, L. M., Gupta, H. V., and Nearing, G. S.: Deep learning rainfall–runoff predictions of extreme events, Hydrol. Earth Syst. Sci., 26, 3377–3392, https://doi.org/10.5194/hess-26-3377-2022, 2022.

---

## Referee Report (RR2)

Dear authors, I am thankful for your responses and I want to say that I am sorry if I appear to be the "mean reviewer 2" in this revision process.

Let me start with an TLDR;

- I think we can stop arguing here and conclude that we need to agree to disagree. I think I could go on forever but it probably adds little to this review process. So to save us all time, I will stop arguing after this review and I only reply here to clarify a few of the statements/misunderstandings.
- That being said, I think the first point, regarding data splitting (TLDR; use the results of the test splits in the paper, not the averaged results of training periods) should be taken seriously.
- One last thing: I mentioned this to Shijie at the EGU in personal communication and I don't want this review process to negatively impact what I said: I think this is an exciting study and I am happy to see people using LSTMs for these kinds of studies. I think it would have been easy to make this paper even more exciting but I understand that I need to wait for another publication from your group to get answers to my questions.

**Data splitting**

I am very happy to hear that running your analysis only on the test splits did barely change the results. What I don't understand is, why you did not change the manuscript so that these results are used in the paper. I don't think that I need to search for literature here to say that in any modeling study, you should use independent test data (which you have!) to analyze/interpret your model. We already know that this is not affecting anything in your results, so it should be a no-brainer to change this and standard practices in statistical analysis with models.

**Input selection**

I think it is somewhat funny that on this point you argue with NSE performance (that it did not or did negatively affect the NSE, when adding more inputs), while in the other points (e.g. multi-basin model) you say that performance is not really important.
Anyway, there is not much to add from my side, I think it would have been a great addition to the paper to include and analyze more input features, especially because it seemed like you already did the experiments. You yourself said in the first rebuttal

*"During our preliminary tests, we had run models with daily averaged sea level pressure, relative humidity, and radiation as additional inputs, which did indeed lead to more clusters in terms of feature importance patterns"*,

which to me sounded exciting and I was curious to hear more about clusters that are different from previous literature and about interpretations/hypotheses of these clusters. You decide to

keep it simple, your choice. If I understand you correctly you want to come back to this in a future publication and I am looking forward to diving into the results then.

**Model setup**

Let us just say that we disagree. I certainly do believe that, no-matter what you want to do, if you use a tool, you should use that tool correctly. Following your argument (which tries to say the other extreme is possible) I could make such a study with uncalibrated hydrology models and then say that my results have any meaning?!

From a very basic point of view: I hope we can agree that a model that is better in modeling the task at hand (meteorological forcings in, discharge out) should have a better understanding of the underlying system/processes, right? Now if I want to make a study that investigates the underlying process understanding of a model, then why would I willingly pick a model that has a worse understanding of the underlying processes? If the model would not have a worse process understanding, it would not be generally worse in predicting discharge, or would it? Since your study is about model interpretation of process understanding, I do not see how this is decoupled from performance. But I see how this example will be used again to say "he only cares about performance", which is wrong. But I'm not so naive to think that good process understanding is decoupled from good performance.

Your line of argument is the ease of interpretation, I got it. But what I am saying is that this is a bad argument. And again, this is not "because the regional model has a higher NSE", but because, without any doubt, this model has a better process understanding. And your study is about analyzing such a process (flood generation). Funnily enough, the one thing where single-basin models have their biggest bias (compared to regional models) is flood peak prediction, which I also mentioned in my previous review (see point on saturation). One could argue that having a (negatively affecting bias) in flood predictions is also affecting the flood generating processes in the model.

---

## Author Response (AR2)

**The line numbers indicated here are consistent with those in the CLEAN (i.e., no changes tracked) version of the revised manuscript.**

**Response to Editor**

**Editor:** Your manuscript has again been reviewed by the initial two reviewers. While reviewer 1 is very happy with the revisions, reviewer 2 points out that major revisions are still needed. Specifically, he points out that model performance and validity would profit from (1) a regional instead of a local model fit and (2) improvements in the cross-validation strategy. I would like to ask you to address these concerns in the revised version of your manuscript.

**Response:** It is our sincere gratitude to the two reviewers who helped improve our manuscript. It is unfortunate that we were unable to convince Reviewer 2 of the reasonability of local models in the first-round revision. We appreciate the critical review that has motivated us to improve the analysis, but we disagree with those comments that suggest the regional model is the only valid approach. Much of what the reviewer writes seems to imply that one should always use the model following the "best" practice shaped for prediction tasks (i.e., achieving the highest performance/predictive accuracy), independent of the research questions, such as "*no matter what you want to do with a model, you should always set up a model in the state-of-the-art for whatever model you use*". We strongly disagree with this presumption. A regional model and a local model can be considered different modeling choices in this application, and we deliberately chose a local model as it is more suitable for our research question. In particular, a local model allows a relatively straightforward interpretation of the flood-generating process at the catchment level. Using a regional model, however, would have completely changed the paper and diverted attention from the main objective of our study: identifying flooding mechanisms from the local and interpretable relationship between meteorological drivers and flood responses.

The reviewer argues a regional model would be a better choice because it captures relationships that can be generalized to ungauged basins and unprecedented events. This may be true, but we cannot agree with the implication that only such a model (i.e., using one mixed-effects framework to fit all available data) can be used to gain physical insights. In general, from a scientific perspective, even parsimonious models can help generate scientific insights. Otherwise, a lot of simplified models (intermediate complexity climate models, integrated assessment models, simple correlation analyses, etc.) would not exist. Importantly, in our case, we decided not to change the paper towards using a regional-modeling approach because employing such a model would require introducing catchment attributes in the modelling, which could confound the interpretation of local flooding mechanisms and therefore make physical insights uncertain or less concise. This was explained in the manuscript and in the previous rebuttal, but it seems the reviewer missed that point. We agree that a regional model may also generate interesting scientific insights related to a different research question than what we ask in our work, and we had planned to employ such a regional approach for future work to avoid overloading the paper.

The same argument as above holds for the selection of input features. Since our goal is interpretability, we restricted ourselves to a few input features whose effects can be relatively

easily interpreted and linked to fundamental physical processes. In fact, as we explicitly expressed in the first-round revision, we do not negate the value of including more input features to potentially discover unknown patterns, but a comprehensive investigation of these unknown patterns and mechanisms is outside the scope of the present study.

For the problem of cross-validation, we do admit that the first version may not have addressed the concern raised by the reviewer. In this round, we reran the models with a more rigorous data split approach and update the results throughout the manuscript. Despite minor changes in most reported numbers, all previous conclusions still hold. However, we appreciate the reviewer's comment, which has helped to make the conclusions more robust.

Overall, we respect and appreciate the comments from Reviewer 2 and try to reconcile the specific research question in the present study with the expectation of using a regional model with more variables, which is undoubtedly our long-term goal of using interpretive deep learning. In response to the suggestions, we tried to address the concerns without losing focus or without the paper becoming unnecessarily long. It is our hope that the revised version will be positively received.

**Response to Reviewer 1**

**Reviewer 1**: I have received a revised manuscript of Jiang et al. In the revised version the authors have addressed most of my main concerns. I think the manuscript has been considerably improved, I have only several editorial suggestions.

**Response:** We are grateful to Reviewer 1 for providing positive comments for the present paper and good suggestions. The responses to the comments follow.

**Reviewer 1**: Line 37: should it be Stein et al 2020 instead of Stein et al 2021?

**Response:** Thank you for pointing it out. Yes, it should be Stein et al 2020 and we have corrected it. (**line 37**)

**Reviewer 1**: Line 97: 20,455 time steps

**Response:** Added as suggested. (**line 97**)

**Reviewer 1**: Line 246: The selection of catchments is not clear here. Please clarify on which basis they were selected and attributed to different regions.

**Response:** We have rewritten the original sentence as "*Moreover, in order to analyze the possible causes of trends, we selected a number of regions where most catchments present consistent trends in certain mechanisms. We investigated those catchments exhibiting significant changes in flooding mechanisms and compared the temporal regional changes in flooding mechanisms with changes in potential flooding drivers.*" (**lines 246-248**)

**Reviewer 1**: Figure 7: Please clarify in the caption how confidence intervals were computed.

**Response:** In the caption, we added "*The shades denote the 95% confidence interval of the proportions, which was calculated as* $\hat{p} \pm 1.96 \times \sqrt{\frac{\hat{p}\,(1-\hat{p})}{n}}$ *($\hat{p}$ is the estimated proportion and n is the sample size).*" (**lines 477-478**)

**Reviewer 1**: Figure 8: I suggest to add 25th and 75th percentiles to indicate spatial variability within regions.

**Response:** Thank you for the suggestion. On the basis of Figure 8, we additionally added new **Figure S6** to show the 25th and 75th percentiles of changes in meteorological drivers. To refer to the figure in the main text, we added "*These figures are robust against spatial variability within regions (see Fig. S6 in the Supplementary Material).*" (**line 539**)

[Figure]

*(new)* **Figure S6**. *The temporal changes of the event-level mechanisms in relevant catchments within the five selected regions (see Fig. 7c in the main text), as well as the changes in average extreme precipitation, mean spring temperatures, and antecedent soil moisture conditions prior to flooding. The notions are the same in Fig. 8 in the main text, except for the shades that further illustrate the 25th and 75th percentiles of the yearly changes in respective meteorological drivers (smoothed by a 20-year moving average window as well).*

**Response to Reviewer 2**

**Reviewer 2**: In the first rounds of reviews, I commented on three topics:

1. Model setup (single basin LSTMs vs regional LSTMs).

2. Selection of input variables.

3. Data splits.

In most parts, the authors argued that my concerns do not require any change to their manuscript apart from adding a new section on "Limitations and Outlooks" that basically lists these concerns.

In one point (selection of input variables), I think it is a pity that the authors limit themselves to their selection and do not report potential novel insights (see detailed comment below). On another point (model setup), I do not agree with the argumentation of the authors, why they think that an inferior model calibration procedure is justified, but in the end it is their decision.

The last point (data splits), is probably the most critical. Reading that "the model was used for statistical purposes instead of a prediction task", which according to the authors justifies the use of a data split that is not independent of the training data is in my opinion a red flag for any kind of study that involves a model. Even if the results would not change (what I sincerely hope for the authors) it is a scientifically wrong thing to do. And to justify it by saying a per-gauge time split is more complicated is in my opinion a really bad argument.

I added some specific comments to the three points below but in general, I have nothing much to add to what I said in the first round. Ultimately, it is up to the editor to decide how to proceed.

**Response:** In the first place, we would like to express our appreciation for the reviewer's comments, which aimed to help improve our manuscript. We regret that the reviewer did not appreciate our response to his first-round comments and the paper's rating has been dropped from excellent/good/excellent to good/fair/good and from minor revision to major revision, even though the paper has not fundamentally changed.

During the first round of review, the reviewer made positive comments like "*To my knowledge, it is the first time that such an analysis (detecting flooding mechanisms and analyzing the change over time) is made using a) deep learning models (here LSTMs) and b) methods from the field of explainable AI (here integrated gradients). My views on LSTMs is no secret and I*

*have often said that you can do more than "just fitting streamflow records" with these models, so naturally, I am quite excited to see someone coming up with such an idea.*" and "*Additionally, I think this paper is exceptionally well written and at least for me personally, everything seemed pretty clear and reasonable. For example, the authors make a couple of assumptions (like grouping the integrated gradient signal into two groups of a) the last 7 days and b) all other days before that), but their reasoning for all these assumptions is clearly articulated and to me, they make sense.*" Overall, the reviewer gave a "minor revision", which is generally assumed to involve minor amendments rather than substantial revisions. We can assure that we had carefully considered every comment raised by the reviewer in the first round and we did not mean to address the issues half-heartedly by placing the work in "*Limitations and Outlooks*". Instead, we fully recognize the potential of using regional LSTM with more variables to help a better process understanding, which is what we had planned for future work as well. We decided not to change the manuscript in that way because it would have changed the paper into a completely different one, making it unnecessarily long, and diverting attention from the main objective of the study: identifying flooding mechanisms from the local and interpretable relationship between meteorological drivers and flood responses using a novel approach based on machine learning.

For the points of input variable selection and model setup, we argue that the criticism that "*they think that an inferior model calibration procedure is justified*" is not fair. The comment implies that an incomplete model is inferior in spite of the fact that even incomplete models can generate scientific insights as well. Model selection should follow the principle of parsimony, otherwise known as Occam's razor, which "*seeks to find an optimal trade-off between the ability of the model to fit data and the model's required complexity to do so*" (Höge et al., 2018). Of course, one could always add more predictors (including the static catchment attributes to build the regional model) to increase model accuracy. However, in light of the tradeoff between accuracy and interpretability, is this approach really useful for extracting concise insights about the local relationship between meteorological drivers and flood responses as intended by our study? In particular, because of the collinearity between the variables that we already have and the variables to be added, the effect of input features might be less interpretable and uncertain (see our reply to the specific comments below).

For the point of data splitting, we acknowledge that our previous practice was less rigorous. We re-ran the models with a more rigorous data split approach and updated the results throughout the manuscript. None of the conclusions have been affected by this change.

Again, we are grateful for the reviewer's comments intended to improve our work. For the comments that are beyond the scope of the specific research question in the present study, we made further clarification in the revision. We have tried to address the concerns without changing the manuscript to a completely different study and without losing its original focus.

**Reference:**

Höge, M., Wöhling, T., & Nowak, W. (2018). A primer for model selection: The decisive role of model complexity. Water Resources Research, 54, 1688– 1715.

**Reviewer 2**: 1. Model Setup

The topic of discussion is, whether it is necessary/recommended or not for this study, to use a regional training setup with LSTMs rather than the per-basin calibrated LSTMs the authors chose to use. Here you say "However, as you mentioned as well in your comment, the choice of which strategy to apply (local model vs. regional model) should be in line with the research purpose."

No, this is not what I have mentioned. I deeply believe that no matter what you want to do with a model, you should always set up a model in the state-of-the-art for whatever model you use. Again, you would probably not calibrate a conceptual hydrology model in a regional fashion if you are interested in a single basin.

There is no reason to believe that a model that is consistently and significantly better than another model (or the same model but trained with a different setup) should not have a better understanding of the underlying physical processes (i.e. flood generation).

**Response:** In the last paragraph above, it seems that the reviewer believes that "better" models are always equivalent to models that are capable of making better predictions. However, we disagree with this assumption. In particular, from a scientific perspective, "better" models should be models that are more suitable for the research question at hand. Hence, different purposes, i.e., research questions, will lead to different best models, even for the same data set (Tredennick et al., 2021). Consequently, we argue that it is inappropriate to ask us to adopt best practices for prediction problems (for which a regional model would be more suitable) in our study that, instead, aims at physical interpretability. In the context of our application, a regional model and a local model can be considered different modeling choices, and we deliberately chose a local model as it is more suitable for our research question. In particular, a local model allows a relatively straightforward interpretation of the flood-generating process at the catchment level.

**Reference**:
Tredennick, A. T., Hooker, G., Ellner, S. P., and Adler, P. B.. 2021. A practical guide to selecting models for exploration, inference, and prediction in ecology. Ecology 102( 6):e03336.

**Reviewer 2**: LSTMs trained to local basins suffer from two problems:

1. They witness saturation effects (which is easily visible when plotting discharge time series and e.g. comparing the max peak the model is able to output during the training period vs validation period).

2. They are unable to model any process that did not happen in that particular basin during the training period. The solution to both of these problems is training LSTMs on a multi-basin dataset, which afaik should be the one and only way how LSTMs should be applied for rainfall-runoff modeling. Looking at e.g. Frame et al. (2022), one can see how the regionally trained LSTM is able to predict unprecedented events in individual basins (e.g. being trained only on

data with q < HQ5 and then tasked to predict > HQ100), which a locally trained model will never be able to do.

**Response:** The rebuttals against using local LSTM models may be true if concerns are about their ability in predicting unprecedented events, but we must emphasize that the argument has little relevance to the objective of the present study. We are not interested in processes that didn't happen in a given catchment as we want to identify catchment-level flood processes in the observational period. It is therefore a poor argument to suggest the invalidity of local modeling for our study, which focuses on interpretability instead of prediction. Assuming the argument holds, even linear approaches should be avoided in hydrology as they cannot accommodate and capture nonlinear processes in hydrological systems. Again, high predictive accuracy cannot be the only guiding principle for model selection.

**Reviewer 2**: "With the local models, since meteorological variables are the only inputs, we are able to focus on how they explain the temporal variation of discharges. In comparison, for a regional model that is supposed to capture both temporal and spatial variations in discharge peaks, it is challenging to distinguish how meteorological variables contribute to temporal variation in flooding within catchments from how they contribute to the spatial variation in flooding across catchments."

I think I disagree with everything you have said here. First, just because something is "more challenging", that doesn't mean in my opinion that the easier (but potentially wrong/biased) approach is justified. And in more detail, I also don't see the problem of looking at flood generating processes through integrated gradients, if static features are also model input arguments. Sure, the static arguments influence the model dynamics, but so does the fact that you have different model weights for every catchment. Just as an example, imagine you have a regionally calibrated model and two different basins A and B. Now we force the model with the same timeseries of meteorological features but using the different sets of static features, corresponding to basin A and basin B. Now imagine that in one basin, the model would produce a flood peak that, by looking at the integrated gradients of the meteorological features, is driven by the precipitation of the last few days, while in the other basin it produces a flood peak that is driven by temperature of the recent days and precipitation of some weeks ago. The difference here is certainly, because the static attributes are different. But does this have an influence on your analysis? I think not, and most likely this model has a much better understanding of the underlying processes. In your case, you would identify these two peaks as generated by different processes, which is most likely correct. So sure, static features have an influence on the model dynamics, but your analysis will still tell you which meteorological inputs are the most important for the peak flow. To be entirely honest, I fail to see what is the real challenge here.

**Response:** We fear the reviewer has misunderstood what we meant by the challenge and regret not clarifying it clearly enough in the first-round revision. What we were mostly concerned about is the issue due to the collinearity between the variables we already used and the variables to be added for regional modeling. For illustration, the new **Figure S7** shows the correlation heatmap between some common static catchment attributes (indicated by purple texts), daily

mean meteorological variables (indicated by green texts), and daily mean river discharges (indicated by black texts) for the 1,077 catchments. The catchment attributes include catchment area, slope, elevation, aridity, snow fraction, drainage density, number of dams, and storage volumes of dams. It is apparent that some catchment attributes and average meteorological drivers are highly correlated, such as slope vs. precipitation, aridity vs. precipitation, snow fraction vs. temperature, etc. The multicollinearity might not be problematic for prediction and forecasting tasks but can seriously impede interpretation, as the multicollinearity can affect the coefficients (weights) of independent variables in a way that limits the physical interpretation of the feature importance.

[Figure]

*(new)* ***Figure S7.*** *The Pearson correlation heatmap between some common static catchment attributes (the first eight attributes, written in purple), daily mean meteorological drivers (in green), and daily mean river discharges (in black) for the 1,077 catchments in the main text (see Fig. 1). The catchment size, slope, elevation, drainage density, number of dams, and storage volumes of dams were derived from the Global Streamflow Indices and Metadata Archive (GSIM, https://doi.org/10.1594/PANGAEA.887477), the aridity index and snowfall fraction were calculated from the catchment-averaged precipitation and temperature. The daily mean meteorological drivers include the daily mean value of catchment-averaged precipitation, temperature, and day length during 1950–2020. Daily mean river discharges were calculated by using the available discharge records during 1950–2020 and they have been represented in mm/d to exclude the effect by catchment size. Each grid represents the correlation between two variables across the 1,077 catchments, with the dark red or dark blue color denoting strong positive or negative correlations.*

We further disagree with the argument that including static catchment attributes will not impact the interpretation. Taking the example given by the reviewer, suppose basin A is located in a mountainous region with a steep slope, basin B is situated on a snowy plain, and peaks in basin A are overall higher than peaks in basin B. When one model predicts peak 1 in basin A and peak 2 in basin B, we would say that not only the variance in meteorological predictors, but

also the difference in catchment attributes, have explained the variance in the two peaks. In that case, it is hard to separate the effects of meteorological predictors from static catchment attributes especially when the number of catchments increases, which will come with a lot of spatial and temporal confounding. For instance, the weight of meteorological predictors will be confounded by catchment attributes due to their interactions. For the reasons outlined above, we believe local modeling is more suitable for the present study, which aims to make inferences about local catchments and avoids confounding and multicollinearity resulting from static catchment attributes.

Note here that we do not deny the value of regional modeling for making inferences. Instead, we do recognize that a regional model may also generate interesting scientific insights related to a different research question, and we had planned this for future work to avoid overloading the paper.

**Reviewer 2**: "...some studies have suggested that training a regional model for all catchments at once may be a better practice (e.g., Nearing et al., 2021)."

I come back to this point, because it sounds like this is just a "suggestion". Let me tell you from experience from the operational side (not only at Google but various companies/agencies) but also from literature: I am not aware of anyone that applies LSTMs that is not using regionally trained LSTMs. In my view, there is really no discussion happening around whether you should use regionally or single basin calibrated models. Again, it is not only about better performance but also why the model can achieve this higher performance (see list above). I don't want to step on someone's toes but from my point of view, the reason why people are sticking to single basin LSTMs is because it was hammered into their head for decades that single basin models are better at capturing local processes than regional models, which is true for conceptual models but it is not anymore for LSTM-based models.

**Response:** Again, we agree that regional modeling is better suited for prediction tasks, as was the practice in industry and a lot of the literature. In the present study, however, reliable inference on the effect of predictors is prioritized, which means we have to consider multicollinearity, confounders, etc. besides accuracy. These factors might not affect the predictive accuracy of regression methods, but they would impair interpretation. Basically, the fact we choose local modeling is a consequence of the objective of the study.

To better clarify the reasons why we chose local modeling instead of regional modeling, as well as the real challenge for future studies of using regional modeling for the investigation, we rewrote the discussion in the section "*Limitations and outlooks*":

"*In this study, we trained LSTM models in a local fashion (i.e., training the model individually for each catchment), rather than a regional fashion (training a single model across multiple catchments), since the main objective of the study is to identify distinguishable patterns of meteorological variables' contributions at local scales. From a prediction standpoint, particularly for unprecedented events and ungauged basins (Nearing et al., 2021; Frame et al., 2022), regional modeling may be a better choice because it is capable of learning more*

*general relationships from a larger variety of hydrological data (Kratzert et al., 2019b).
However, for the regional modeling, both meteorological time series and static catchment
attributes are used as inputs to distinguish response behaviors across time and space. Adding
such static attributes would introduce substantial multicollinearities among the considered
variables (see Fig. S7 in the Supplementary Material for illustration). Multicollinearity might
not be a problem for ML models when they are used for prediction, as long as the collinearity
between variables remains stationary (Dormann et al., 2013). Nevertheless, for our study that
aims to interpret the effects of predictors on responses, high multicollinearity in predictors
indicates considerable information may be shared among the collinear sets. This would result
in difficulties in separating the physical effects of these variables – this is also the case in
traditional regression models (Hartono et al., 2020). Therefore, interpreting flooding
mechanisms with regional LSTM models may become more challenging than with local LSTM
models that use only meteorological time series, since some catchment attributes would
confound the interpretation. In this study, we therefore employed simple local models, which
avoids confounding and multicollinearity resulting from static catchment attributes. However,
in light of the benefit of regional modeling that can provide insights into how flooding
mechanisms vary spatially by geographic and climatic characteristics of catchments, how to
deal with these challenges in the interpretation merits more exploration in future studies.*"
(**lines 565-582**)

**References:**

Nearing, G. S., Kratzert, F., Sampson, A. K., Pelissier, C. S., Klotz, D., Frame, J. M., Prieto, C., and Gupta, H. V.: What Role Does Hydrological Science Play in the Age of Machine Learning?, Water Resources Research, 57, 2021.

Frame, J. M., Kratzert, F., Klotz, D., Gauch, M., Shalev, G., Gilon, O., Qualls, L. M., Gupta, H. V., and Nearing, G. S.: Deep learning rainfall–runoff predictions of extreme events, Hydrol. Earth Syst. Sci., 26, 3377-3392, 2022.

Kratzert, F., Klotz, D., Shalev, G., Klambauer, G., Hochreiter, S., and Nearing, G.: Towards learning universal, regional, and local hydrological behaviors via machine learning applied to large-sample datasets, Hydrology and Earth System Sciences, 23, 5089–5110, 2019b.

Dormann, C. F., Elith, J., Bacher, S., Buchmann, C., Carl, G., Carré, G., Marquéz, J. R. G., Gruber, B., Lafourcade, B., Leitão, P. J., Münkemüller, T., McClean, C., Osborne, P. E., Reineking, B., Schröder, B., Skidmore, A. K., Zurell, D., and Lautenbach, S.: Collinearity: a review of methods to deal with it and a simulation study evaluating their performance, Ecography, 36, 27-46, 2013.

Hartono, N. T. P., Thapa, J., Tiihonen, A., Oviedo, F., Batali, C., Yoo, J. J., Liu, Z., Li, R., Marrón, D. F., Bawendi, M. G., Buonassisi, T., and Sun, S.: How machine learning can help select capping layers to suppress perovskite degradation, Nat. Commun., 11, 4172, 2020.

**Reviewer 2:** 2. Input feature selection

"We agree that including more inputs is beneficial to uncovering patterns related to flood mechanisms that are likely to be overlooked. During our preliminary tests, we had run models with daily averaged sea level pressure, relative humidity, and radiation as additional inputs, which did indeed lead to more clusters in terms of feature importance patterns"

Reading that you already did these experiments, which led to different results, but decided to not include them because they probably don't agree with the "known" patterns is in my opinion simply sad. I agree with the first reviewer on something he mentioned in a slightly different context, which is that you artificially limit yourself in this study. You have the potential to report new findings and even if you can't explain them, you could still present them and potentially start a new discussion in this area. Instead you limit yourself and the model to only check if your method identifies the same patterns as previous studies with different approaches.

"The performance did not drop much by not including more variables" I think this is not important here, but as you said yourself, the clusters of flood generating processes changed. So which one is "the truth"? I have nothing to add to this point and ultimately, it is the decision of the authors. I think it is just a missed opportunity and generally, I am not happy with the fact that all major reviewer comments are put into a new "Limitations and outlooks" section and that this suffices.

**Response:** The same argument as above holds for the selection of input features. Performance is not everything, and our goal is interpretability, which is not necessarily improved by more input features. For example, using the rigorous data splitting strategy in line with the third major concern, we have re-run the models by including daily averaged sea level pressure, relative humidity, and global radiation as inputs in addition to precipitation, temperature, and day length we used. In this round, adding the three kinds of variables still did not substantially improve the predictive accuracy. The mean NSE values for testing data in the 10-fold cross-validation changed from 0.641 to 0.644, while the median dropped from 0.716 to 0.702. We argue that the fact that performance did not change much is not unimportant here. Instead, it implies the newly added input features may not provide additional information to the model prediction. This can be validated by the new **Figure S8**, which shows the average of normalized aggregated contributions for the respective input variables in each catchment. The results indicate that compared to precipitation, temperature, and day length, the new input features do not show substantial contributions.

[Figure]

*(new)* **Figure S8**. *The average normalized aggregated contribution for respective input variables in each catchment if sea level pressure, relative humidity, and global radiation were added into the model in the main text in addition to precipitation, temperature, and day length. The sea level pressure, relative humidity, and global radiation were retrieved from E-OBS dataset (cited in the main text) and they were processed as catchment-averaged time series by the method described in the main text. Model settings and interpretations are also the same as in the main text, where only precipitation, temperature, and day length are used. Here the aggregated contribution indicates the contributions of a variable in all the 180 days to the target peak discharges (i.e., $\Sigma_1^{180}\bar{\phi}_i$, see the notions in Section 2.3 in the main text). The aggregated contribution for each variable has been normalized per peak discharge for comparability. The color reflects the average value of the normalized aggregated contribution for respective input variables in each catchment. Darker colors indicate that the specific variables have a greater impact (either positive or negative) on peak discharges. The figure implies that the sea level pressure, relative humidity, and global radiation are less important features.*

In spite of these interpretation results, we won't assert that the three variables have no effect on flooding, since the used interpretation technique (i.e., integrated gradients) does not measure how important a feature is in the real world, but simply how important a feature is to the model. It is perhaps the multicollinearity in the variables that makes some variables (e.g., radiation)

redundant to the model, but this might not be the case in reality. As a result, the interpretation results would need to be handled very carefully if no adequate physical knowledge exists to justify their inclusion. Therefore, instead of using more predictors that result in less interpretability, we restricted ourselves to few input features (temperature, precipitation, and day length) whose effect can be relatively easily interpreted and understood. We do admit it is a limitation of the present study and we have not negated the value of including more input features to potentially discover unknown patterns, as we already stated in the "*Limitations and outlooks*" section. However, our manuscript is not intended to cover everything about unknown patterns and implied mechanisms, for which a comprehensive investigation and understanding would fall outside the scope of the present study. For better clarification, we improved our statement in the "*Limitations and outlooks*", which now reads:

"*The multicollinearity also exists in meteorological drivers at daily scales, which requires careful handling of the interpretation results if adding more predictors. For example, radiation is usually an important driver of snowmelt that favors flooding (Merz and Blöschl, 2003), but the interpretation method might not assign it high importance when it is combined with day length as an additional predictor due to the high correlation between the two variables (see Fig. S8 in the Supplementary Material for an example). This is because the used interpretation technique does not measure how important a feature is in the real world, but how important it is to the model. Therefore, it is not necessarily better to add more input features to a model in terms of process understanding, which can be even misleading if the interpretation results are not justified by sufficient physical knowledge (Kroll and Song, 2013). In this study, instead of using more predictors that result in less interpretability, we restricted ourselves to few input features whose effect can be relatively easily interpreted and understood. Therefore, we only selected daily precipitation, temperature, and day length as meteorological inputs, the combination of which results in uncovering three well-known flooding mechanisms. The results are physically interpretable and comparable with findings from other studies that used classical methods. Incorporating more meteorological drivers into the model might, in theory, allow for the identification of additional flooding mechanisms that may be overlooked. However, multicollinearity and confounding can pose a challenge to interpretability, especially when the recognized patterns cannot be linked to fundamental physical processes. Therefore, we leave how to resolve the trade-off as an open question for future studies.*" (**lines 584-598**)

**Reference**:

Merz, R. and Blöschl, G.: A process typology of regional floods, Water Resources Research, 39, 2003.

Kroll, C. N. and Song, P.: Impact of multicollinearity on small sample hydrologic regression models, Water Resources Research, 49, 3756-3769, 2013.

**Reviewer 2:** 2. Data split I think this is probably the most worrisome point to me. I think the applied data splitting (random in time) introduces a severe data leakage. You are not testing your method on unseen data, not even in these cases where you predict on a "test sample". The

reason is that if you randomly select timesteps to be train/test data then three adjacent timesteps could e.g. be "train - test - train". Since each timestep is predicted from an input sequence of 180 days, e.g. the first train sample and the test sample only differ in a single time step of data. And without any doubt, the discharge data is highly auto-correlated. In Figure S1-S3, it is e.g. possible that for the models for which this time step appears to be in the test data (dashed lines) the previous and next timestep of that event is a training step. And in this case you can not at all argue that this is independent test data. And that these plots show that the signal is the same for all 10-folds could be just because of this effect, because there is not really any point in your input time series that wasn't seen during training of any of these models.

"Firstly, runoff data available in the GRDC dataset is not temporally complete in many catchments in Europe, with missing data sometimes occurring for several months or years irregularly. This complicates carrying out a unified temporal k-fold cross-validation across these catchments." This is related to what I wrote above: Just because something is "more complicated", this doesn't mean you shouldn't do it. In fact, it isn't that hard to loop over basins and do time series splits per basin. How difficult it is to include different data splits for each basin in your training pipeline depends on your code. It is a built-in function in the open source library NeuralHydrology (https://github.com/neuralhydrology/neuralhydrology/ Disclaimer: I am one of the developers) and would work out of the box if you would use this for training your models.

"We should emphasize that the model was used for statistical purposes instead of a prediction task, thus the split of the training dataset and the testing dataset is only to ensure the model has learned a generalizable relationship between variables." I don't understand your point here. For any kind of statistical analysis with any model (data driven or not) you want to make the analysis on an independent dataset. The point is, that your test dataset is not independent of the training dataset as there is data leakage.

"The generalizable relationship should hold not only for the testing dataset but also for the training dataset." This is not necessarily true. First, in the extreme case you could overfit on the training data, meaning your model remembers every sample and thus is not generalizing at all. Second, have you ever looked at e.g. the NSE of an LSTM during the training period and compared this to the test period (with a non-random splitting). You will see that the LSTM achieves a much higher NSE during the training period and I would be more than cautious to draw any conclusions from this on the models generalization capabilities.

**Response:** On this point, we agree that the previous random splitting approach is less rigorous than the one suggested by the reviewer. Therefore, we re-ran our models with rigorous 10-fold cross-validation. In **lines 164-166**, the method description was modified as "*To improve the robustness of model evaluation and analysis, we fitted 10 independent LSTM models for each of the 1,077 catchments. Specifically, the data for each catchment was divided into 10 folds without shuffling the temporal sequence, and each fold was tested once with a model trained with the remaining 9 folds.*"

In the more rigorous experiments, the median NSE for the 1,077 catchments drops to 0.72 and the number of catchments to be analyzed is now 943 (using an average NSE above 0.5 in the testing periods as a criterion). However, the conclusions about the proportion and trends

(including the overall trends and those in specific hotspots) are consistent with the original manuscript, except for the minor changes in numbers (see revised Figures 5-8 below). Besides, using the IG scores based on the peaks in testing datasets alone does not yield substantial impacts on our conclusion in subsequent analyses, either (see new **Figure S4-S5** below). In the revision, we have updated the manuscript throughout with the new results.

[Figure]

*(revised)* ***Figure 5****: The cluster centroids and variance for the three clusters and their respective proportions of all peak discharge events in each catchment….*

[Figure]

*(revised)* ***Figure 6****: The dominant flooding mechanisms and their relevance to catchment attributes and seasonality….*

[Figure]

*(revised)* **Figure 7**…

[Figure]

*(revised)* **Figure 8**: *The temporal changes of the event-level mechanisms in relevant catchments within the five selected regions*…

[Figure]

*(new)* ***Figure S4****: The same case as in Fig. 5 in the main text, but we use the IG scores based on the peaks in testing datasets alone to perform cluster analysis. The events identified with snowmelt, recent precipitation, or antecedent precipitation as the primary causes account for 16.6%, 47.7%, and 35.7% of all the 52,247 annual maximum peak discharges, which is only slightly different from using the averaged IG scores from the 10-fold models for individual peaks.*

[Figure]

*(new)* **Figure S5**: *The same case as in Fig. 7 in the main text, but we use the IG scores based on the peaks in testing datasets alone to perform cluster analysis and trend analysis.*

---

## Author Response (AR3)

**The line numbers indicated here are consistent with those in the CLEAN (i.e., no changes tracked) version of the revised manuscript.**

**Response to Editor**

**Editor:** Thank you very much for your detailed responses to the reviewers' comments. Because of the disagreement of one of the former reviewers with the suitability of the (local) modeling strategy chosen, I looked for a third person to review your manuscript. This new reviewer agrees with the previous two reviewers and myself that the study is of interest to the HESS readership. They also reflect on the suitability of the local modeling approach chosen and think that the local approach is suitable to answer the research question at hand. Therefore, we will conclude this discussion and 'accept' the local modelling strategy as the strategy of your choice. The previous reviewer also criticized the data splitting. The new reviewer does not comment on this issue but I personally think that this point has sufficiently been addressed by testing the alternative splitting approach and presenting its results in the Supplementary Materials. The comments that remain to be addressed are the two minor comments by the new reviewer. Thanks for addressing these and I am looking forward to reading the next (and hopefully final) version of your manuscript.

**Response:** We are grateful to the editor for providing us with the opportunity to refute/clarify some arguments, which have significantly improved the manuscript and inspired new research questions. We also appreciate the three reviewers for their positive comments and constructive suggestions. In the revision, we made appropriate modifications and clarifications for the new comments.

**Response to Reviewer 2**

**Reviewer 2:** Dear authors, I am thankful for your responses and I want to say that I am sorry if I appear to be the "mean reviewer 2" in this revision process.

Let me start with an TLDR;

● I think we can stop arguing here and conclude that we need to agree to disagree. I think I could go on forever but it probably adds little to this review process. So to save us all time, I will stop arguing after this review and I only reply here to clarify a few of the statements/misunderstandings.

● That being said, I think the first point, regarding data splitting (TLDR; use the results of the test splits in the paper, not the averaged results of training periods) should be taken seriously.

● One last thing: I mentioned this to Shijie at the EGU in personal communication and I don't want this review process to negatively impact what I said: I think this is an exciting study and I am happy to see people using LSTMs for these kinds of studies. I think it would have been easy

to make this paper even more exciting but I understand that I need to wait for another publication from your group to get answers to my questions.

**Response:** Once again, we wish to express our gratitude to Reviewer 2. We thought it was an honest and thought-provoking debate/arguing, which helped the formulation of new research questions for us. The reviewer's expertise was greatly appreciated, his thoughtful insights in the modeling would benefit not only this study but also our following ones.

**Reviewer 2:** Data splitting – I am very happy to hear that running your analysis only on the test splits did barely change the results. What I don't understand is, why you did not change the manuscript so that these results are used in the paper. I don't think that I need to search for literature here to say that in any modeling study, you should use independent test data (which you have!) to analyze/interpret your model. We already know that this is not affecting anything in your results, so it should be a no-brainer to change this and standard practices in statistical analysis with models.

**Response:** We are grateful to the reviewer for helping to make the analysis more rigorous and reasonable. In the last revision, we have updated the analysis results based on the new splits with the reviewer's comments.

**Reviewer 2:** Input selection – I think it is somewhat funny that on this point you argue with NSE performance (that it did not or did negatively affect the NSE, when adding more inputs), while in the other points (e.g. multi-basin model) you say that performance is not really important. Anyway, there is not much to add from my side, I think it would have been a great addition to the paper to include and analyze more input features, especially because it seemed like you already did the experiments. You yourself said in the first rebuttal "During our preliminary tests, we had run models with daily averaged sea level pressure, relative humidity, and radiation as additional inputs, which did indeed lead to more clusters in terms of feature importance patterns", which to me sounded exciting and I was curious to hear more about clusters that are different from previous literature and about interpretations/hypotheses of these clusters. You decide to keep it simple, your choice. If I understand you correctly you want to come back to this in a future publication and I am looking forward to diving into the results then.

**Response**: As we responded in the last revision, we did not deny the value of adding more input variables for possibly new insights from interpretations/hypotheses of these clusters. What we were concerned about is that multicollinearity in meteorological drivers at daily scales is likely causing instability in model interpretation, making the clustering less robust. In spite of the challenges, we will attempt to provide a definite answer to the exciting problem in a future publication.

**Reviewer 2:** Model setup – Let us just say that we disagree. I certainly do believe that, no-matter what you want to do, if you use a tool, you should use that tool correctly. Following your argument (which tries to say the other extreme is possible) I could make such a study with uncalibrated hydrology models and then say that my results have any meaning?!

From a very basic point of view: I hope we can agree that a model that is better in modeling the task at hand (meteorological forcings in, discharge out) should have a better understanding of the underlying system/processes, right? Now if I want to make a study that investigates the underlying process understanding of a model, then why would I willingly pick a model that has a worse understanding of the underlying processes? If the model would not have a worse process understanding, it would not be generally worse in predicting discharge, or would it? Since your study is about model interpretation of process understanding, I do not see how this is decoupled from performance. But I see how this example will be used again to say "he only cares about performance", which is wrong. But I'm not so naive to think that good process understanding is decoupled from good performance.

Your line of argument is the ease of interpretation, I got it. But what I am saying is that this is a bad argument. And again, this is not "because the regional model has a higher NSE", but because, without any doubt, this model has a better process understanding. And your study is about analyzing such a process (flood generation). Funnily enough, the one thing where single-basin models have their biggest bias (compared to regional models) is flood peak prediction, which I also mentioned in my previous review (see point on saturation). One could argue that having a (negatively affecting bias) in flood predictions is also affecting the flood generating processes in the model

**Response**: We agree with the arguments that regional modeling is better in some cases, most of which we have already responded to in the previous revisions. However, we think the reviewer seems to have still ignored or misunderstood our concerns about regional modeling. We were not susceptive to the ability of regional modeling can have a better underlying process understanding, the challenge we thought of is how to disentangle the roles of meteorological drivers and catchment attributes given the possible confounding and multicollinearity resulting from static catchment attributes. In this study, we did not intend to address these challenges since they were beyond its scope, but we will have an exploration of in our next studies, as we stated in previously revised manuscript, "*in light of the benefit of regional modeling that can provide insights into how flooding mechanisms vary spatially by geographic and climatic characteristics of catchments, how to deal with these challenges in the interpretation merits more exploration in future studies*". We thank the reviewer for bringing it up.

**Response to Reviewer 3**

**Reviewer 3:** This study applied the explainable machine learning method to examine river flooding mechanisms between meteorological forcings and streamflow response. They worked on large sample European catchments, associated the patterns of three identified mechanisms with hydrological processes and further investigated the temporal trends. The ms is well-written, and I am fascinated by the topic and related analyses from hydrological perspective.

**Response:** We appreciate the positive comments from the Anonymous Referee.

**Reviewer 3:** Reading through the previous rounds of reviews and responses, I find the main question discussed is the impacts of using locally trained LSTMs (one LSTM for each catchment) versus regional LSTMs (one LSTM for all catchments with attributes used) on this study. In other words, is it appropriate to use locally trained models here to identify flooding mechanisms? I have been thinking about this interesting question for some time. Training regional LSTM with attributes is certainly a more appropriate way to get better performance, which has been shown in previous studies as the reviewer mentioned. Let's put these two types of models in the context of this study. For local LSTM, the streamflow responses only rely on meteorological forcings which totally determine the gradient contributions. Given that the local model has a performance gap to the regional LSTM model, we can safely infer that the used attributes play an important role in regional modeling and fairly contribute to the gradient dynamics. This implies that the gradient contributions of meteorological variables would behave differently in the local and regional models. As expected, the authors mentioned that different gradient patterns emerge when testing regional modeling.

It's interesting to think why there exist different gradient patterns between two modeling forms and which one is more reliable. Back to this study, the concerning question is whether the analyses done in one specific form are still valid given these differences. In my view this should be evaluated by the consistency between the identified mechanisms with our established domain knowledge. Reading through the authors' hydrological interpretations of identified mechanisms from ML, my feeling is that the results and related analyses are convincing to safely draw the conclusions. However, I to some extent agree with the point that one reviewer mentioned, due to not reporting these differences, the authors kind of limit themselves in the range of prior knowledge. Emerging differences may give us chances to better understand these models and learn new knowledge. Therefore, it would be nice to report some identified differences caused by training forms in the paper to inform readers and educe a more thorough examination in the future study. These are my thoughts and hopefully they can help to understand this question.

**Response:** Thanks to the reviewer for providing thoughtful comments and helpful suggestions. We agree that catchment attributes would play an important role in regional modeling and affect

the gradient dynamics in some way. Also, we agree that it would be interesting to examine the impact of modeling strategy (local vs. regional) on the interpretation of the models and ultimately on the understanding of flooding mechanisms. In spite of this, we prefer not to elaborate on differences in the manuscript that have only been identified preliminarily, in order to avoid jumping to a hasty conclusion and losing the original focus of the current study. As we indicated in the previous revisions, we have already planned to have a systematic examination and comparison in our next studies to give a detailed and insightful answer to this open question.

We appreciate the reviewers' suggestions about informing readers and stimulating a deeper examination. In **lines 588-592**, we supplemented the following statement, "*An immediate question to address is whether adopting different modeling strategies will result in different interpretations regarding the gradient contributions of meteorological forcings, which ultimately leads to alternative understandings of flooding mechanisms. The emerging differences may provide us with an opportunity to gain new insights into flooding mechanisms from these models.*"

**Reviewer 3:** Another point that concerns me is, in line 155, the authors state forcings over the past 180 days are used to predict the following day. The correct setup should be using forcings of 180 days to predict the last day's streamflow (180th, here larger number means the most recent time), not the next day (181st). The same-day precipitation is very important to predict the streamflow and should not be missed.

**Response:** Many thanks for the good suggestion. We agree with the reviewer that same-day precipitation is very important to predict streamflow. Our previous practice followed the common practice of using LSTM in rainfall-runoff prediction, which aims to build a predictive relationship between past meteorological data and the discharge the following day. Another consideration was the possible mismatch between the time resolution of precipitation and discharge, one of which is daily and the other may be only quasi-daily (depending on when the discharges were recorded every day in a gauge). Therefore, the prediction models only took into account the lagged meteorological forcings up till the day before each daily discharge.

However, taking precipitation on the day of the flood peak into account can be more appropriate for the purpose of flood classification, which would also be in line with common practice in similar studies. Therefore, we updated our results in the manuscript by using the new input series (which still are minor changes in reported numbers, and all previous conclusions hold). Moreover, we added the following statement for clarification in the revision, "*Note that we included predictors on the same day as the output in the model, since precipitation on that day could also affect the discharge, especially in small catchments with quick catchment response times. However, the conclusions do not change even if using LSTM models to predict discharge on the next day (i.e., the prediction models consider the lagged meteorological forcings up till the day before each daily discharge)*" (**lines 157-160**).

**Reviewer 3:** I am also a little confused at the separation of training and testing period. Line 258 mentioned "NSE value computed in the testing period". What's the testing period in time dimension? I didn't find a specific time span mentioned in the main text.

**Response:** As a result of the 10-fold cross-validation, the testing period is 1/10 the sample size of each catchment, while the exact length differs due to variable sample sizes of catchments. For clarification, we added "*The predictive performance of each model was evaluated independently based on testing data, i.e., 1/10 of the data for each catchment, which ranged from 2 to 7 years due to the 20-70 years of sample size available in studied catchments*" in the Methodology (**lines 170-172**).